# Multi-domain and complex protein structure prediction using inter-domain interactions from deep learning

Yuhao Xia[1,2], Kailong Zhao[1,2], Dong Liu[1], Xiaogen Zhou [1] & Guijun Zhang [1✉]

Accurately capturing domain-domain interactions is key to understanding protein function and designing structure-based drugs. Although AlphaFold2 has made a breakthrough on single domain, it should be noted that the structure modeling for multi-domain protein and complex remains a challenge. In this study, we developed a multi-domain and complex structure assembly protocol, named DeepAssembly, based on domain segmentation and single domain modeling algorithms. Firstly, DeepAssembly uses a population-based evolutionary algorithm to assemble multi-domain proteins by inter-domain interactions inferred from a developed deep learning network. Secondly, protein complexes are assembled by means of domains rather than chains using DeepAssembly. Experimental results show that on 219 multi-domain proteins, the average inter-domain distance precision by DeepAssembly is 22.7% higher than that of AlphaFold2. Moreover, DeepAssembly improves accuracy by 13.1% for 164 multi-domain structures with low confidence deposited in AlphaFold database. We apply DeepAssembly for the prediction of 247 heterodimers. We find that DeepAssembly successfully predicts the interface (DockQ ≥ 0.23) for 32.4% of the dimers, suggesting a lighter way to assemble complex structures by treating domains as assembly units and using inter-domain interactions learned from monomer structures.

[1] College of Information Engineering, Zhejiang University of Technology, HangZhou 310023, China. [2] These authors contributed equally: Yuhao Xia, Kailong Zhao. ✉email: zgj@zjut.edu.cn

Most proteins in nature are composed of multiple domains that represent compact and independent folding units within the protein structure. Appropriate inter-domain interactions are essential to facilitate the implementation of multiple functions in a cooperative way[1,2]. Meanwhile, structure-based drug design often relies on the interaction of different domains. Capturing the domain-domain orientation accurately is crucial to elucidate their functions and facilitate drug discovery[3]. However, multi-domain proteins and complexes are often more flexible than single domains, with a high degree of freedom in the linker or interaction region connecting the domain structure[4]. This flexibility of inter-domain orientation is a challenge both for experimental and computational methods.

Recently, great progress has been made in protein structure prediction due to the introduction of deep learning techniques[5–8]. In particular, the end-to-end protein structure prediction framework, AlphaFold2[9], built on the attention-based equivariant transformer network, was able to predict unknown structures of proteins with an unprecedented accuracy as witnessed in the CASP14 experiment[10]. With AlphaFold2, achieving a global distance test (GDT) score of 90 or higher for most targets is within reach, approaching experimental accuracy. Nonetheless, it is worth noting that several of the CASP14 targets, especially multi-domain targets, were not predicted to a 90 GDT score suggesting that there are further improvements in multi-domain prediction[3]. In fact, just about a third of proteins in the Protein Data Bank[11] (PDB) contain multi-domain structures[12]. The lack of multi-domain structures makes the PDB database biased toward proteins that are easy to crystallize and single domain structures. This means that AlphaFold2 is more biased toward single domain prediction as it is trained on the PDB. AlphaFold2 structure predictions for multi-domain proteins are thus less accurate as on the domain level[3].

Fortunately, the single domain prediction is largely solved by AlphaFold2, which provides a feasible approach to multi-domain protein structure modeling, that is, based on a divide-and-conquer strategy, splitting the sequence into domains, generating models for each individual domain, and finally assembling each domain model into a full-length model[1,4]. This approach may further improve the accuracy of multi-domain protein modeling by focusing on capturing inter-domain interactions on the basis of high precision single domain models, and to some extent lower the threshold of computing hardware compared to end-to-end methods.

Often, studies in modeling multi-domain proteins by assembling domain structures can be divided into two categories, linker-based domain assembly and inter-domain rigid body docking. Linker-based methods, such as Rosetta[13] and AIDA[14], primarily focus on the construction of linker models by exploring the conformational space, with domain orientations loosely constrained by physical potential from generic hydrophobic interactions. Meanwhile, domain assembly can also be perceived as a docking problem. The previously proposed DEMO[1] and SADA[15] assemble the single domain structure by rigid body docking, which are essentially a template-based method that guides domain assembly by detecting available templates. However, structural alignment is limited by the number of multi-domain proteins in PDB, and the difficulty of capturing the orientation between domains from the template may increase as the number of domains increases[15]. The inter-residue distance predicted by deep learning improves the problem of insufficient number of multi-domain protein templates to a certain extent. But given the fact that it is currently relatively easy to obtain high-precision single domain structures, it may be more urgent and important to pay more attention to the capture of inter-domain interactions in deep learning.

In addition to multi-domain targets, could protein complex structure prediction be achieved by domain assembly? Considering that intra-protein domain-domain interactions are not physically different from inter-protein interactions; their structure or function may be viewed essentially as the embodiment of inter-domain interactions. We hypothesize that inter-domain interactions learned from intra-protein could be used to predict the protein complex structure. In practice, many proteins that form complexes in prokaryotes are fused into long, single-chain, multi-domain proteins in eukaryotes[16,17]. The same physical forces that drive protein folding are also responsible for protein-protein associations[16,18]. Thereby, it is very likely that the deep learning model built by learning the inter-domain interactions in the existing PDB monomer structures have already learned the representations necessary to model protein complexes consisting of multiple single-chain proteins[19].

Until now, there is no study for predicting the structure of protein complexes by specifically learning inter-domain interactions and using the modeling approach of domain assembly. Conventional approaches for predicting the protein complex structure include docking[20–22] and template-based methods[23–25], which are limited by force-field accuracy and experimentally resolved multimeric structures, respectively. Recent methods incorporate coevolution-based contact/distance prediction and deep learning techniques to predict complex structures. The three-track network of the end-to-end version of RoseTTAFold[5] generates protein-protein complex structure models by combining features from discontinuous crops of the protein sequence. Meanwhile, studies have been carried out whereby AlphaFold2 is adapted to predict the protein complex structure[16,26]. They demonstrate that the same network models from AlphaFold2 developed for single protein sequences can be adapted to predict the structures of multimeric protein complexes without retraining[16]. Alternatively, an AlphaFold2 model, AlphaFold-Multimer[27], trained specifically on multimeric inputs, significantly increases the accuracy of predicted multimeric interfaces while maintaining high intra-chain accuracy. In fact, these methods are performed by feeding the combined protein sequences into the deep learning models, for some large targets, most available GPU memory may have difficulty meeting the requirements of its inference. Therefore, it is highly desirable to develop a more lightweight approach, such as predicting the structure of protein complexes through domain assembly using domains in each chain as assembly units.

In this work, we proposed a framework, DeepAssembly, to assemble multi-domain proteins through inter-domain interactions specifically predicted by deep learning. Different from DEMO and SADA, as a data-driven deep learning method, DeepAssembly avoids the time-consuming template alignment process and does not depend on templates entirely, which pay more attention to the capture of inter-domain interactions to improve the prediction accuracy of inter-domain orientations. Experimental results indicate that DeepAssembly builds models with higher accuracy than AlphaFold2 and improves the inter-domain orientations of those multi-domain protein structures with lower accuracy in AlphaFold Protein Structure Database[28] (AlphaFold database). Furthermore, we use the same network model trained on monomeric multi-domain proteins to predict protein-protein complex structures with domain assembly. It demonstrates that the proposed method, DeepAssembly, can be applied to protein complex structure prediction by using inter-domain interactions learned from intra-protein, and we provide evidence of such. Meanwhile, it also provides a more lightweight approach to assemble protein structures using domains as units, possibly reducing the GPU memory requirements to some extent.

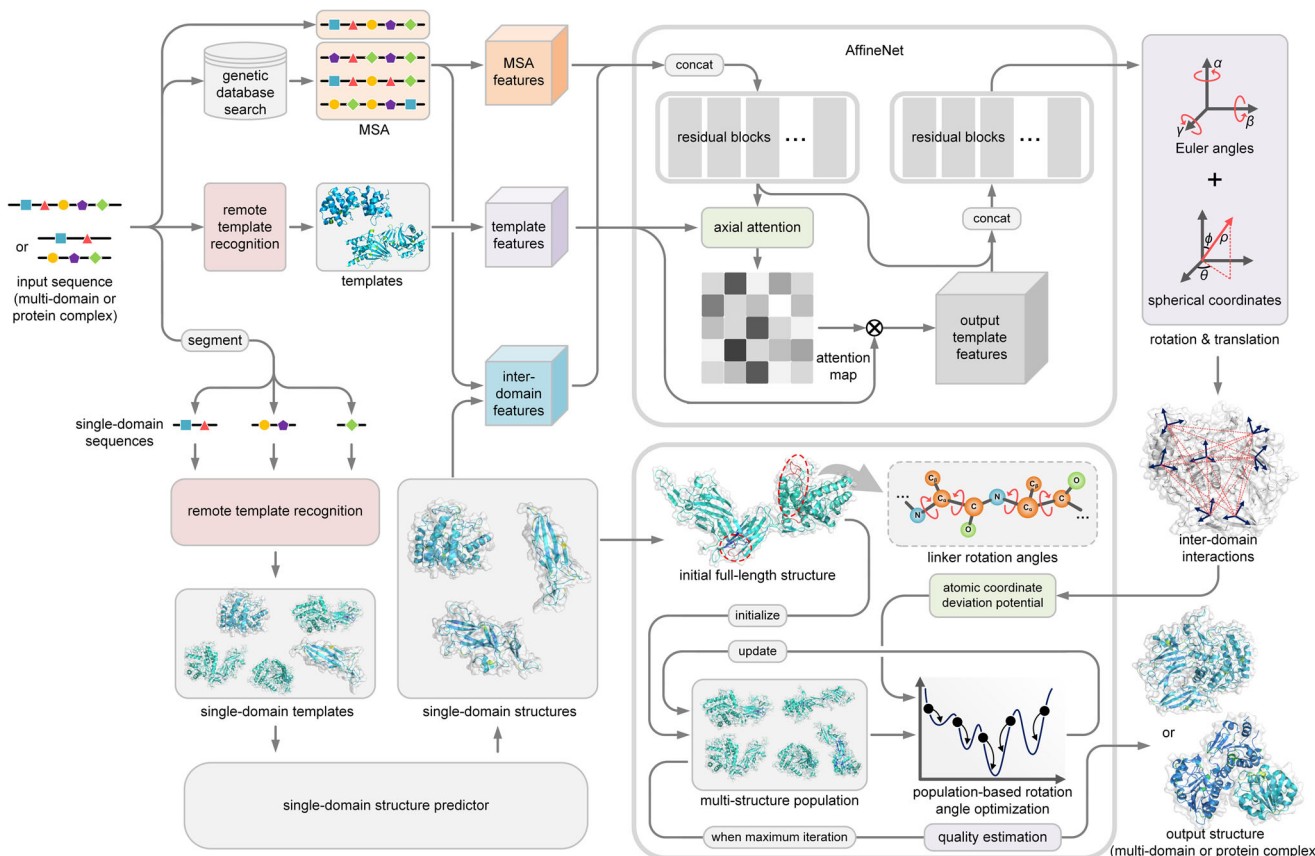

**Fig. 1 Pipeline of DeepAssembly.** The input of DeepAssembly is the sequence of multi-domain protein (or protein complex), which is segmented into single-domain sequences and generated single-domain structures by the single-domain structure predictor. Features extracted from the MSAs, templates and domain boundary information are fed into a deep neural network to predict inter-domain interactions. In the domain assembly module, the full-length structure is constructed using the predicted single-domain structures, and the population-based rotation angle optimization is performed under the guidance of an energy potential transformed from the predicted inter-domain interactions. Finally, the output structure is selected by the model quality estimation.

## Results

**Overview of the method**. DeepAssembly is designed to automatically construct multi-domain protein or complex structure through inter-domain interactions from deep learning. Figure 1 shows an overview of the DeepAssembly protocol. Starting from the input sequence of multi-domain protein (or protein complex), DeepAssembly first generates multiple sequence alignments (MSAs) from genetic databases and searches for templates using our recently developed remote template recognition method, PAthreader[29]. Meanwhile, the input sequence is split into single-domain sequences by a domain boundary predictor, and then the structure for each domain is generated by a single-domain structure predictor (remote template-enhanced AlphaFold2). Subsequently, features extracted from MSAs, templates and domain boundary information are fed into a deep neural network (AffineNet) with self-attention to predict inter-domain interactions. Finally, DeepAssembly performs the creation of the initial full-length structure using the single-domain structures, followed by iterative population-based rotation angle optimization[30–32]. The domain assembly simulation is driven by the atomic coordinate deviation potential transformed from predicted inter-domain interactions, where the best model by our in-house model quality assessment[33] is selected as the final output structure.

**Accurate predictions on multi-domain proteins by domain assembly**. We first evaluate the performance of DeepAssembly on the test set of 219 non-redundant multi-domain proteins, and compare with the end-to-end protein structure prediction method

AlphaFold2 (Supplementary Data 1). For DeepAssembly, the single-domains are individually predicted by the single-domain structure predictor (PAthreader) from the segmented sequences, as depicted in the DeepAssembly protocol. We use root-mean-square deviation (RMSD) and template modeling score[34] (TM-score) to evaluate the accuracy of the built models. Here, the TM-score is a metric defined to evaluate the topological similarity between protein structures (Supplementary Note 1), taking values (0,1], where a higher value indicates closer structural similarity. Figure 2a shows the full-chain TM-score and RMSD of the built multi-domain protein models with respect to the PDB structures. The detailed evaluation results are listed in Supplementary Table 1. On average, the models built by DeepAssembly achieve a TM-score of 0.922 and RMSD of 2.91 Å, which are both better than 0.900 and 3.58 Å of AlphaFold2. We present in Fig. 2b the accuracy comparison between DeepAssembly and AlphaFold2 on each target. DeepAssembly achieves a higher TM-score than AlphaFold2 on 66% of the test cases, and a lower RMSD on 67% cases (Fig. 2b). Especially, DeepAssembly has succeeded in building accurate multi-domain protein models with TM-score > 0.9 for 81% targets. From Fig. 2c, we can also observe that DeepAssembly constructs more low-RMSD models, especially those with RMSD of <0.5 Å. It demonstrated the reliability of multi-domain protein models predicted by DeepAssembly.

We then investigate the factors that contribute to the performance of DeepAssembly. Here, we test the control version of DeepAssembly that uses AlphaFold2-predicted domains as the single-domain input, denoted as "DeepAssembly (AF2 domain)".

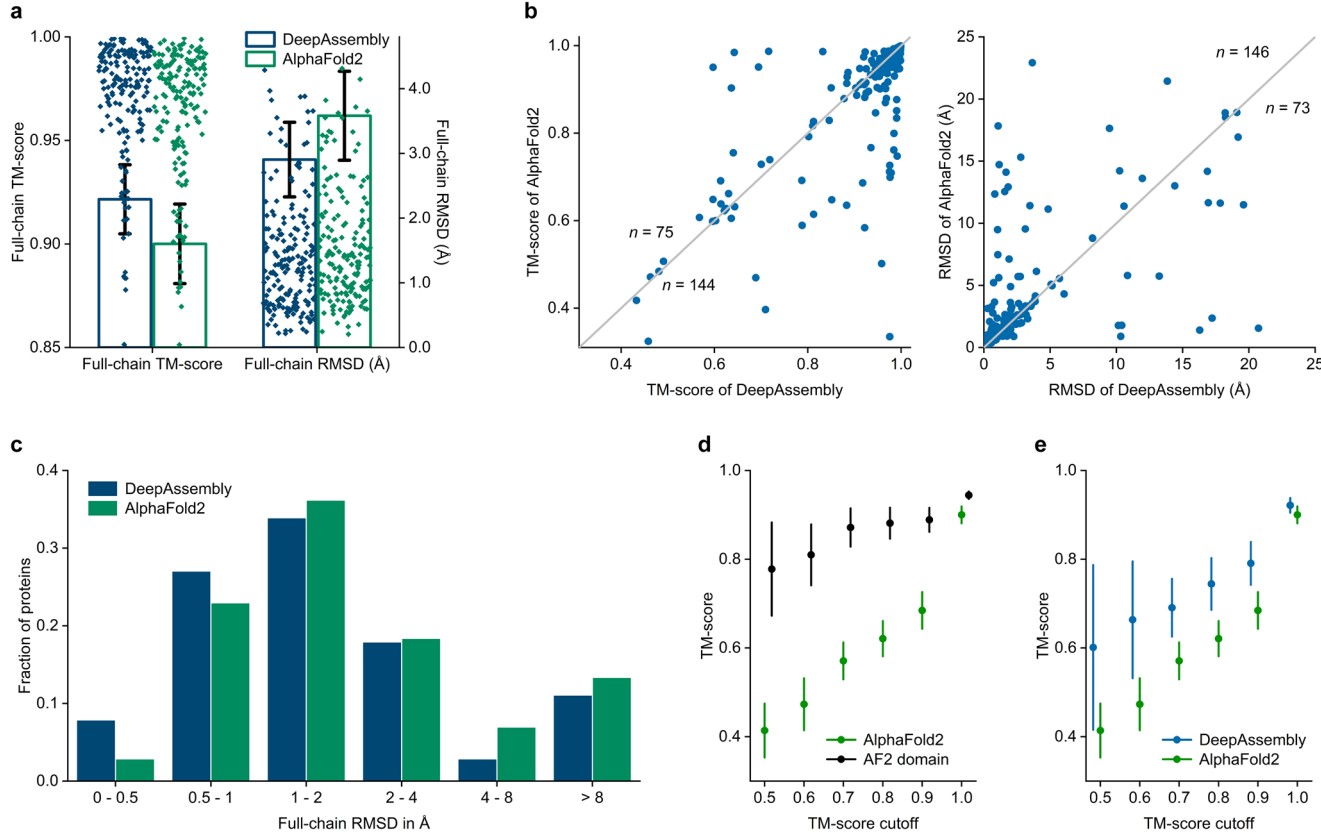

**Fig. 2 Results of assembling predicted domain structures. a** Average TM-score over 219 multi-domain proteins for DeepAssembly and AlphaFold2. The y-axis on the right represents the full-chain RMSD of proteins. Error bars are 95% confidence intervals. **b** Head-to-head comparison of the TM-scores (left panel) and RMSDs (right panel) on each test case between DeepAssembly and AlphaFold2. "n" refers to the number of points on either side of the diagonal. **c** Histogram of backbone RMSD for full-chain models by DeepAssembly and AlphaFold2. **d** TM-score comparisons between AlphaFold2 full-length model and its single domain parts (AF2 domain) on the corresponding proteins of the models predicted by AlphaFold2 with TM-score less than each cutoff. **e** TM-score comparisons between DeepAssembly and AlphaFold2 on the corresponding proteins of the models predicted by AlphaFold2 with TM-score less than each cutoff. "cutoff" refers to a TM-score value on the x-axis, which reflects a TM-score interval less than this value.

Even if the domains are replaced by AlphaFold2 predictions, DeepAssembly still shows improvement compared with Alpha-Fold2 (Supplementary Table 1). The results confirm that the factor that contributes to DeepAssembly performance may be the predicted inter-domain interactions applied to the domain assembly module. This is potentially more important and more challenging. For the AlphaFold2 model with lower prediction accuracy, the gap in the TM-scores between the full-length model and its single domain parts is even larger (Fig. 2d). This also indicates that the main bottleneck for AlphaFold2 structure prediction on multi-domain proteins is capturing the correct inter-domain orientations, followed by modeling quality on single domain parts.

We further analyze the performance of DeepAssembly with single-domains generated by PAthreader and AlphaFold2 in the case of MSA with different number of sequences. Supplementary Fig. 1 is the scatterplot of the TM-score of models predicted by DeepAssembly and DeepAssembly (AF2 domain) versus the number of effective sequences (Neff) in MSAs. The fitting curves in Supplementary Fig. 1 show that the TM-score increases as Neff increases until it largely saturates when Neff is higher than 100. In addition, as Neff increases, the curves representing DeepAssembly and DeepAssembly (AF2 domain) gradually tend to coincide. This indicates that when the MSA is shallow, the template information can greatly improve the performance, while when the MSA is deep enough, the co-evolutionary information plays a major role.

DeepAssembly models improve more obviously on AlphaFold2 models with lower TM-score than on the higher TM-score models (Fig. 2e). Especially for the proteins corresponding to AlphaFold2 models with TM-score < 0.5, the average TM-score of DeepAssembly models is increased by 45.2% compared with AlphaFold2 (with an average increase from 0.414 to 0.601). Similar conclusion can also be drawn from the percentage of the improved proteins by DeepAssembly and the average TM-score improvement rate for the improved proteins in Supplementary Fig. 2a,b. With the lower the accuracy of models predicted by AlphaFold2, the higher TM-score improvement rate of DeepAssembly for these proteins. Specifically, DeepAssembly improves 71.4% of targets predicted by AlphaFold2 with a TM-score less than 0.5, and achieves an average TM-score improvement rate of 72.4% on these improved proteins. The results reflect the potential of DeepAssembly to improve the accuracy of multi-domain proteins that inter-domain orientations are difficult to capture correctly.

Figure 3 shows an example of ERdj5 (PDB ID: 3APO), its J-domain and six tandem thioredoxin (Trx) domains are contained in a single plane but divided into the N- and C-terminal clusters (Fig. 3a). It is functionally meaningful that the redox-active sites in both Trx3 and Trx4 are exposed on the J-domain side of the Trx-containing plain[35] (Fig. 3a, right). Therefore, the correct inter-domain orientation between N- and C-terminal clusters is a key structural feature in the acceleration of ERAD by ERdj5[35]. We built the full-length structure of ERdj5

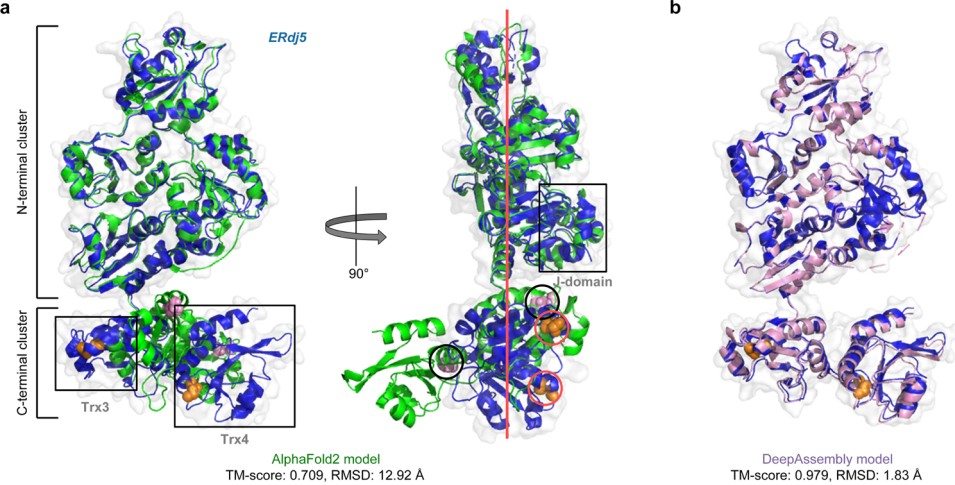

**Fig. 3 Structure modeling results on a PDI family member protein, ERdj5 (PDB ID: 3APO).** Predicted models by AlphaFold2 (green) (**a**) and DeepAssembly (pink) (**b**) are shown along with the experimental structure (blue cartoon and surface). The side chains of the redox-active sites in Trx3 and Trx4 of the experimental structure are shown in orange spheres (circled in red). The side chains of the redox-active sites in Trx3 and Trx4 of the AlphaFold2 model are shown in pink spheres (circled in black). The Trx-containing plain is shown in red line.

by AlphaFold2. We found that the C-terminal cluster composed of Trx3 and Trx4, and the N-terminal cluster composed of the other five domains achieve TM-scores of 0.976 and 0.951, respectively. Nevertheless, AlphaFold2 predicts an incorrect orientation for the C-terminal cluster in comparison to the experimental structure, which results in a TM-score of 0.709 for the full-length model (Fig. 3a). More importantly, in the AlphaFold2 model, the redox-active sites in Trx3 and Trx4 are located respectively on both sides of the Trx-containing plain, which may make it difficult to perform normal functions. Given domains enhanced by PAthreader, we reassemble the full-length model through DeepAssembly. It can be seen from Fig. 3b that DeepAssembly captures the correct orientation between the N- and C-terminal clusters, and the model achieves a TM-score of 0.979, which increases by 38.1% compared with AlphaFold2.

**DeepAssembly corrects the domain orientation of structures in AlphaFold database.** The European Bioinformatics Institute (EBI) AlphaFold database is a very nice resource to have. However, some issues are likely to emerge in the future[36]. We need to know if and how to update the AlphaFold database. DeepAssembly may provide a lightweight approach to update the low confidence multi-domain structures in AlphaFold database.

To analyze DeepAssembly's potential to improve the quality of multi-domain protein models with lower AlphaFold2 prediction accuracy, we create a set of 164 multi-domain proteins from *H. sapiens* proteome in AlphaFold database with the TM-scores < 0.8 for the corresponding AlphaFold2 structures and the experimental structures in PDB contain more than 85% of solved residues (see Methods). We predict these multi-domain proteins using DeepAssembly and compare them with structures deposited in AlphaFold database (Supplementary Data 2). We find that DeepAssembly has outperformed AlphaFold2 on the test cases (Fig. 4a). On average, the models predicted by DeepAssembly achieve a TM-score of 0.690, which is 13.1% higher than that (0.610) of AlphaFold2 structures. The reassembly of these multi-domain proteins by DeepAssembly considerably improves the accuracy (Fig. 4b). DeepAssembly successfully constructs more models (47% of the cases) with TM-score >0.7 compared to AlphaFold2 (Fig. 4a), and there are 27% of prediction models that TM-score is improved to over 0.8 by DeepAssembly. DeepAssembly also achieves a better performance than AlphaFold2 on

66% of cases (Fig. 4c). These results demonstrate that structures with low confidence deposited in AlphaFold database can be improved by DeepAssembly by correcting their inter-domain orientations with large errors.

A prominent example is the soluble epoxide hydrolase (sEH) (PDB ID: 3WK4_A), a candidate target for therapies for hypertension or inflammation[37]. The structure prediction of AlphaFold2 for the sEH has errors in the inter-domain orientation with a full-chain TM-score of 0.689 and RMSD of 6.13 Å (Fig. 4d). In addition, we also assess the inter-domain accuracy of AlphaFold2 structure by the predicted aligned error (PAE) plot, which captures the model's global inter-domain structural errors[9,28]. The PAE plot for the AlphaFold2 structure is shown in Fig. 4e. The two low-error squares correspond to the two domains. For residues within domains, the PAE is low, with TM-scores of 0.990 and 0.995 for the two single domains. In contrast, the orientation of the two domains has a high PAE. The high PAE across the whole inter-domain region indicates that in this case AlphaFold2 does not predict relative domain orientation. For the DeepAssembly model, which shows close similarity to the experimental structure with a TM-score of 0.996 and RMSD of 0.54 Å (Fig. 4f). The low PAE across the whole region reflects that DeepAssembly not only accurately models the single domains, but also captures the correct inter-domain orientation between them (Fig. 4g).

Another example is human complement factor B (PDB ID: 2OK5_A), the central protease of the complement system of immune defense, consisting of five domains[38]. In the AlphaFold2 model, almost all domains are predicted accurately (average TM-score is 0.961), however, there is an obvious difference in orientation between the serine protease (SP) domain and the other domains relative to the experimental structure (Fig. 4h). As visible in Fig. 4i, there is a high PAE across the inter-domain region between SP domain and the other four domains, although the four domains have a low PAE in orientation to each other. The difference is that the full-chain model predicted by DeepAssembly is closely consistent with the experimental structure (Fig. 4j). The DeepAssembly model gives low PAE in the entire model including all domains and their inter-domain regions (Fig. 4k). These results again demonstrate the ability of DeepAssembly at the levels of multi-domain protein assembly, especially in the aspect of inter-domain orientation prediction.

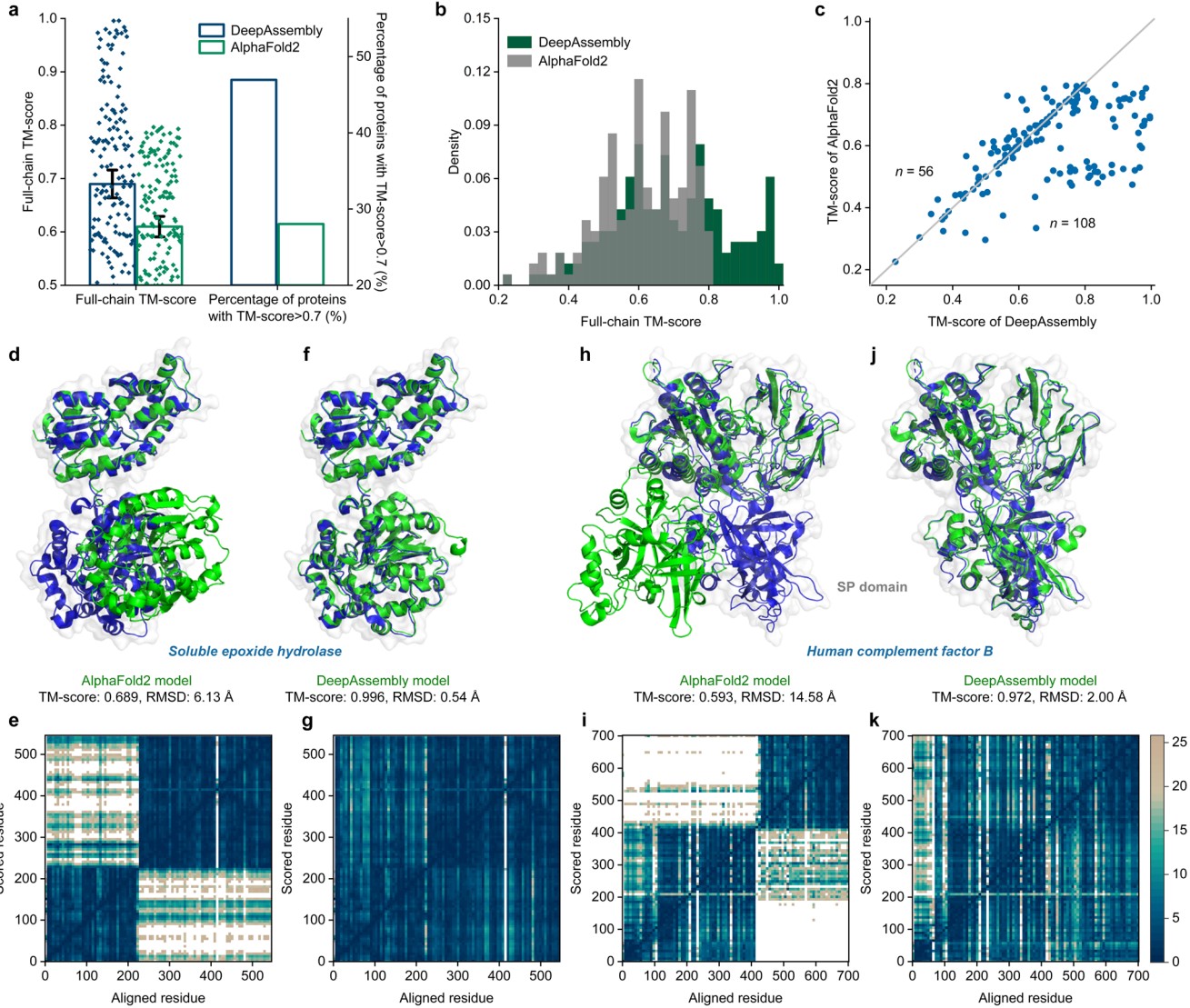

**Fig. 4 DeepAssembly corrects the inter-domain orientation of multi-domain protein structures in AlphaFold database. a** Average TM-score over 164 multi-domain proteins for DeepAssembly and AlphaFold2. The y-axis on the right represents the percentage of targets with TM-score>0.7. Error bars are 95% confidence intervals. **b** Distribution of the full-chain TM-score for the predicted models. **c** Head-to-head comparison of the TM-score value on each test case between DeepAssembly and AlphaFold2. "n" refers to the number of points on either side of the diagonal. Predicted models of AlphaFold2 (**d**, **h**) and DeepAssembly (**f**, **j**) for soluble epoxide hydrolase (PDB ID: 3WK4_A) and human complement factor B (PDB ID: 2OK5_A). Experimentally determined structures are colored in blue, and the predicted models are colored in green. PAE plots for the predicted models by AlphaFold2 (**e**, **i**) and DeepAssembly (**g**, **k**). The color at (x, y) corresponds to the expected distance error in residue y's position, when the prediction and true structure are aligned on residue x.

Although it is impressive that AlphaFold2 accurately predicts protein structure, there are still quite a few structures in the AlphaFold database with lower confidence in the alignment of certain domains. This highlights the necessity for AlphaFold database update. Importantly, the possibility to correct inter-domain orientation by DeepAssembly can help to improve the quality of multi-domain protein structures with low confidence, providing an efficient way for updating of the AlphaFold database.

**Performance on the CASP14 and 15 targets**. We apply DeepAssembly to CASP14 and CASP15 targets to further evaluate its performance. We selected a total of 30 multi-domain targets from CASP14 and 15 as test proteins according to the "Domain Definition" on the CASP official website, of which 17 are from CASP14 and 13 are from CASP15. These targets all have known

PDB structures that been released. We assembled full-length models by DeepAssembly on these CASP targets with single-domains generated by PAthreader, and compared with Alpha-Fold2 (Supplementary Data 3).

Supplementary Table 2 summarizes the results of targets from CASP14 and CASP15 predicted by AlphaFold2 and DeepAssembly. On the CASP14 targets, the average TM-score of DeepAssembly is 0.850, which is higher than 0.832 for AlphaFold2. DeepAssembly also achieved a better performance than Alpha-Fold2 on most of the test targets (Supplementary Fig. 3a). Especially for the target T1024, a membrane protein that consists of two domains, DeepAssembly correctly predicted its inter-domain orientation with a full-length TM-score of 0.957, which is an improvement over 0.797 by AlphaFold2 (Supplementary Fig. 4a). For the targets from CASP15, the models predicted by DeepAssembly achieved an average TM-score of 0.584, which is

slightly higher than 0.567 for AlphaFold2. Specifically, DeepAssembly obtained models with a higher TM-score than AlphaFold2 on 62% of the targets (Supplementary Fig. 3b). Supplementary Fig. 4b shows an example of target T1121. AlphaFold2 accurately predicted its two individual domains (T1121-D1 and T1121-D2) with TM-scores of 0.944 and 0.953, respectively. However, the full-length models directly predicted by AlphaFold2 obtained a lower TM-score (0.743). In this case, DeepAssembly generated a highly accurate model with a TM-score of 0.925, demonstrating its ability to improve the accuracy of multi-domain protein structure prediction by capturing the correct inter-domain orientation.

Furthermore, in order to analyze the effect of the single-domain structures used on the final full-length model accuracy, we further assembled the model through DeepAssembly with the experimental domain structures. The results show that the average TM-score on the CASP14 and CASP15 targets by using the experimental domain structures achieved 0.903 and 0.778, respectively. The example, T1137s1, is a challenging target as its second domain (T1137s1-D2) is relatively difficult to predict. As illustrated in Supplementary Fig. 4c, the TM-score of AlphaFold2 model is 0.446. In this target, the two predicted single-domain structures used by DeepAssembly have TM-scores of 0.870 and 0.503, respectively, where the lower accuracy of the second domain affects the quality of full-length model, resulting in a TM-score of 0.474. However, by using experimental single-domain structures, DeepAssembly generated a highly accurate model with a TM-score of 0.949 (Supplementary Fig. 4c). This indicates that predicted accuracy of the assembled full-length model could be further improved if the high-quality single-domain models are obtained. Meanwhile, the quality of single-domain model is also an important factor in the accurate multi-domain protein structure prediction in addition to inter-domain orientation.

**Applying DeepAssembly to protein complex structure prediction.** It is well known that domains, as independent folding and functional units, reappear across species in protein structure. For some specific inter-domain interactions, they are not only present in monomeric proteins, but also implicit in protein complexes. We find an example of the Cdc42/Cdc42GAP/ALF3 complex (PDB ID: 1GRN), consisting of two protein chains (Supplementary Fig. 5a), to support our hypothesis. There is a fusion protein linked by RhoA and the GAP domain of MgcRacGAP (PDB ID: 5C2K) (Supplementary Fig. 5b). Interestingly, the crystal structure of the fusion protein is very close to that of complex 1GRN (TM-score = 0.93), although its sequence similarity to complex 1GRN is less than 40%. This fusion protein may be derived from the fusion of two unique protein chains. The original chains form the domains of the fusion protein, but the relative orientation between domains remains largely unchanged. It may be that specific interactions between the two chains of the complex have been preserved during evolution, emerging as a relatively stable inter-domain pattern in other protein structures. Inspired by this, we further conclude that there is essentially no difference between the prediction of protein complexes and multi-domain proteins, and the inter-domain interactions learned from monomeric multi-domain proteins could be applied to protein complex structure prediction through domain-level assembly.

Based on the above conclusion, we test the potential of DeepAssembly for assembling complexes of known structures over a test set of 247 heterodimeric protein complexes. Here, the structure of complex is constructed using DeepAssembly, in which each sequence is split into single-domain sequences to construct the inter-domain paired MSAs, and all domains in each chain are treated as assembly units that are predicted separately by single-domain structure predictor. It should be noted that the inter-domain interactions used here are from the same deep learning model trained by monomeric proteins as when assembling multi-domain proteins. In addition, the built models are evaluated using DockQ score[39]. DockQ measures the interface quality (Supplementary Note 1), interfaces with score >0.23 are considered correct. The success rate (SR) represents the percentage of cases whose DockQ are greater than 0.23.

In the test set, DeepAssembly successfully predicted the interface (DockQ $\geq$ 0.23) in 32.4% of cases (Supplementary Data 4). We compared its performance with RoseTTAFold and AlphaFold-linker (Supplementary Table 3). The RoseTTAFold models are obtained by running its end-to-end version, and AlphaFold-linker models are obtained by adding a 21-residue repeated Glycine-Glycine-Serine linker between each chain and then running it as a single chain through the AlphaFold2[27]. Overall, DeepAssembly has an improvement of 74.2% compared to RoseTTAFold (SR = 18.6%), is better than RoseTTAFold for 66% cases, and generates more high-quality models (Fig. 5a). Meanwhile, our method can almost achieve the performance of AlphaFold-linker (SR = 40.9%) with less computational resource requirements. The number of medium models generated by DeepAssembly is close to AlphaFold-linker (Fig. 5a), especially it is superior to AlphaFold-linker on 36% cases. This validated the effectiveness of our method that is based on domain assembly and demonstrated that the deep learning model built by learning the inter-domain interactions in the existing PDB monomer structures can capture the protein-protein interactions as well. In addition, we tested the performance of AlphaFold-Multimer (version 2.1.1) with the default settings. It is currently the state-of-art method for multimer structure prediction, achieving an SR of 65.2% on the test set. However, this method was trained on the entire PDB, and the redundancy between the test set and its training set is not removed, which makes a direct comparison difficult[26,40]. Nevertheless, DeepAssembly has higher interface accuracy than AlphaFold-Multimer on 21% cases. Figure 5b shows an example of a heterodimer composed of chains E and G of the viral RNA polymerase (PDB ID: 4Q7J). We successfully predict the interface (DockQ = 0.64), showing a low inter-chain predicted error in the PAE map. For the model built by AlphaFold-Multimer, the individual chains are predicted to be correct, while their relative position is wrong (DockQ = 0.003), as can be seen from the blocks with a high predicted error in the PAE map. This shows that DeepAssembly is complementary with AlphaFold-Multimer. On the other hand, there are three of the complexes could not be successfully modeled by AlphaFold-Multimer due to the GPU memory limitations. DeepAssembly may provide a more lightweight way to assemble complex structures by treating domains as assembly units.

We continue to analyze the performance for different subsets of the test set. We first investigated the effect of the number of effective sequences (Neff) in paired MSAs on the outcome. It is obvious that MSA Neff has a large impact on the performance, the proportion of proteins that are correctly predicted to interface increases with deeper MSA (Fig. 5c). Too shallow MSAs do not provide sufficient co-evolutionary signals, while too deep alignments might contain false positives resulting in noise masking the sought after co-evolutionary signal[26]. It can be seen from Fig. 5c that the performance gradually tends to saturation with larger Neff scores; even, for the subset with Neff > 512, the fraction of models with DockQ score > 0.6 is reduced compared to the subset with fewer Neff. We then divided the proteins by kingdom (Fig. 5d). The SRs for each kingdom is: 28.8% for eukaryotes, 42.5% for bacteria, 50.0% for archaea, 33.3% for virus, and 10.5% for mixed kingdoms (e.g., one viral protein interacting with one eukaryotic). The higher performance on bacteria and

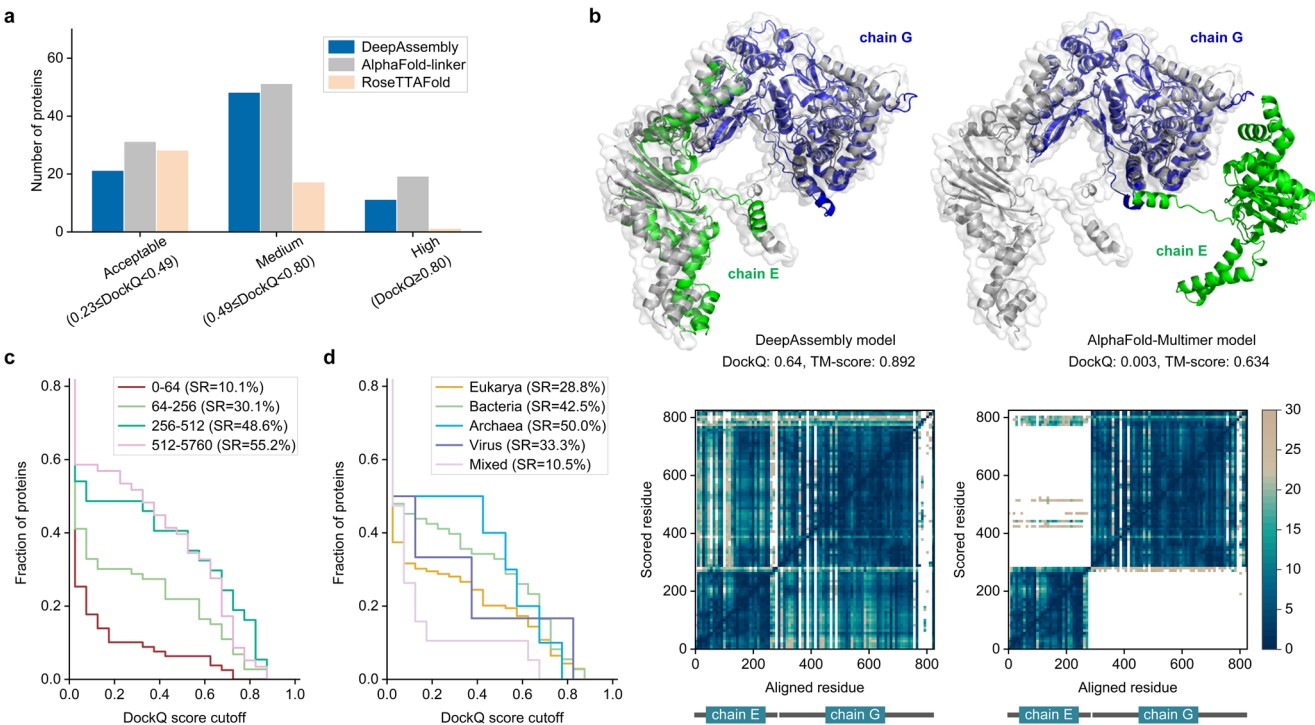

**Fig. 5 Results of protein complex prediction. a** The number of models by DeepAssembly, AlphaFold-linker and RoseTTAFold within each DockQ score threshold. **b** Example of a heterodimer of viral RNA polymerase (PDB ID: 4Q7J) predicted by DeepAssembly and AlphaFold-Multimer. Experimentally determined structure is colored in gray, and the predicted model is colored by chain. Below are PAE heat maps showing the predicted error. **c** Fraction of proteins with DockQ score higher than each cutoff for different paired MSA depths. **d** Fraction of proteins with DockQ score higher than each cutoff for different kingdoms.

archaea is consistent with the observation that more evolutionary information is available for them than others (Supplementary Fig. 6). It is relatively difficult to predict protein interactions in eukaryotes due to the smaller number of sequenced genomes and higher numbers of paralogous genes compared to prokaryotes[41]. Similarly, the lower SRs of mixed kingdoms is due to the lack of inter-chain co-evolutionary signals, since paired MSAs are generated by hits pairing of the interacting chains from the same organism[26,42].

Further, through several cases of predicted outcomes, we examine the complementarity between co-evolutionary information and structural templates. Figure 6a, b display two cases (3F3F_B-D and 3MML_A-B) with sufficient co-evolutionary signals, the DockQ scores achieve 0.611 and 0.699, respectively, by using remote template information. This suggests that structural templates can jointly promote the performance improvement with co-evolutionary information. In the absence of remote templates, co-evolutionary information is the main driver. Two models built without structural templates are displayed in Fig. 6c (2Y69_A-B) and Fig. 6d (4Q35_A-B), with DockQ scores of 0.790 and 0.685 respectively. However, generating paired MSAs is impractical in many cases because there are few co-evolutionary signals across species, e.g., pathogen-host interactions[16]. An interesting successful assembly is obtained modelling chains A and B from the complex with PDB ID 3ALZ (Fig. 6e), the measles virus hemagglutinin bound to its cellular receptor SLAM. As a complex with chains from mixed kingdoms, we found that its paired MSA depth was almost 0, implying that it is quite difficult to predict the interface using only such insufficient co-evolutionary signals. Even then we correctly predicted its interface with a DockQ score of 0.675, through the interactions learned from the structural template. Similarly, for 4Q7J_E-G (Fig. 5b), in the absence of co-

evolutionary signals, its interface is also successfully predicted (DockQ = 0.64). In the one remaining incorrect model (1JMU_I-F, Fig. 6f), there is not sufficient inter-chain co-evolutionary information and suitable templates, making the assembly difficult and resulting in a DockQ score close to 0. These suggest that the roles of co-evolutionary information and structural templates are complementary. Especially for scenarios without co-evolutionary signals, our method can identify the correct inter-chain interface through the inter-domain interactions learned from the remote templates, it could reduce the dependence on MSA to a certain extent. After all, in nature, the protein folding process is inherently driven by physical force fields, which itself does not consider whether there is a co-evolutionary relationship.

We also applied DeepAssembly to generate models of the heterotrimer Survivin-Borealin-INCENP core complex (PDB ID: 2QFA) and the heterotetramer NuA4 core complex (PDB ID: 5J9T). As illustrated in Supplementary Fig. 7a, DeepAssembly generated a highly-quality model with a DockQ score of 0.828 for three-chain hetero-complex 2QFA. In addition, for the four-chain hetero-complex 5J9T, the complex model built by DeepAssembly also achieved a DockQ score of 0.760 (Supplementary Fig. 7b), showing the potential that DeepAssembly could also be applied to hetero-complexes with more than two chains.

**Application to transmembrane protein structure prediction.** Membrane proteins are involved in a variety of essential cellular functions including molecular transporters, ion channels and signal receptors, which constitute about half of current drug targets[43]. Therefore, the structural determination of membrane proteins is critical to advancing our understanding of their function as well as the drug design. Here, we apply DeepAssembly to transmembrane protein structure prediction.

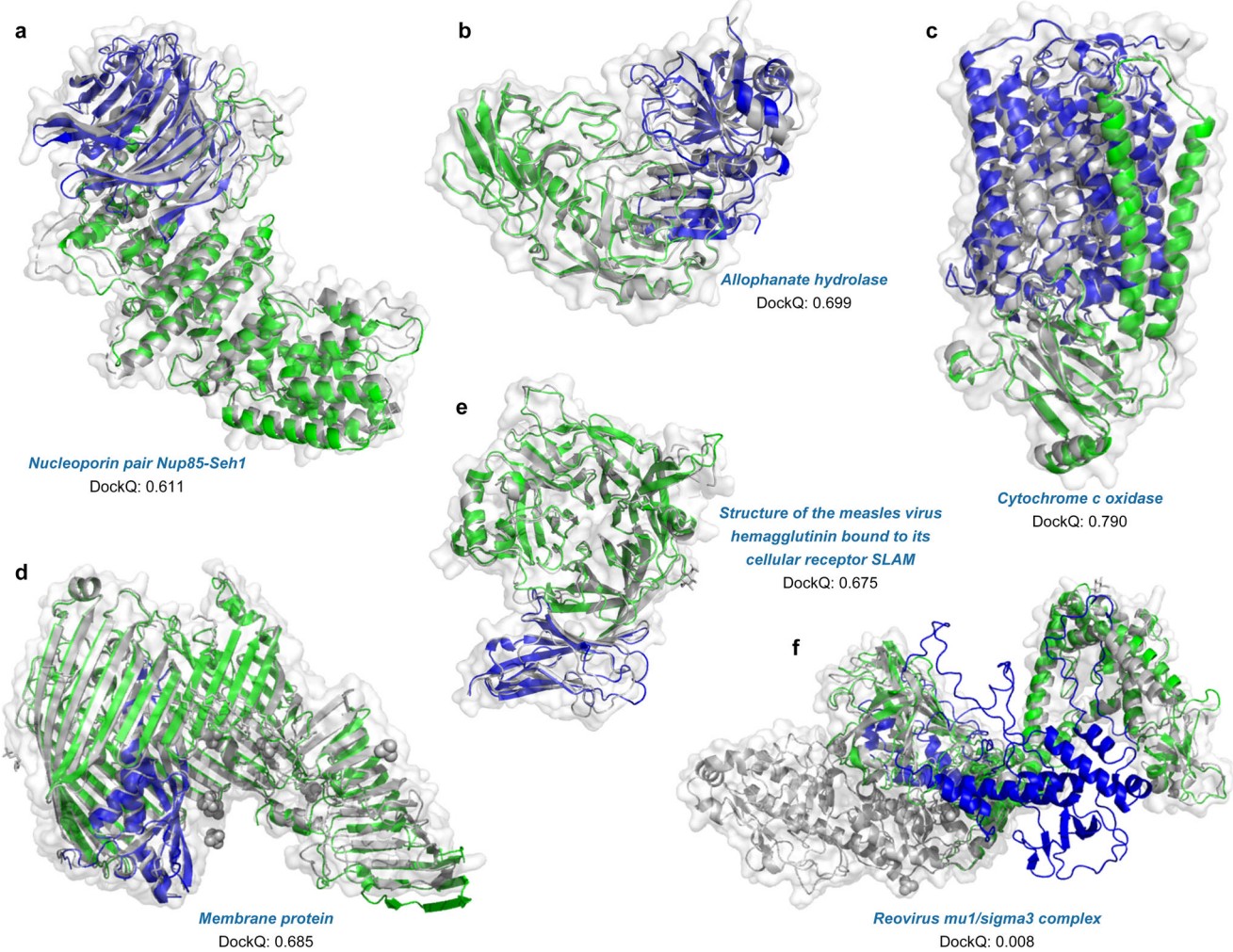

**Fig. 6 Examples of the protein complex structures built by DeepAssembly. a** Nucleoporin pair Nup85-Seh1 (PDB ID: 3F3F) (DockQ = 0.611).
**b** Allophanate hydrolase (PDB ID: 3MML) (DockQ = 0.699). **c** Cytochrome c oxidase (PDB ID: 2Y69) (DockQ = 0.790). **d** Membrane protein (PDB ID:
4Q35) (DockQ = 0.685). **e** Structure of the measles virus hemagglutinin bound to its cellular receptor SLAM (PDB ID: 3ALZ) (DockQ = 0.675). **f** Reovirus
mu1/sigma3 complex (PDB ID: 1JMU) (DockQ = 0.008). Experimentally determined structure is colored in gray, and the predicted model is colored
by chain.

Figure 7 displays several examples of the transmembrane protein models predicted by DeepAssembly. The first example is Monocarboxylate transporter 1 (MCT1) (PDB ID: 7CKR_A), which is a multi-domain protein that catalyzes the movement of many monocarboxylates across the plasma membrane through an ordered mechanism[44]. The model built by DeepAssembly had the correct inter-domain orientation, yielded a TM-score of 0.882 (Fig. 7a). In addition, the target T1024 in CASP14 (PDB ID: 6T1Z_A) is also a transmembrane protein, which consists of two domains. In this case, DeepAssembly generated a more accurate model with a TM-score of 0.957, compared to AlphaFold2's 0.797 (Supplementary Fig. 4a). Figure 7b, c show the results of DeepAssembly on two homo-oligomeric proteins with C2-symmetry type. The models predicted by DeepAssembly achieved DockQ scores of 0.881 (Fig. 7b) and 0.796 (Fig. 7c), respectively. Furthermore, for OmpU, an outer membrane protein of Vibrio cholerae[45] (PDB ID: 6EHB), DeepAssembly correctly predicted the structures of its three channels (average TM-score = 0.985), as well as their interfaces (DockQ = 0.591) (Fig. 7d). Figure 7e displays an example of tetrameric potassium channel KcsA (PDB ID: 2QTO), which is an ion channel that is capable of selecting potassium ions. The tetramer model for KcsA built by DeepAssembly

achieved a DockQ score of 0.756, and the model gave the correct overall shape of the pore used to filter out potassium ions (shown as sphere in Fig. 7e). These examples demonstrate the potential of DeepAssembly on transmembrane proteins, which may help to understand the function of transmembrane proteins, and provide insights into their molecular interaction mechanism.

However, there is still room for further improvement. Since our deep learning model is only trained on the data set of mainly soluble multi-domain proteins, which are different from the structural characteristics of membrane proteins, our method still has some limitations for the assembly of some membrane proteins that are large or flexible across transmembrane regions. For example, for the target of acid-sensing ion channel (PDB ID: 2QTS) with three chains, there is a deviation between experimental structure and predicted model (DockQ = 0.402), especially in the transmembrane region (Fig. 7f). This may be due to the highly flexible structures within the transmembrane region, with each subunit in the PDB structure of the chalice-shaped homotrimer having a different orientation between its transmembrane helices and extracellular region, even though they share the same amino acid sequence.

Moreover, the number of membrane proteins is rather limited compared with a great deal of soluble protein structures resolved

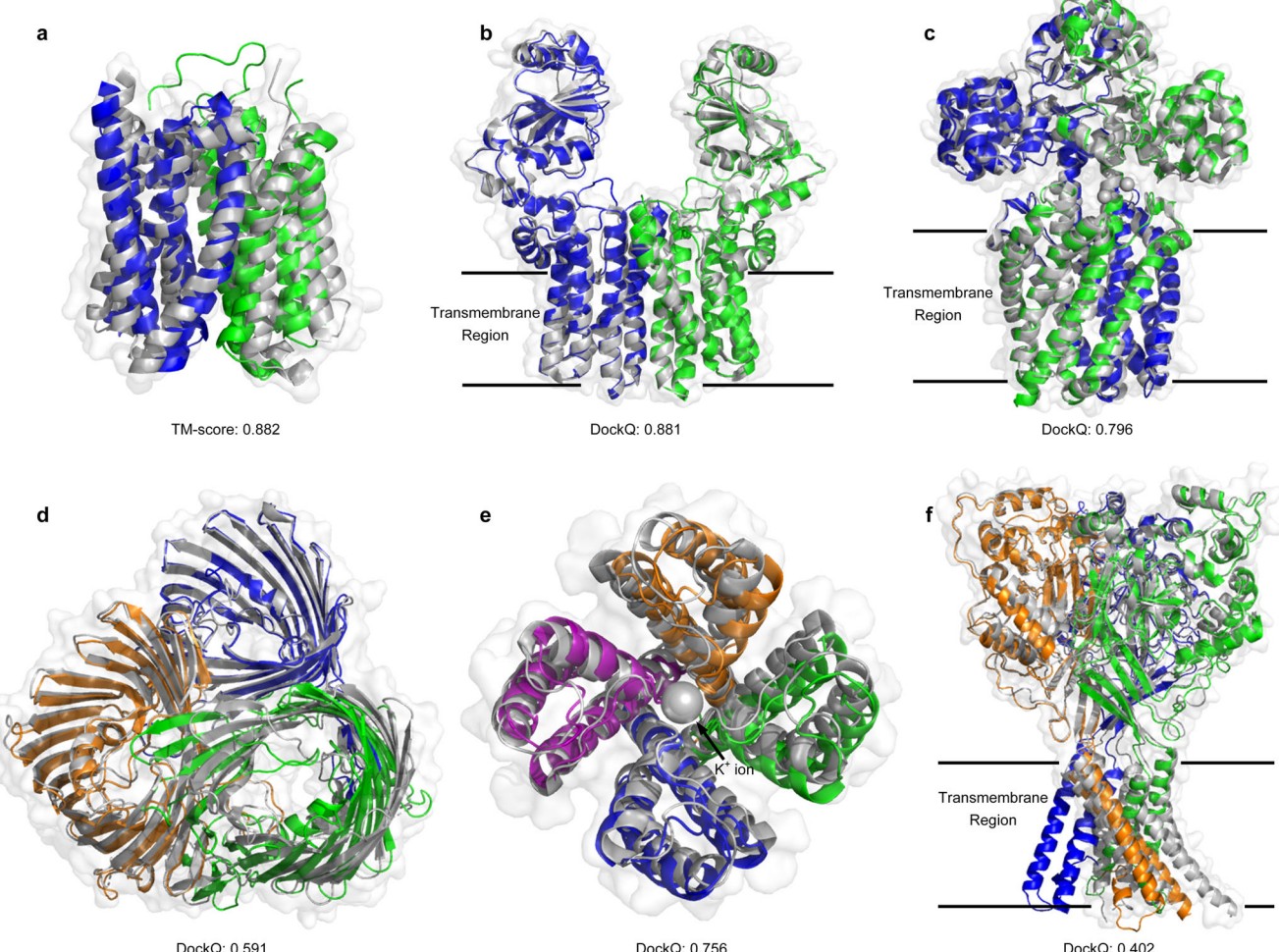

**Fig. 7 Examples of the transmembrane protein structures predicted by DeepAssembly.** The reference PDB structures are colored in gray, and the different domains or chains of the predicted models are colored by blue, green, orange, and purple. **a** Monocarboxylate transporter 1 (MCT1) (PDB ID: 7CKR_A). **b** CDP-alcohol phosphotransferase (PDB ID: 4O6M). **c** magnesium transporter (PDB ID: 2ZY9). **d** OmpU, an outer membrane protein of Vibrio cholerae (PDB ID: 6EHB). **e** potassium channel KcsA (PDB ID: 2QTO). The potassium ion in the ion channel is shown as sphere. **f** acid-sensing ion channel (PDB ID: 2QTS).

in PDB, as their experimental determination is subject to the complicated membrane environment. This poses a major obstacle to training deep learning model directly on such a small dataset of membrane proteins[43]. To address the challenge, some statistical and transfer learning-based methods may be promising, such as DeepTMP[43], which predicts the transmembrane protein inter-chain contact by transferring the knowledge learned from the initial model trained on a large dataset of soluble protein complexes. This approach alleviates the limitation of training deep learning model directly on few membrane proteins, and simultaneously provides insights into how we further improve the complex structure prediction by leveraging multi-domain protein data.

**Assembling multi-domain proteins with experimental domains.** In order to rigorously test the model performance of DeepAssembly, we assemble multi-domain proteins with experimental domains to exclude the influence of single-domains on the final structure. In this experiment, DeepAssembly is compared with SADA, DEMO and AIDA (Supplementary Data 1), and for a fair comparison, we here exclude templates in DeepAssembly with a sequence identity >30%, as done in SADA and DEMO. Figure 8 presents a summary of the models

assembled by DeepAssembly, SADA, DEMO and AIDA. The detailed data are listed in Supplementary Table 4. It can be seen from Fig. 8a that DeepAssembly outperformed the other methods. The models built by DeepAssembly achieve an average TM-score of 0.856, which is higher than 0.763 for SADA, 0.702 for DEMO, and 0.589 for AIDA. Figure 8b provides a comparison between DeepAssembly and other methods, which suggests that DeepAssembly generates the highest number of models with each TM-score cutoff. In particular, DeepAssembly builds models with TM-score > 0.9 for 64% out of the 219 targets. In Fig. 8c, we list a head-to-head TM-score comparison of DeepAssembly with other methods, which shows that DeepAssembly has 70%, 78% and 87% of models with higher TM-score than SADA, DEMO and AIDA, respectively. We can imagine that, under the condition that single-domain structure is relatively easy to be resolved, our method with the way of domain assembly could help experimental scientists accelerate the analysis of the structure of multi-domain proteins or protein complexes, and help to further promote the progress of experimental techniques.

Figure 8d, e show two representative examples of multi-domain proteins assembled by DeepAssembly using experimental domain structures. The first example is the 30 S ribosomal protein S4e from Thermoplasma acidophilum (PDB ID: 3KBG_A), which

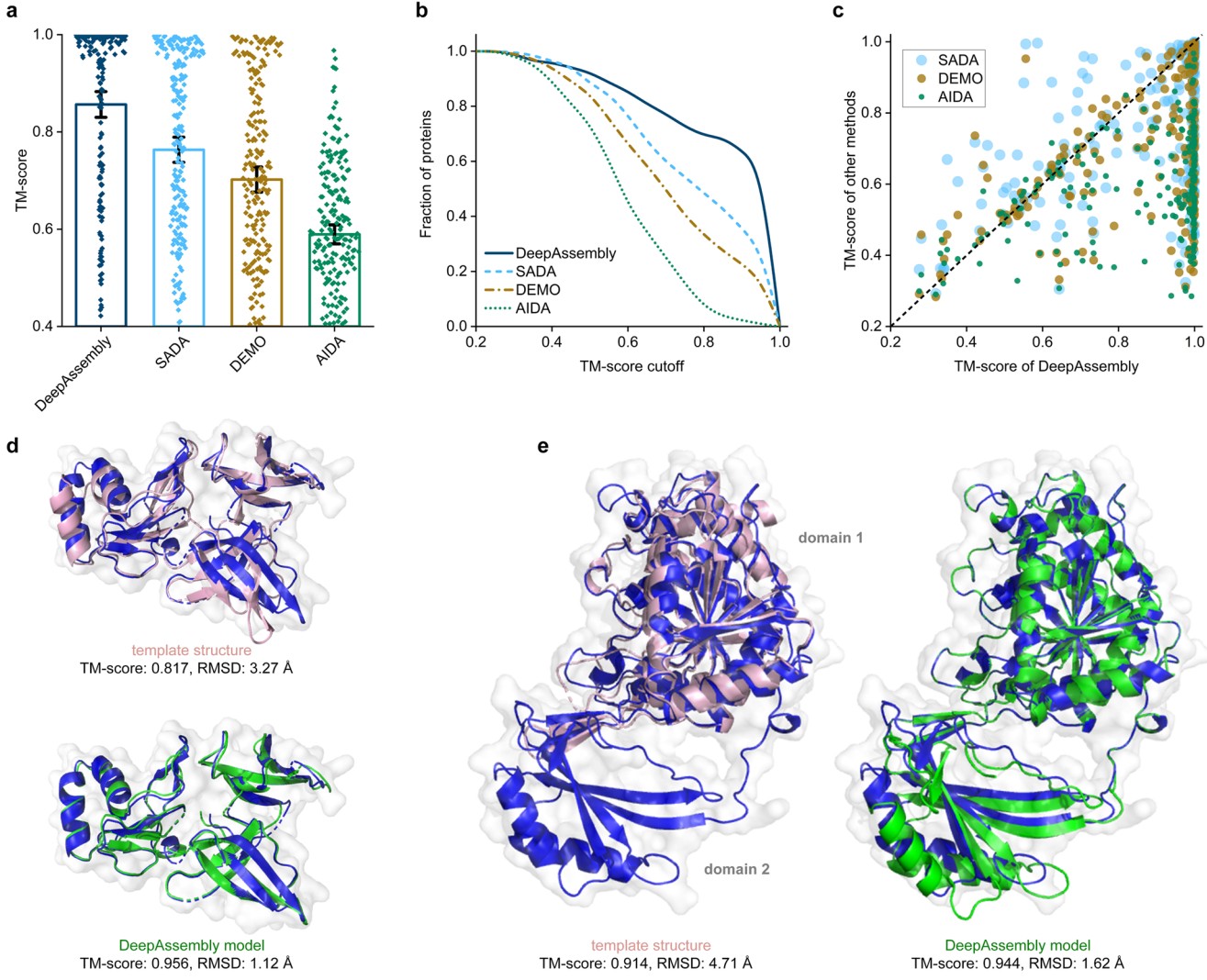

**Fig. 8 Results of assembling experimental domain structures. a** Average TM-scores of the assembly models by DeepAssembly, SADA, DEMO and AIDA ($n = 219$ proteins). Error bars are 95% confidence intervals. **b** Comparison between DeepAssembly, SADA, DEMO and AIDA on 219 multi-domain proteins based on the fraction of proteins with TM-score higher than each TM-score cutoff. **c** Head-to-head comparison of the TM-scores on each test case between DeepAssembly and other methods. Examples of the assembly structures built by DeepAssembly for 3KBG_A (**d**) and 2D1C_A (**e**). Experimental structures are colored in blue, the predicted models are colored in green, and the remote templates searched by PAthreader are colored in pink.

contains three continuous domains. It can be seen from the Fig. 8d that the template detected by PAthreader achieves a TM-score of 0.817, even though its sequence similarity is only 30%, compared to the TM-score of 0.589 for template obtained by HHsearch[46] (see Supplementary Fig. 8a). The model predicted by DeepAssembly using the PAthreader templates achieves the best quality among different methods with a TM-score of 0.956. Another example of multi-domain protein is TT0538 protein from Thermus thermophilus HB8 (PDB ID: 2D1C_A) comprising two domains. Different from the previous example, for this case, we only identify the template of one of its domains (domain 1) through PAthreader, and the structure of the other domain (domain 2) is missing in the template, even if the template achieves a TM-score of 0.914 (Fig. 8e). Similarly, the same domain is also missing from the template found using HHsearch (Supplementary Fig. 8b). Disappointingly, these templates contain information within only one of the domains, which is a limited contribution to capturing interactions between the domains. Nevertheless, DeepAssembly successfully assembles the full-length model with a TM-score of 0.944. Overall, as in the above two case studies, it is shown that the remote template

from our PAthreader plays a critical role in the model quality improvement, however, the performance of DeepAssembly is not entirely template dependent. When the template provided is of poor quality, DeepAssembly can still capture the correct inter-domain interactions.

**Inter-domain interactions prediction.** A key component of DeepAssembly is the deep neural network-based inter-domain interactions prediction, which is used to guide the domain assembly procedure. Our predicted inter-domain interactions are represented by affine transformations between inter-domain residues, which contain rotations and translations between domains. Here, we devise an affine transformation-based potential called atomic coordinate deviation (ACD), which is minimized in domain assembly simulations to make the domain relative positions of the conformation gradually satisfy the predicted inter-domain interactions (see details in Methods section). Figure 9a presents a strong correlation (Pearson $r = 0.80$, $R^2 = 0.63$) between predicted atomic coordinate deviation (pACD) and true atomic coordinate deviation (tACD) on 2,190 models (10 models for each protein) assembled by DeepAssembly

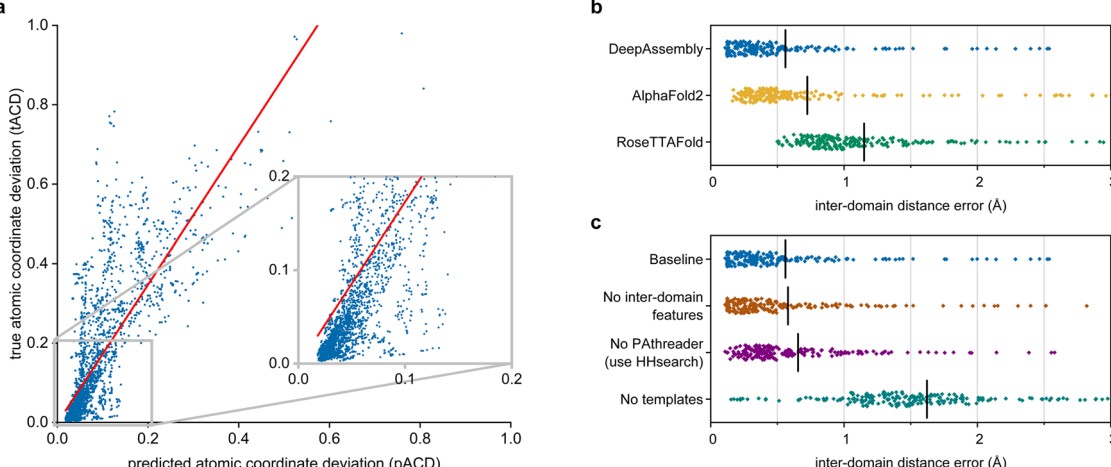

**Fig. 9 Evaluation of predicted inter-domain interactions. a** Correlation between predicted atomic coordinate deviation (pACD) and true atomic coordinate deviation (tACD) (Pearson $r = 0.80$, $R^2 = 0.63$, $n = 2{,}190$). **b** Swarm plots displaying the errors ($n = 219$) of predicted inter-domain distance obtained from different methods. Each point represents one sample with the mean errors marked by a black "|". The averages are 0.560, 0.724 and 1.151 Å for DeepAssembly, AlphaFold2 and RoseTTAFold, respectively. **c** Ablation study of inter-domain distance prediction accuracy. "Baseline" denotes the full information is used. "No inter-domain features" denotes a version of DeepAssembly without inter-domain features. "No PAthreader" means that the PAthreader templates are replaced with those searched by HHsearch. "No templates" represents a model without template features. The averages are 0.560, 0.580, 0.655 and 1.621 Å, respectively.

using experimental domain structures. Where, pACD represents the coordinate deviation under the predicted inter-domain interactions, and tACD is for the true interactions from experimental structure. Furthermore, in Supplementary Fig. 9a, b, we observe a strong correlation between pACD and accuracy for these models as well. This indicates the potential of DeepAssembly to capture the correct domain relative positions in the optimization of pACD based on predicted inter-domain interactions.

Predicted inter-domain interactions include inter-residue distance between domains. We compare the DeepAssembly with AlphaFold2 and RoseTTAFold according to the predicted inter-domain distances (Supplementary Data 1). The inter-domain distance of DeepAssembly is directly from AffineNet, and distances predicted by AlphaFold2 and RoseTTAFold are extracted from their output pkl and npz files, respectively. Here we define the inter-domain distance error to evaluate the predicted distance precision (Supplementary Note 1), which is calculated as the errors (Å) between the predicted and true inter-domain distances. We present in Fig. 9b the comparison of the inter-domain distance errors of DeepAssembly, AlphaFold2 and RoseTTAFold. For all the methods, the input MSAs are the same, which are generated by HHblits[46] searching Uniclust30[47] and BFD[48] databases, and only the distance between inter-domain residues is considered when the error is calculated. On average, the inter-domain distance error of DeepAssembly is 0.560 Å, which is 22.7% less than AlphaFold2 (0.724 Å) and 51.3% less than RoseTTAFold (1.151 Å). In addition, the data list in Supplementary Table 5 indicates that the average and median errors of our method are all less than other methods for different categories of domain count. Especially for targets with more than four domains, the average distance error of DeepAssembly is reduced by ~30% compared with AlphaFold2. The above improvements may come from such aspects as the DeepAssembly network trained specifically by multi-domain proteins from the MPDB, the inter-domain features embedded in the network, and higher quality templates from our remote template recognition algorithm. On the other hand, the prediction about orientation may improve the prediction accuracy of inter-domain distance to some extent, because they may promote each other.

Figure 9c shows an ablation study to estimate the contributions to the improved performance of the different components of our approach (Supplementary Table 6 provides detailed results). We trained a variety of neural network models on the MPDB data using the same procedure over different input features. Under each setting, the test set is the 219 proteins above. When inter-domain features are removed from the baseline network (the full information is used; "Baseline" in Fig. 9c), the average error of inter-domain distance increases from 0.560 Å to 0.580 Å. This difference implies that inter-domain features are instrumental in capturing the correct inter-domain interactions, perhaps because they encourage the network to be more concerned with learning the covariation relationships between domains. A comparison of "Baseline" and "No PAthreader" suggests that the network using PAthreader templates may decrease the error over using HHsearch templates by ~14.5% (from 0.655 Å to 0.560 Å). The improvement is mainly due to the higher quality templates provided by PAthreader (see Methods). When template features are not used, the model performance is reduced, with an average error increased from the baseline network to 1.621 Å. This suggests that the template itself plays an important role in improving interaction prediction. In summary, all of the above factors have contributed to the improvement over our network, and templates may be an area that deserves continued attention.

Finally, we investigated the performance of AffineNet in the intra-domain distance and orientation. We present in Supplementary Table 7 the errors of intra- and inter-domain distance and orientation predicted by AffineNet. The results show that AffineNet has higher prediction accuracy in the intra-domain than inter-domain, which may be due to the flexibility of inter-domain orientation that makes the inter-domain interaction more difficult to be accurately captured than that of intra-domain. This also reflects that the inter-domain distance and orientation prediction is more challenging than intra-domain.

## Discussion
Since the structure predictions for multi-domain proteins are less accurate as on the domain level, it is a significant challenge to accurately capture the inter-domain orientation and thus

determine the multi-domain protein structure. In this work, we have proposed DeepAssembly, an automatic domain assembly protocol for multi-domain protein through inter-domain inter-actions predicted by a deep learning network. The results show that DeepAssembly is able to capture the correct inter-domain orientation, especially for the inter-domain distance, the average prediction error is lower than that of AlphaFold2.

In general, DeepAssembly performs best on the test set of multi-domain proteins because this deep learning model is trained specifically on multi-domain proteins to accurately cap-ture domain-domain interactions. Combined with the single-domain enhanced by our PAthreader, DeepAssembly constructs multi-domain protein structure with higher average accuracy than that modeled directly by AlphaFold2 on the given set. We found that AlphaFold2's inaccurate prediction for multi-domain protein structures is mainly due to the incorrect inter-domain orientation. For cases where the AlphaFold2-predicted structure is not accurate, DeepAssembly tends to perform more satisfac-torily. Meanwhile, DeepAssembly shows the potential to improve the accuracy of structures with low confidence deposited in AlphaFold database, providing an important solution for AlphaFold database update. In addition, DeepAssembly exhibits more excellent performance for assembling multi-domain pro-teins with experimental domain structures.

Moreover, we demonstrate that DeepAssembly can be applied to protein complex structure prediction by using inter-domain interactions learned on multi-domain proteins. It provides a more lightweight way to assemble protein structures by treating domains as assembly units, reducing the requirements for computational resources to some extent. Despite the promising assembly results, the applicability and accuracy of DeepAssembly could be further improved in several aspects. Firstly, in the absence of sufficient co-evolutionary signals and appropriate templates, our method still has room for improvement in per-formance. In this case, the introduction of physical and chemical features, e.g., protein-protein interface preferences are also cru-cial. Secondly, the domain-domain interaction can be further extended, because the inter-domain interactions learned only from monomeric multi-domain proteins are relatively limited after all. We also apply DeepAssembly to transmembrane protein complexes. The results show that it could correctly predict the interface of some homo-oligomeric membrane proteins, but the performance for targets that are large or flexible across trans-membrane regions is still not satisfactory. This is because the number of membrane proteins is rather limited and they have different structural characteristics, making it difficult for the model trained on mainly soluble proteins to effectively capture their inter-chain interactions. Address this challenge, it may be promising to borrow from some of the transfer learning methods.

## Methods

**Development set**. We collected a multi-domain protein dataset from our previously developed multi-domain protein structure database (MPDB) (http://zhanglab-bioinf.com/SADA/) (until September, 2021, with 48,225 entries) to develop the pipeline. This dataset contains a total of 10,593 multi-domain proteins, with each protein chain having 40 to 1,000 residues, sharing <40% sequence identity and having a resolution within 3.0 Å. These multi-domain proteins contain between 2-9 domains, determined by DomainParser[49] or confirmed by domain infor-mation for the corresponding entries in CATH[50] and SCOPe[51,52]. Specifically, there are 8,399 (79.3%) two-domain proteins, 1,681 (15.9%) three-domain proteins and 513 (4.8%) proteins with four

domains or more. In order to train a deep learning model for predicting inter-domain interactions, 1,0064 (95%) multi-domain proteins in this dataset are randomly selected as the training set, and the remaining 529 (5%) are used as the validation set.

**Test set**. We use multi-domain proteins with known structures from the comprehensive dataset in previous study DEMO[1] to test the developed pipeline. These proteins share <30% sequence identity, are collected by separately clustering the proteins with different domain types and structures from the template library[1]. Here, all the proteins in the initial DEMO dataset that have >30% sequence identity with any protein in the training set and vali-dation set are excluded, resulting in a total of 219 structures as the final test set.

To evaluate the performance of DeepAssembly in improving the inter-domain orientation, we further construct an indepen-dent test set of 164 multi-domain proteins. All entries come from 23,391 *H. sapiens* protein structures deposited in AlphaFold database (https://alphafold.ebi.ac.uk/), the PDB structure for each entry is directly downloaded from the PDB at https://www.rcsb.org/. Of these 23,391 proteins, the protein that has any of the following features is removed: (i) without experimental structure in the PDB; (ii) is confirmed as containing only one domain by DomainParser or corresponding entries in CATH and SCOPe database; (iii) having redundancy between each other for the UniProt primary accession; (iv) with a TM-score > 0.8 for the corresponding structure in AlphaFold database; (v) with less than 85% of solved residues for the experimental structure in the PDB, leading to a final set of 164 multi-domain proteins.

Further, we used 247 heterodimeric protein complexes with known interfaces from a previous study[41] to test the performance of DeepAssembly on protein complex prediction. These com-plexes share <30% sequence identity and have a resolution between 1-5 Å. The dataset consists of 56% eukaryotic proteins, 30% bacterial, 4% archaea, 2% virus and 8% from mixed kingdoms. For eukaryotic proteins, there are 36% from *H. sapiens*.

**Remote template recognition based on three-track alignment**. We use our recently developed remote template recognition method, PAthreader[29], to search for high-quality templates for inter-domain interactions prediction. PAthreader finds an opti-mal alignment between the query sequence and template sequence by a three-track alignment algorithm (sequence align-ment, residue pair alignment, and distance-structure profile alignment). Specifically, the distance profiles are represented as the probability distribution of pairwise residue distances pre-dicted by our in-house inter-residue distance predictor, Deep-DisPRE. The structure profiles are histogram distributions of pairwise residue distances, which are extracted from PAcluster80, a master structure database constructed by clustering PDB and AlphaFold database. The templates are eventually ranked by "rankScore" calculated with linear weighting of alignment score and predicted DMscore by a trained deep learning model. During training, in order to enhance the generalization ability of the network model and avoid overfitting, we restrict the available templates to up to 10 with the highest "rankScore" to increase the diversity of the template features used for training. At inference time, considering the influence of the template quality on the final structure, we provide at most the top 5 templates with the highest "rankScore" to the trained model according to the following criteria: (i) the template with "rankScore" >0.6; (ii) "rankScore" difference between the previous selected <0.03; (iii) the length of template more than 30% of the query sequence.

**Domain parsing and single-domain structure determination.**
Starting from the input sequence, we first predict the domain boundary through our developed DomBpred[53] by clustering the spatially close residues according to the inter-residue distance using a domain-residue clustering algorithm. Then we segment the input sequence according to the determined domain boundary to form several single-domain sequences. Next, the structural model of each domain is generated using an extended version of AlphaFold2 updated by replacing the template search component (HHsearch[46]) of AlphaFold2 with PAthreader. Specifically, remote templates for the sequence of each domain are searched separately through PAthreader with the same parameters used in template recognition for full-length sequence. For the sequence of each domain, we feed the top 4 templates identified by PAthreader into AlphaFold2 model and run the program with default parameters, selecting the first ranked model output as the final single-domain structural model. For single-domain with disconnected sequence, we first concatenate its disconnected sequences into a continuous sequence, and then input this sequence into the PAthreader model to generate the single-domain structure (Supplementary Fig. 10).

**Inter-domain interaction representation.** We represent the inter-domain interactions by affine transformation between the respective local coordinate system spaces of the inter-domain residues. For an inter-domain residue, the local coordinate system is established using three atoms Cα, C and N through the Gram-Schmidt process (Supplementary Note 2). The affine transformation includes the rotation and translation components of the local coordinate system transformation, reflects the relative position of the inter-domain residue pair. Here we convert the rotation matrix equivalently to Euler angles ($\alpha$, $\beta$ and $\gamma$), and project the translation vector into the spherical coordinate system as distance $r$, polar angle $\theta$ and azimuthal angle $\phi$ (see details in Supplementary Note 3). The six-dimensional (6D) vector ($\alpha$, $\beta$, $\gamma$, $r$, $\theta$, $\phi$) fully reflects the interaction of the inter-domain residue pair and its elements are predicted by a deep learning model we developed.

**Featurization.** The input features for the network are computed and aggregated into the following three categories: MSA features, template features and inter-domain features. The MSA features include one-hot encoded amino acid sequence, position-specific scoring matrix, positional entropy and co-evolutionary information, extracted from the MSAs generated by searching the Uniclust30[47] (version 2018_08) and the Big Fantastic Database[48] (BFD) using HHblits[46] (version 3.2.0) with default parameters and $e$-value of 1e-3. The template features are derived directly from the top $N_{templ}$ remote template structures searched by PAthreader ($N_{templ} = 10$ for training and 5 for inference). For each template, the local coordinate system for each residue is first establish, then the six elements ($\alpha$, $\beta$, $\gamma$, $r$, $\theta$, $\phi$) used to represent the interaction between residues are calculated, and finally they are converted into bins and one-hot encoded. In addition, for the complex, we connect its chains before search for its templates as a single chain through PAthreader, and then extract the template features of the connected single chain from the searched templates, which contain the features between two domains belonging to different chains. In order to pay more attention to the learning of inter-domain interaction, we especially introduce inter-domain features, including inter-domain contact information by pre-trained MSA Transformer[54] and a mask from the predicted domain boundary, indicating if the paired residues are in two different domains. More detailed descriptions of the features are listed in Supplementary Table 8.

**Network architecture.** We develop a deep learning model named AffineNet to predict the inter-domain interaction from the input features. The network architecture consists of basic residual blocks and an axial attention module. The MSA features are first concatenated with inter-domain features, and then the feature number is converted to 64 by a convolution layer with filter size 1. Next, the output is fed into the stack of 8 residual blocks, each consisting of two 1×1 convolution layer and a combined 3×3 convolution layer. In order to take full advantage of the remote template information, template features are fed into axial attention module along with the above output feature map. The axial attention alternates attention on the rows and columns of all features. The query, key and value are obtained from the input feature through the linear layer, and the attention map is the product of query and key. The softmax function is applied to the attention map to generate the weight map, which is then multiplied with the value map to obtain the final template features. Finally, the output template feature is concatenated with the feature map combined MSA and inter-domain features, and fed into the stack of 20 residual blocks. The convolution layer throughout the network is followed by the instance normalization layer and the ELU activation layer. After the last residual block, the network branches out into six independent paths, each consisting of a convolution layer followed by softmax activation.

**Training.** During the training, 95% of the proteins are randomly selected from the development set as the training set and the remaining 5% as the validation set. The sequence of large targets with more than 368 residues is randomly cropped to 368 lengths due to the limited GPU memory. The network is implemented in Python with Tensorflow1.14 and trained for at most 50 epochs. The Adam optimizer is adopted to minimize the loss of the prediction, where the total loss is a sum over the 6 individual cross-entropy losses with equal weight. The learning rate is initially set to 1e-4 and will gradually decrease as the epoch increases. The network model with the least validation loss is selected. As seen from the learning curves, the training and validation losses converge well for our deep learning model (Supplementary Fig. 11). It takes about 14 days to train the network on one NVIDIA Tesla V100s GPU.

**Affine transformation-based potential conversion.** To help guide the domain orientation assembly, the predicted distributions are converted into an affine transformation-based energy potential. For each inter-domain residue pair $(i, j)$, the interaction between them is represented as a 6D vector ($\alpha_{(i,j)}$, $\beta_{(i,j)}$, $\gamma_{(i,j)}$, $r_{(i,j)}$, $\theta_{(i,j)}$, $\phi_{(i,j)}$), and then the elements in the vector are equivalently converted into the affine matrix representing the transformation from the local coordinate system space of residue $j$ to that of residue $i$, according to the following formula:

$$\mathbf{A}_{(i,j)} = \begin{bmatrix} \mathbf{R}_{(i,j)} & \vec{\mathbf{t}}_{(i,j)} \\ \mathbf{0} & 1 \end{bmatrix}, \forall (i,j) \in \mathcal{S}_{\text{inter domain}} \quad (1)$$

where the rotation matrix $\mathbf{R}_{(i,j)}$ and translation vector $\vec{\mathbf{t}}_{(i,j)}$ are

calculated as follows:

$$\mathbf{R}_{(i,j)} = \begin{bmatrix} \cos\gamma_{(i,j)} & -\sin\gamma_{(i,j)} & 0 \\ \sin\gamma_{(i,j)} & \cos\gamma_{(i,j)} & 0 \\ 0 & 0 & 1 \end{bmatrix} \circ \begin{bmatrix} \cos\beta_{(i,j)} & 0 & \sin\beta_{(i,j)} \\ 0 & 1 & 0 \\ -\sin\beta_{(i,j)} & 0 & \cos\beta_{(i,j)} \end{bmatrix} \circ$$

$$\begin{bmatrix} 1 & 0 & 0 \\ 0 & \cos\alpha_{(i,j)} & -\sin\alpha_{(i,j)} \\ 0 & \sin\alpha_{(i,j)} & \cos\alpha_{(i,j)} \end{bmatrix} \tag{2}$$

$$\vec{\mathbf{t}}_{(i,j)} = \begin{bmatrix} r_{(i,j)}\sin\theta_{(i,j)}\cos\phi_{(i,j)}, & r_{(i,j)}\sin\theta_{(i,j)}\sin\phi_{(i,j)}, & r_{(i,j)}\cos\theta_{(i,j)} \end{bmatrix}^{\mathrm{T}} \tag{3}$$

where the elements $\alpha_{(i,j)}, \beta_{(i,j)}, \gamma_{(i,j)}, r_{(i,j)}, \theta_{(i,j)}, \phi_{(i,j)}$ are respectively derived from the values corresponding to the bin with the maximum probability in the predicted distributions.

According to the affine matrix transformed from the predicted distributions, an energy potential called atomic coordinate deviation (ACD) is constructed to guide the domain assembly, defined as:

$$\mathcal{F}_{\mathrm{ACD}} = \frac{1}{N_{\mathrm{tot}}} \sum_{(i,j),k} \left\| \mathbf{A}_i^{-1} \circ \vec{\mathbf{x}}_{j,k} - \mathbf{A}_{(i,j)} \circ \mathbf{A}_j^{-1} \circ \vec{\mathbf{x}}_{j,k} \right\|_2, \tag{4}$$

$$\forall (i,j) \in \mathcal{S}_{\mathrm{inter\ domain}}, \forall k \in \mathcal{S}_{\mathrm{atoms}}$$

where $N_{\mathrm{tot}}$ is the number of all the cumulative terms. $\mathcal{S}_{\mathrm{inter\ domain}}$ represents a set of all inter-domain residue pairs (note that both $(i,j)$ and $(j,i)$ are included), and $\mathcal{S}_{\mathrm{atoms}}$ is the set of backbone atoms in the residue ($\mathcal{S}_{\mathrm{atoms}} = \{N, C_\alpha, C, C_\beta\}$). The vector $\vec{\mathbf{x}}_{j,k}$ represents the position of atom $k$ in the $j$-th residue of the target structure relative to ground coordinate system. The matrices $\mathbf{A}_i$ and $\mathbf{A}_j$ represent the affine transformations from the local coordinate system of residues $i$ and $j$ to the ground coordinate system, respectively (the establishment process of the local coordinate system and the calculation details for the affine transformation are described in Supplementary Note 2).

**Population-based domain assembly and full atom-based refinement.** We assemble the full-length protein model in the domain assembly module. The overall architecture consists of three stages: first initial model creation stage, followed by iterative model update stages where a pool of multi-structures is maintained during the population-based rotation angle optimization, and finally full atom-based refinement. At the initialization stage, 1,000 initial full-length models are generated where the rotation angles $\varphi$, $\psi$ in the eight linker residues near their domain boundaries are set to random angles (the influence of the number of movable residues near the domain boundaries on the final structure is analyzed in Supplementary Note 4). To speed up the assembly, these initial models are represented as coarse-grained (centroid) models in Rosetta. The centroid model is a reduced representation of protein structure, in which the backbone remains fully atomic but each side chain is represented by a single artificial atom (centroid). In the iterative annealing stage, series of rotation angle sampling, new structure generation, and structure pool selection steps are repeated iteratively with coarse-grained representation. At each iteration, 1,000 new individuals are generated through evolution (crossover and mutation) between linker rotation angles of individuals in the current pool, and then the lowest energy between each new individual and its original individual is selected to be retained in the new pool according to the optimization objective. This process is repeated for 500 iterations (the accuracy of the final models at different iteration numbers is shown in Supplementary Fig. 12), and the objective function is the atomic coordinate deviation potential $\mathcal{F}_{\mathrm{ACD}}$. For the pool that reaches the maximum iteration, the top 10 centroid models (ranked by energy) are selected to be converted to full-atomic models by generating side chains in Rosetta, followed by full-atom refinement by *FastRelax* protocol with ref2015 energy function in Rosetta[55].

**Quality assessment of built models.** We use a developed model quality assessment tool, GraphCPLMQA[33], a graph coupled network based on embeddings of protein language model to assess the quality of the model built by DeepAssembly. GraphCPLMQA utilizes both the sequence and structure embeddings generated by the protein language model, with various protein model structure features. They are input to the encoder-decoder module of the graph coupling network, and the mapping relationship among sequence, structure and quality is obtained to predict the quality of protein models.

**Evaluation metrics.** We use three types of metrics to evaluate the quality of the multi-domain protein and complex model built by DeepAssembly, and the predicted inter-domain distance precision. The first type of metrics is one that measure the closeness between the built model and the experimental structure in PDB. In this respect, we adopt the RMSD and TM-score[34] between the predicted multi-domain protein model and the corresponding experimental structure calculated by TM-score tool downloaded at https://zhanggroup.org/TM-score/. The second type is one that measures the interface quality of predicted protein complex model. In this regard, we report DockQ score[39] calculated by public tool (https://github.com/bjornwallner/DockQ/). Besides the above metrics, we also reported the inter-domain distance error to evaluate the predicted inter-domain distance precision. The detailed descriptions of above metrics are given in Supplementary Note 1.

**Statistics and reproducibility.** All data were carefully collected and analyzed using standard statistical methods. Comprehensive information on the statistical analyses used is included in various places, including the figures, figure legends and results.

**Reporting summary.** Further information on research design is available in the Nature Portfolio Reporting Summary linked to this article.

## Data availability

The authors declare that the data supporting the results and conclusions of this study are available within the paper and its Supplementary Information. A full list with the links of multi-domain proteins used in this study is available in Supplementary Data 1. The sequence database of Uniclust30_ 2018_08 used in this study is available at https://uniclust.mmseqs.com/. The sequence database of Big Fantastic Database (BFD) used in this study is available at https://bfd.mmseqs.com/. The source data underlying Figs. 2, 3, 4, 5, 6, 8, 9 are provided in Supplementary Data file.

## Code availability

The online server of DeepAssembly is made freely available at http://zhanglab-bioinf.com/DeepAssembly/.

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

## Acknowledgements

We thank members of the Guijun Zhang lab for discussion and feedback. We thank Haitao Zhu for helping to develop the server and organize the experimental data. Computational resources were provided by the College of Information Engineering at Zhejiang University of Technology. This work was supported by the National Key R&D Program of China [2022ZD0115103], the National Nature Science Foundation of China [62173304, 62203389], the Key Project of Zhejiang Provincial Natural Science Foundation of China [LZ20F030002].

## Author contributions

G.Z. conceived and designed the research. G.Z., Y.X. and K.Z. developed the pipeline and performed the test. G.Z. and K.Z. developed the method for template recognition and developed the server. G.Z. and D.L. developed the method for model quality assessment. G.Z., Y.X. and X.Z. analyzed data. G.Z. and Y.X. wrote the manuscript, and all authors read and approved the final manuscript.

## Competing interests

The authors declare no competing interests.

**Additional information**

**Peer review information** : *Communications Biology* thanks Shuguang Yuan and the other, anonymous, reviewer(s) for their contribution to the peer review of this work. Primary Handling Editors: Yuedong Yang and Anam Akhtar. A peer review file is available.

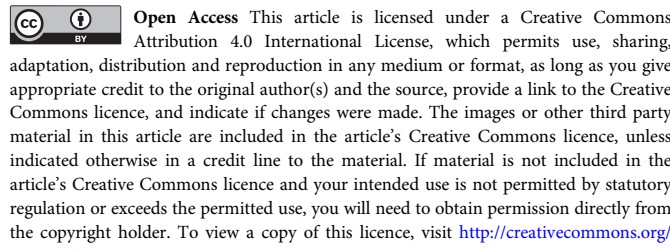

