## [Peer Review File · Communications Biology]

Reviewers' comments:

Reviewer #1 (Remarks to the Author):

This manuscript describes a novel method (DeepAssembly) for assembling multiple domains by minimizing the atomic coordinate deviation predicted by the AffineNet. In addition, the authors expand the DeepAssembly to the complex structure prediction of heterodimers. Overall, DeepAssembly is an interesting study. The manuscript is written well. However, there still are some questions needed to be addressed.

1. DeepAssembly uses the proposed PAtreader to search for the remote template of a target sequence and its multi-domain. What is the detailed criteria of screening? Are the templates with a high sequence identity excluded? For the heterodimer, how to generate the template features between two domains belonging to different chains?

2. Why does DeepAssembly use different numbers and criteria for training and inference on single-domain?

3. In the case of MSA with a large number of sequences, the template information seems to have little impact on the AlphaFold2. Can the authors investigate the performance of DeepAssembly with single-domains generated by PAtreader and AlphaFold2 in that situation?

4. There are some domains that are spatially adjacent but disconnected in sequence. How does DeepAssembly predict the single-domain structures with disconnected sequences?

5. DeepAssembly uses four residues near the domain boundaries to generate the initial full-length structure and optimizes it. Is this too small to handle the full-length structure which is made up of two domains with a long loop?

6. Can the authors analyze the influence of the number of iterations on the final structure?

7. DeepAssembly uses the Euler angle and spherical coordinate system to represent the rotation matrix and translation vector, and divides the angles to several bins. Would it be better to predict the angle in terms of sines and cosines?

8. DeepAssembly obtains an average TM-score of 0.602 on the targets predicted by AlphaFold2 with a TM-score less than 0.5, compared with 0.414 of AlphaFold2. The improvement of 45.4% is not consistent with the value shown in the Supplementary Figure 1b.

9. Can DeepAssembly be compared with AlphaFold-linker or AlphaFold-gap on the heterodimers? In addition, how about the performance of DeepAssembly on hetero-complexes with more than 2 chains?

10. It is a bit surprising that DeepAssembly obtains a better average TM-score (0.916) with single-domains predicted by PAtreader than that with experimental single-domains (0.855). Can the authors explain it?

11. DeepAssembly is compared with other methods according to the predicted inter-domain distance. Did the authors directly use the distance predicted by the AffineNet or use the structure selected by the evolutionary algorithm? If the latter is used, did the authors use the structure refined by FastRelax?

12. Does the AffineNet use the cross-entropy losses on both intra- and inter-domain? If so, how about the performance of AffineNet in the intra-domain distance and orientation?

Reviewer #2 (Remarks to the Author):

Xia et al presented a new tool to predict Domain-based multi-domain protein and complex structure. It is an interesting work. However, there are already several similar tools were already published before. So, it would be interesting to make comparison between this new tool and others, including the algorithm difference, merits and demerits, applicable system.

The benchmark is mainly focusing on water soluble proteins. It would be interesting to see how it perform well for membrane proteins as well.

Moreover, I don't see too much about the sidechain conformation details. It would be necessary to present this information as well.

Reviewer #3 (Remarks to the Author):

The study presented by Zhang and colleagues seeks to demonstrate that a new algorithm, DeepAssembly, is a significant advance over other multidomain/multiprotein structure prediction algorithms, AlphaFold2, AlphaFoldMultimer and RosettaFold2, but also previous algorithms from the lap DEMO and SADA.

Despite the limitations pointed out in the concluding summary/perspective, DeepAssembly does a considerably better job than the others at all levels. However, the devil is in the details. Specifically, this reviewer is not convinced that a commonly agreed-on multidomain or complexed protein data base is used. Rather, results are reported by 219 and then 165 proteins but is is not clear how these are selected. It would be more convincing to apply DeepAssembly to Casp14 and 15 targets first.

For a more general and biological audience of communications biology, the interest in this topic would be marginal in absence of a clear prediction how cell signaling or transmembrane proteins (or their mechanism of activation) can be predicted. The paper seems more suitable for an expert journal whose readers appreciate the details of the algorithms and findings. As presented at the moment, there is too little information to introduce readers to some of the parameters, e.g. TMScore, DockQ score and cut-offs etc. The figures need a lot more explanation, e.g. what does n refer to in Fig.2.

Finally, we tried to use the DeepAssembly server but did not receive any feedback/results within a week - no information on queue status etc. was given. So the server might not be working yet.

Gui-Jun Zhang
College of Information Engineering
Zhejiang University of Technology
Hangzhou 310023, China
zgj@zjut.edu.cn
August 29, 2023

Dear Reviewers,

Thank you for reviewing our manuscript entitled “Domain-based multi-domain protein and complex structure prediction using inter-domain interactions from deep learning” (COMMSBIO-23-1703) and for proposing valuable suggestions. The comments helped us improve our manuscript and served as important guidance to our research. We have carefully considered the comments and have revised the manuscript accordingly. All modified parts are highlighted in the manuscript.

Thank you again for reviewing our manuscript. We hope you find that the changes are satisfactory.

Sincerely,

Yuhao Xia, Kailong Zhao, Dong Liu, Xiaogen Zhou, and Guijun Zhang

Response to Reviewer #1

We very much appreciate for your comments and suggestions, which help to significantly improve the quality and description of our manuscript. In the following, we include point-by-point response to the comments, where all changes have been highlighted in the manuscript.

This manuscript describes a novel method (DeepAssembly) for assembling multiple domains by minimizing the atomic coordinate deviation predicted by the AffineNet. In addition, the authors expand the DeepAssembly to the complex structure prediction of heterodimers. Overall, DeepAssembly is an interesting study. The manuscript is written well. However, there still are some questions needed to be addressed.

Response: We appreciate your positive comments on this work. The point-by-point response to the comments are shown below.

1. DeepAssembly uses the proposed PAtreader to search for the remote template of a target sequence and its multi-domain. What is the detailed criteria of screening? Are the templates with a high sequence identity excluded? For the heterodimer, how to generate the template features between two domains belonging to different chains?

Response: Thank you for raising these important questions. We apologize for not providing a detailed explanation of these aspects. We respond to these questions as follows:

(1) What is the detailed criteria of screening?

PAtreader finds an optimal alignment between the query sequence and template sequence by a three-track alignment algorithm [R1]. The templates are eventually ranked by “rankScore”. We use different numbers and screening criteria of templates for training and inference. During training, in order to enhance the generalization ability of the network model and avoid overfitting, we restrict the available templates to up to 10 with the highest “rankScore” to increase the diversity of the template features used for training. At inference time, considering the influence of the template quality on the final structure, we provide at most the top 5 templates with the highest “rankScore” to the trained model according to the following criteria: (i) the template with “rankScore” >0.6; (ii) “rankScore” difference between the previous selected <0.03; (iii) the length of template more than 30% of the query sequence.

We have added corresponding descriptions in the manuscript.

with linear weighting of alignment score and predicted DMscore by a trained deep learning model. During training, in order to enhance the generalization ability of the network model and avoid overfitting, we restrict the available templates to up to 10 with the highest “rankScore” to increase the diversity of the template features used for training. At inference time, considering the influence of the template quality on the final structure, we provide at most the top 5 templates with the highest “rankScore” to the trained model according to the following criteria: (i) the template with “rankScore” >0.6; (ii) “rankScore” difference between the previous selected <0.03; (iii) the length of template more than 30% of the query sequence.

(2) Are the templates with a high sequence identity excluded?

This is twofold, with DeepAssembly using templates with different sequence identity cutoffs in different experiments for fair comparison with other methods.

(a) On the one hand, in order to thoroughly evaluate DeepAssembly's ability in real-world scenarios, we used single-domains predicted by PAM-Threader to assemble multi-domain proteins. In this experiment, we maximized the performance of DeepAssembly, and compared it with the current state-of-art end-to-end method AlphaFold2. In the inference process of AlphaFold2, the templates with a high sequence identity are not excluded. Therefore, for a fair comparison, we also did not exclude templates according to sequence identity in this experiment.

(b) On the other hand, in order to rigorously test the model performance of DeepAssembly, we assembled multi-domain proteins with experimental domains to exclude the influence of single-domains on the final structure. In this experiment, DeepAssembly was compared with SADA, DEMO and AIDA. In SADA and DEMO, structural templates with a sequence identity $>30\%$ to the query are excluded [R2, R3], so for a fair comparison, templates with sequence identity $>30\%$ of the query are also excluded in DeepAssembly.

We apologize for not providing detailed experimental settings, and we have added corresponding descriptions in the manuscript.

In order to rigorously test the model performance of DeepAssembly, we assemble multi-domain proteins with experimental domains to exclude the influence of single-domains on the final structure. In this experiment, DeepAssembly is compared with SADA, DEMO and AIDA, and for a fair comparison, we here exclude templates in DeepAssembly with a sequence identity $>30\%$, as done in SADA and DEMO. **Fig. 8** presents a summary of the models assembled by DeepAssembly, SADA, DEMO and AIDA. The detailed

(3) For the heterodimer, how to generate the template features between two domains belonging to different chains?

For the heterodimer, we connect its two chains before search for its templates as a single chain through PAM-Threader. Although PAM-Threader is designed for single-chain proteins, it can also be used to recognize templates of protein complexes [R1]. Subsequently, we extract the template features of the connected single chain from the searched templates by calculating the distance and orientation between residues in the templates. Therefore, the extracted template features of the connected single chain contain the template features between two domains belonging to different chains.

We have added corresponding descriptions in the manuscript.

between residues are calculated, and finally they are converted into bins and one-hot encoded. In addition, for the complex, we connect its chains before search for its templates as a single chain through PAM-Threader, and then extract the template features of the connected single chain from the searched templates, which contain the features between two domains belonging to different chains. In order to pay more attention to

2. Why does DeepAssembly use different numbers and criteria for training and inference on single-domain?

Response: Thank you for raising this important point. We apologize for the unclear descriptions.

(1) During the training process, in order to enhance the generalization ability of the network model and avoid overfitting, we used the top 10 templates with the highest “rankScore” searched by PAtreader to increase the diversity of the template features used for training. While for the inference, considering the influence of the template quality on the final structure, we therefore used different numbers of PAtreader templates with stricter screening criteria compared with the training process, to obtain high-precision predicted inter-domain interactions and single-domain structures.

(2) In addition, we used predicted and experimental single-domains in domain assembly, respectively, depending on the experimental scenarios. First, considering that in real-world scenarios, most proteins do not have experimentally solved domains, we therefore used single-domains predicted by PAtreader to assemble multi-domain proteins to evaluate DeepAssembly's performance in real-world scenarios. Meanwhile, we added the controlled experiment to assemble multi-domain with single-domains by AlphaFold2 to investigate the factors that contribute to the performance of DeepAssembly. Second, we used experimental domains to assemble multi-domain proteins, which is to exclude the influence of single-domains on the final structure to rigorously test the model performance of DeepAssembly.

We have added corresponding descriptions in the manuscript.

with linear weighting of alignment score and predicted DMscore by a trained deep learning model. During training, in order to enhance the generalization ability of the network model and avoid overfitting, we restrict the available templates to up to 10 with the highest “rankScore” to increase the diversity of the template features used for training. At inference time, considering the influence of the template quality on the final structure, we provide at most the top 5 templates with the highest “rankScore” to the trained model according to the following criteria: (i) the template with “rankScore” >0.6; (ii) “rankScore” difference between the previous selected <0.03; (iii) the length of template more than 30% of the query sequence.

We then investigate the factors that contribute to the performance of DeepAssembly. Here, we test the control version of DeepAssembly that uses AlphaFold2-predicted domains as the single-domain input, denoted as “DeepAssembly (AF2 domain)”. Even if the domains are replaced by AlphaFold2 predictions,

In order to rigorously test the model performance of DeepAssembly, we assemble multi-domain proteins with experimental domains to exclude the influence of single-domains on the final structure. In this experiment, DeepAssembly is compared with SADA, DEMO and AIDA, and for a fair comparison, we here exclude templates in DeepAssembly with a sequence identity >30%, as done in SADA and DEMO. **Fig. 8** presents a summary of the models assembled by DeepAssembly, SADA, DEMO and AIDA. The detailed

3. In the case of MSA with a large number of sequences, the template information seems to have little impact on the AlphaFold2. Can the authors investigate the performance of DeepAssembly with single-domains generated by PAtreader and AlphaFold2 in that situation?

Response: Thank you for your valuable suggestion. Based on your suggestion, we investigated the performance of DeepAssembly with single-domains generated by PAtreader and AlphaFold2 in the case of MSA with a large number of sequences. **Figure R1** is the scatterplot of the TM-score of models predicted by DeepAssembly and DeepAssembly (AF2 domain) versus the number of effective sequences (Neff) in MSAs. “DeepAssembly” and “DeepAssembly (AF2 domain)” represent DeepAssembly with single-domains generated by PAtreader and AlphaFold2, respectively. The fitting curves in **Figure R1** show that the TM-score increases as Neff increases until it largely saturates when Neff is higher than 100. In addition, as Neff increases, the curves representing DeepAssembly and DeepAssembly (AF2 domain) gradually tend to coincide. This indicates that when the MSA is shallow, the template information can greatly improve the performance, while when the MSA is deep enough, the co-evolutionary information plays a major role. This is as the reviewer pointed out, in the case of large MSA depth, the template information has relatively little impact compared to the co-evolutionary information.

Table R1 shows the performance of DeepAssembly and DeepAssembly (AF2 domain) on the targets with MSA Neff in the ranges of (100,1000) and (1000,10000). Specifically, when Neff scores between 100 and 1000, the average TM-score of DeepAssembly and DeepAssembly (AF2 domain) are 0.923 and 0.922, respectively. For the targets with Neff greater than 1000, the average TM-score of DeepAssembly and DeepAssembly (AF2 domain) are 0.932 and 0.924, respectively. The results show that in the case of MSA with a large number of sequences, the performance of DeepAssembly with PAtreader single-domains is superior to that with AlphaFold2 domains, although the improvement is not very significant compared to the case with less Neff in MSAs.

Figure R1. Scatterplot of the TM-score of models predicted by DeepAssembly and DeepAssembly (AF2 domain) versus the number of effective sequences (Neff) in MSAs. The orange and blue curves are obtained by fitting the orange and blue dots with ExpDec1 function in Origin. The shaded area is the 95% confidence interval.

Table R1. The performance of DeepAssembly and DeepAssembly (AF2 domain) on the targets with MSA Neff in the ranges of (100,1000) and (1000,10000).

Method	100 < Neff < 1000		1000 < Neff < 10000	
	RMSD (Å)	TM-score	RMSD (Å)	TM-score
DeepAssembly	2.91	0.923	2.20	0.932
DeepAssembly (AF2 domain)	3.02	0.922	2.61	0.924

We have added corresponding descriptions in the manuscript and Supplementary Information.

We further analyze the performance of DeepAssembly with single-domains generated by PAtreader and AlphaFold2 in the case of MSA with different number of sequences. **Supplementary Fig. 1** is the scatterplot of the TM-score of models predicted by DeepAssembly and DeepAssembly (AF2 domain) versus the number of effective sequences (Neff) in MSAs. The fitting curves in **Supplementary Fig. 1** show that the TM-score increases as Neff increases until it largely saturates when Neff is higher than 100. In addition, as Neff increases, the curves representing DeepAssembly and DeepAssembly (AF2 domain) gradually tend to coincide. This indicates that when the MSA is shallow, the template information can greatly improve the performance, while when the MSA is deep enough, the co-evolutionary information plays a major role.

Supplementary Fig. 1 | Scatterplot of the TM-score of models predicted by DeepAssembly and DeepAssembly (AF2 domain) versus the number of effective sequences (Neff) in MSAs. The orange and blue curves are obtained by fitting the orange and blue dots with ExpDec1 function in Origin. The shaded area is the 95% confidence interval.

4. There are some domains that are spatially adjacent but disconnected in sequence. How does DeepAssembly predict the single-domain structures with disconnected sequences?

Response: Thank you for raising the important question. We apologize for not providing too much details on this. For single-domain with disconnected sequence, we first concatenate its disconnected sequences into a continuous sequence, and then input this sequence into the PAtreader model to generate the single-domain structure (**Figure R2**).

Figure R2. Diagram of the process for predicting the single-domain structure with disconnected sequences. Two single-domain structures are colored by blue and red, respectively.

We have added corresponding descriptions in the manuscript and Supplementary Information.

AlphaFold2 with PAtreader. Specifically, remote templates for the sequence of each domain are searched separately through PAtreader with the same parameters used in template recognition for full-length sequence. For the sequence of each domain, we feed the top 4 templates identified by PAtreader into AlphaFold2 model and run the program with default parameters, selecting the first ranked model output as the final single-domain structural model. For single-domain with disconnected sequence, we first concatenate its disconnected sequences into a continuous sequence, and then input this sequence into the PAtreader model to generate the single-domain structure (**Supplementary Fig. 10**).

Supplementary Fig. 10 | Diagram of the process for predicting the single-domain structure with disconnected sequences. Two single-domain structures are colored by blue and red, respectively.

5. DeepAssembly uses four residues near the domain boundaries to generate the initial full-length structure and optimizes it. Is this too small to handle the full-length structure which is made up of two domains with a long loop?

Response: Thank you for pointing out this important issue. **Based on your valuable suggestion, we increased the number of movable residues near the domain boundaries, which helped to further improve the performance of our method.** In order to investigate the influence of the number of movable residues near the domain boundaries on the final structure, we added the experiment by increasing the number of movable residues to 8, and reassembling the full-length structure using PAtreader single-domains on the test set. **Table R2** summarizes the performance of DeepAssembly for assembling the full-length structure using 4 and 8 residues near the domain boundaries, respectively. The results show that the average TM-score is improved from 0.916 to 0.922 by using 8 residues near the domain boundaries. In addition, the number of models with TM-score >0.9 also increased from 173 to 178. This suggests that more movable residues near the domain boundaries may increase the flexibility of the linker during optimization, making it easier to sample the correct inter-domain orientation, thereby improving the prediction accuracy.

Table R2. The performance of DeepAssembly for assembling the full-length structure using 4 and 8 residues near the domain boundaries, respectively.

Method	RMSD (Å)	TM-score	#TM-score>0.9
DeepAssembly (4 residues)	3.12	0.916	173
DeepAssembly (8 residues)	2.91	0.922	178

We show a case of three-domain protein (PDB ID: 1GRI_A) which is made up of three domains with two long loops. In **Figure R3**, experimental structure is colored in gray, and the predicted models of DeepAssembly using 4 and 8 residues near the domain boundaries are colored in green and blue, respectively. The two long loops connecting the three domains in the experimental structure are colored in red. It can be seen from **Figure R3** that the DeepAssembly model by using 8 movable residues in the linker is more accurate than using 4 residues (the TM-score improved from 0.752 to 0.958). In **Figure R3a**, the linker in the predicted structure is broken and there is a large deviation compared with the experimental structure (shown in the yellow boxes). However, by increasing the number of movable residues in linker, the inter-domain orientation of the predicted structure is not only improved, but the linker's shape is also closer to the experimental structure (**Figure R3b**).

It means that originally set 4 residues are too few to handle the full-length structure which is made up of two domains with a long loop, as pointed out by the reviewer. **Therefore, we set the number of movable residues near the domain boundaries to 8 in the domain assembly module of DeepAssembly. We re-performed the experiments assembling full-length structures and updated the corresponding experimental data and result analysis in the manuscript and Supplementary Information.**

Figure R3. The case of a three-domain protein (PDB ID: 1GRI_A) which is made up of three domains with two long loops. Experimental structure is colored in gray, and the predicted models of DeepAssembly using 4 and 8 residues near the domain boundaries are colored in green (**a**) and blue (**b**), respectively. The two long loops connecting the three domains in the experimental structure are colored in red.

Thank you again for your valuable suggestion, which helped to further improve the performance of our method. We have added corresponding descriptions and updated the experimental data in the manuscript.

full atom-based refinement. At the initialization stage, 1,000 initial full-length models are generated where the rotation angles φ , ψ in the eight linker residues near their domain boundaries are set to random angles (the influence of the number of movable residues near the domain boundaries on the final structure is analyzed in **Supplementary Section 4**). To speed up the assembly, these initial models are represented

Supplementary Section 4. The influence of the number of movable residues near the domain boundaries on the final structure

In order to investigate the influence of the number of movable residues near the domain boundaries on the final structure, we assemble the full-length structure using 4 and 8 residues near the domain boundaries, respectively. The result shows that the average TM-score is improved from 0.916 to 0.922 by using 8 residues near the domain boundaries (see table). In addition, the number of models with TM-score >0.9 also increased from 173 to 178. We show a case of three-domain protein (PDB ID: 1GRI_A) which is made up of three domains with two long loops. It can be seen from the figure that the DeepAssembly model by using 8 movable residues in the linker is more accurate than using 4 residues (the TM-score improved from 0.752 to 0.958). The linker in the predicted structure is broken and there is a large deviation compared with the experimental structure (shown in the yellow boxes). However, by increasing the number of movable residues in linker, the inter-domain orientation of the predicted structure is not only improved, but the linker's shape is also closer to the experimental structure. This suggests that more movable residues near the domain boundaries may increase the flexibility of the linker during optimization, making it easier to sample the correct inter-domain orientation, thereby improving the prediction accuracy. Therefore, we set the number of movable residues near the domain boundaries to 8 in the domain assembly module of DeepAssembly.

Figure. The case of a three-domain protein (PDB ID: 1GRI_A) which is made up of three domains with two long loops. Experimental structure is colored in gray, and the predicted models of DeepAssembly using 4 and 8 residues near the domain boundaries are colored in green (a) and blue (b), respectively. The two long loops connecting the three domains in the experimental structure are colored in red.

Table. The performance of DeepAssembly for assembling the full-length structure using 4 and 8 residues near the domain boundaries, respectively. #TM-score>0.9 represents the number of models with TM-score > 0.9.

Method	RMSD (Å)	TM-score	#TM-score>0.9
DeepAssembly (4 residues)	3.12	0.916	173
DeepAssembly (8 residues)	2.91	0.922	178

Supplementary Table 1. Summary of the performance of DeepAssembly and AlphaFold2 for predicting multi-domain proteins. #TM-score>0.9 represents the number of models with TM-score > 0.9.

Method	RMSD (Å)	TM-score	#TM-score>0.9
DeepAssembly	2.91	0.922	178
DeepAssembly (AF2 domain)	3.11	0.919	176
AlphaFold2	3.58	0.900	166

Supplementary Table 4. TM-score of the models assembled by DeepAssembly, SADA, DEMO and AIDA using experimental domain structures. "2dom", "3dom", and "m4dom" represent the classification of proteins with two, three, and more than four domains, respectively.

Method	2dom		3dom		m4dom		all	
	average	median	average	median	average	median	average	median
DeepAssembly	0.896	0.985	0.851	0.984	0.725	0.840	0.856	0.976
SADA	0.840	0.903	0.709	0.688	0.582	0.555	0.763	0.782
DEMO	0.779	0.782	0.649	0.628	0.517	0.504	0.702	0.698
AIDA	0.671	0.662	0.514	0.497	0.424	0.405	0.589	0.581

6.Can the authors analyze the influence of the number of iterations on the final structure?

Response: Thank you for your valuable suggestion. Based on your suggestion, we supplemented experiments to analyze the influence of the number of iterations on the final structure. We set the number of iterations to 100, 300, 500, 700 and 900, respectively, and then performed experiments to generate the final structures on the test set. **Figure R4** shows the average TM-score and RMSD of the final structures at different iteration numbers. When the number of iterations is small (e.g., the number of iterations = 100), since the conformational sampling is not sufficient at this time, the global minimum in the conformational space has not been explored, so the accuracy of the final structure is relatively low, especially for those proteins with more domains. The accuracy of the final structure increases as the number of iterations increases until it largely saturates when the number of iterations is greater than 500, at which point the optimization process gradually converges. Therefore, considering both the accuracy of the final structure and the sampling efficiency, we set the number of iterations to 500 during the optimization process of DeepAssembly.

Figure R4. The average TM-score and RMSD of the final structures at different iteration numbers.

The following are the revisions we have made in the manuscript and Supplementary Information.

individual and its original individual is selected to be retained in the new pool according to the optimization objective. This process is repeated for 500 iterations (the accuracy of the final models at different iteration numbers is shown in **Supplementary Fig. 12**), and the objective function is the atomic coordinate deviation

Supplementary Fig. 12 | The average TM-score and RMSD of the final structures at different iteration numbers. The accuracy of the final structure increases as the number of iterations increases until it largely saturates when the number of iterations is greater than 500, at which point the optimization process gradually converges.

7.DeepAssembly uses the Euler angle and spherical coordinate system to represent the rotation matrix and translation vector, and divides the angles to several bins. Would it be better to predict the angle in terms of sines and cosines?

Response: Thank you for your valuable suggestion. **Based on your suggestion, we retrained a network model to predict the angle in terms of sines and cosines.** During the training process, the same training set and model parameters are used as before. Subsequently, we converted the predicted sines and cosines of angles into the energy potential to guide the generation of the final full-length structure, and compared it with the original way of predicting the angles directly. Here, the full-length structures are all assembled by single-domain predicted by PAtreader, and all used 8 movable residues near the domain boundaries. **Table R3** summarizes the performance of DeepAssembly for assembling the full-length structure by predicting the angles directly and predicting the angle in terms of sines and cosines, respectively. “*” indicates predicting the angle in terms of sines and cosines. The results show that it seems to have no obvious difference in the accuracy of the final structure by predicting the angle in terms of sines and cosines. However, this is a novel and interesting idea that converts the angle into sines and cosines, and provides an important line of thought for our subsequent research.

Table R3. The performance of DeepAssembly for assembling the full-length structure by predicting the angles directly and predicting the angle in terms of sines and cosines, respectively. “*” indicates predicting the angle in terms of sines and cosines.

Method	RMSD (Å)	TM-score	#TM-score>0.9
DeepAssembly	2.91	0.922	178
DeepAssembly*	3.37	0.914	173

8.DeepAssembly obtains an average TM-score of 0.602 on the targets predicted by AlphaFold2 with a TM-score less than 0.5, compared with 0.414 of AlphaFold2. The improvement of 45.4% is not consistent with the value shown in the Supplementary Figure 1b.

Response: Thank you for raising this important point. We apologize for the ambiguous description. In fact, **the value corresponding to the 0.5 cutoff in the Supplementary Figure 1b is calculated differently from the 45.4% improvement mentioned in the text.** Supplementary Figure 1b shows the average TM-score improvement rate for targets improved by DeepAssembly at each cutoff. Specifically, there are 7 targets predicted by AlphaFold2 with a TM-score <0.5. The average TM-score of DeepAssembly and AlphaFold2 is 0.602 and 0.414, respectively, with a **45.4%** ($= (0.602 - 0.414) / 0.414 * 100%$) improvement. In addition, there are 71.4% (5 of 7) are improved by DeepAssembly on these 7 targets predicted by AlphaFold2 with a TM-score < 0.5. For these targets improved by DeepAssembly, the average TM-score improvement rate is **73.5%** ($= (78.3% + 35.4% + 15.8% + 188.7% + 49.1%) / 5$), as shown in the Supplementary Figure 1b. **Table R4** shows the TM-scores and the TM-score improvement rates for these 7 targets.

Table R4. TM-scores and the TM-score improvement rates for targets predicted by AlphaFold2 with a TM-score less than 0.5.

PDB ID	TM-score		TM-score improvement rate (%)
	DeepAssembly	AlphaFold2	
1mkfA	0.708	0.397	78.3
2hjqA	0.635	0.469	35.4
2jz4A	0.462	0.471	\
2yrqA	0.484	0.418	15.8
3uitD	0.477	0.484	\
4lziA	0.967	0.335	188.7
3hjlA	0.483	0.324	49.1
Average	0.602	0.414	73.5

We thank you again for pointing this out. In order to avoid misunderstandings, we have clarified the corresponding descriptions in the manuscript.

(Note: The above data has been updated in the revised manuscript because we re-performed the experiments according to the Comment 5, and the original Supplementary Fig. 1 was updated to Supplementary Fig. 2.)

with AlphaFold2 (with an average increase from 0.414 to 0.601). Similar conclusion can also be drawn from the percentage of the improved proteins by DeepAssembly and the average TM-score improvement rate for the improved proteins in **Supplementary Fig. 2a,b**. With the lower the accuracy of models predicted by AlphaFold2, the higher TM-score improvement rate of DeepAssembly for these proteins. Specifically, DeepAssembly improves 71.4% of targets predicted by AlphaFold2 with a TM-score less than 0.5, and achieves an average TM-score improvement rate of 72.4% on these improved proteins. The results reflect the potential of DeepAssembly to improve the accuracy of multi-domain proteins that inter-domain orientations are difficult to capture correctly.

9.Can DeepAssembly be compared with AlphaFold-linker or AlphaFold-gap on the heterodimers? In addition, how about the performance of DeepAssembly on hetero-complexes with more than 2 chains?

Response: Thank you for your valuable suggestion.

(1) Can DeepAssembly be compared with AlphaFold-linker or AlphaFold-gap on the heterodimers?

Based on your valuable suggestion, we compared the performance of DeepAssembly and AlphaFold-linker on the heterodimers. For AlphaFold-linker, we obtained predicted models by adding a 21-residue

repeated Glycine-Glycine-Serine linker between each chain and then running it as a single chain through the AlphaFold2 model [R4].

Table R5 summarizes the performance of DeepAssembly and AlphaFold-linker on the 247 heterodimers. The success rate of DeepAssembly and AlphaFold-linker are 32.4% and 40.9%, respectively. AlphaFold-linker obtained more acceptable models ($0.23 \leq \text{DockQ} < 0.49$). However, for the medium model ($0.49 \leq \text{DockQ} < 0.80$), the number of which generated by DeepAssembly is close to AlphaFold-linker (48 vs. 51). In addition, DeepAssembly achieved a higher DockQ score than AlphaFold-linker on 36% of the test cases.

The results show that our method can almost achieve the performance of AlphaFold-linker with less computational resource requirements by treating domains as assembly units, especially it is superior to AlphaFold-linker on several proteins. This further validates the effectiveness of our method that is based on domain assembly.

Table R5. The performance of DeepAssembly and AlphaFold-linker on the heterodimers. Acceptable: $0.23 \leq \text{DockQ} < 0.49$, Medium: $0.49 \leq \text{DockQ} < 0.80$, High: $\text{DockQ} \geq 0.80$.

Method	Success rate (SR) (%)	Acceptable Count	Medium Count	High Count
DeepAssembly	32.4	21	48	11
AlphaFold-linker	40.9	31	51	19

We have added the corresponding description in the manuscript, and the following are the revisions we have made in the manuscript.

In the test set, DeepAssembly successfully predicted the interface ($\text{DockQ} \geq 0.23$) in 32.4% of cases. We compared its performance with RoseTTAFold and AlphaFold-linker (**Supplementary Table 3**). The RoseTTAFold models are obtained by running its end-to-end version, and AlphaFold-linker models are obtained by adding a 21-residue repeated Glycine-Glycine-Serine linker between each chain and then running it as a single chain through the AlphaFold2²⁷. Overall, DeepAssembly has an improvement of 74.2% compared to RoseTTAFold (SR = 18.6%), is better than RoseTTAFold for 66% cases, and generates more high-quality models (**Fig. 5a**). Meanwhile, our method can almost achieve the performance of AlphaFold-linker (SR = 40.9%) with less computational resource requirements. The number of medium models generated by DeepAssembly is close to AlphaFold-linker (**Fig. 5a**), especially it is superior to AlphaFold-linker on 36% cases. This validated the effectiveness of our method that is based on domain assembly and demonstrated that the deep learning model built by learning the inter-domain interactions in the existing PDB monomer structures can capture the protein-protein interactions as well. In addition, we

Supplementary Table 3. Results of DeepAssembly, AlphaFold-linker and RoseTTAFold on the heterodimers. Acceptable: $0.23 \leq \text{DockQ} < 0.49$, Medium: $0.49 \leq \text{DockQ} < 0.80$, High: $\text{DockQ} \geq 0.80$.

Method	Success rate (SR) (%)	Acceptable Count	Medium Count	High Count
DeepAssembly	32.4	21	48	11
AlphaFold-linker	40.9	31	51	19
RoseTTAFold	18.6	28	17	1

(2) In addition, how about the performance of DeepAssembly on hetero-complexes with more than 2 chains?

Based on your valuable suggestion, we tested the performance of DeepAssembly on hetero-complexes with more than two chains. We applied it to generate models of the heterotrimer Survivin-Borealin-INCENP core complex (PDB ID: 2QFA) and the heterotetramer NuA4 core complex (PDB ID: 5J9T). As illustrated in **Figure R5a**, DeepAssembly generated a highly-quality model with a DockQ score of 0.828 for three-chain hetero-complex 2QFA. In addition, for the four-chain hetero-complex 5J9T, the complex model built by DeepAssembly also achieved a DockQ score of 0.760 (**Figure R5b**), showing the potential that DeepAssembly could also be applied to hetero-complexes with more than two chains.

Figure R5. Structures generated by DeepAssembly for hetero-complexes with more than two chains. The reference PDB structures are colored in gray, and the different chains of the predicted models are colored by blue, green, orange, and purple. **(a)** Survivin-Borealin-INCENP core complex (PDB ID: 2QFA). **(b)** NuA4 core complex (PDB ID: 5J9T).

We have added the corresponding description in the manuscript, and the following are the revisions we have made in the manuscript and Supplementary Information.

We also applied DeepAssembly to generate models of the heterotrimer Survivin-Borealin-INCENP core complex (PDB ID: 2QFA) and the heterotetramer NuA4 core complex (PDB ID: 5J9T). As illustrated in **Supplementary Fig. 7a**, DeepAssembly generated a highly-quality model with a DockQ score of 0.828 for three-chain hetero-complex 2QFA. In addition, for the four-chain hetero-complex 5J9T, the complex model built by DeepAssembly also achieved a DockQ score of 0.760 (**Supplementary Fig. 7b**), showing the potential that DeepAssembly could also be applied to hetero-complexes with more than two chains.

Supplementary Fig. 7 | Structures generated by DeepAssembly for hetero-complexes with more than two chains. The reference PDB structures are colored in gray, and the different chains of the predicted models are colored by blue, green, orange, and purple. **a** Survivin-Borealin-INCENP core complex (PDB ID: 2QFA). **b** NuA4 core complex (PDB ID: 5J9T).

10. It is a bit surprising that DeepAssembly obtains a better average TM-score (0.916) with single-domains predicted by PAMBERT than that with experimental single-domains (0.855). Can the authors explain it?

Response: Thank you for raising this important point. This is because DeepAssembly uses templates with different sequence identity cutoffs for fair comparison with other methods in the two experiments.

(1) On the one hand, in order to thoroughly evaluate DeepAssembly's ability in real-world scenarios, we used single-domains predicted by PAMBERT to assemble multi-domain proteins. In this experiment, we maximized the performance of DeepAssembly, and compared it with the current state-of-art end-to-end method AlphaFold2. In the inference process of AlphaFold2, the templates with a high sequence identity are not excluded. Therefore, for a fair comparison, we also did not exclude templates according to sequence identity in this experiment.

(2) On the other hand, in order to rigorously test the model performance of DeepAssembly, we assembled multi-domain proteins with experimental domains to exclude the influence of single-domains on the final structure. In this experiment, DeepAssembly was compared with SADA, DEMO and AIDA. In SADA and DEMO, structural templates with a sequence identity >30% to the query are excluded [R2, R3], so for a fair comparison, templates with sequence identity >30% of the query are also excluded in DeepAssembly. This appropriately reduces the TM-score of DeepAssembly compared to that in the

previous experiment.

Therefore, DeepAssembly has a better average TM-score with single-domains predicted by PAM-Threader than that with experimental single-domains.

We apologize for not providing detailed experimental settings, and we have added corresponding descriptions in the manuscript.

In order to rigorously test the model performance of DeepAssembly, we assemble multi-domain proteins with experimental domains to exclude the influence of single-domains on the final structure. In this experiment, DeepAssembly is compared with SADA, DEMO and AIDA, and for a fair comparison, we here exclude templates in DeepAssembly with a sequence identity >30%, as done in SADA and DEMO. Fig. 8 presents a summary of the models assembled by DeepAssembly, SADA, DEMO and AIDA. The detailed

11. DeepAssembly is compared with other methods according to the predicted inter-domain distance. Did the authors directly use the distance predicted by the AffineNet or use the structure selected by the evolutionary algorithm? If the latter is used, did the authors use the structure refined by FastRelax?

Response: We apologize for not describing it clearly. In the section “Inter-domain interactions prediction”, we compare DeepAssembly with other methods by directly using the inter-domain distance predicted by the AffineNet. The inter-domain distances of AlphaFold2 and RoseTTAFold are extracted from their output pkl and npz files, respectively. Then we define the inter-domain distance error, err_{dist} , to evaluate the predicted inter-domain distance precision, which is calculated as the errors (Å) between the predicted inter-domain distance and the true inter-domain distance extracted from experimental structure, with smaller value indicating higher predicted distance precision.

$$err_{\text{dist}} = \frac{1}{N_{\text{pair}}} \sum_{(i,j)} |d_{(i,j)}^{\text{pre}} - d_{(i,j)}^{\text{true}}|, \quad \forall (i,j) \in \mathcal{S}_{\text{inter domain}}, \quad i < j$$

where $d_{(i,j)}^{\text{pre}}$ and $d_{(i,j)}^{\text{true}}$ are the predicted distance and the true distance of inter-domain residue pair (i,j), respectively. $\mathcal{S}_{\text{inter domain}}$ represents the set of inter-domain residue pairs, and N_{pair} is the number of inter-domain residue pairs.

We have added the corresponding description in the manuscript and Supplementary Information.

Predicted inter-domain interactions include inter-residue distance between domains. We compare the DeepAssembly with AlphaFold2 and RoseTTAFold according to the predicted inter-domain distances. The inter-domain distance of DeepAssembly is directly from AffineNet, and distances predicted by AlphaFold2 and RoseTTAFold are extracted from their output pkl and npz files, respectively. Here we define the inter-domain distance error to evaluate the predicted distance precision (Supplementary Section 1), which is calculated as the errors (Å) between the predicted and true inter-domain distances. We present in Fig. 9b the comparison of the inter-domain distance errors of DeepAssembly, AlphaFold2 and RoseTTAFold. For

Inter-domain distance error. We define the inter-domain distance error, err_{dist} , to evaluate the predicted inter-domain distance precision, which is calculated as the errors (Å) between the predicted inter-domain distance and the true inter-domain distance extracted from experimental structure, with smaller value indicating higher predicted distance precision.

$$err_{\text{dist}} = \frac{1}{N_{\text{pair}}} \sum_{(i,j)} \left| d_{(i,j)}^{\text{pre}} - d_{(i,j)}^{\text{true}} \right|, \quad \forall (i,j) \in \mathcal{S}_{\text{inter domain}}, \quad i < j \quad (S4)$$

where $d_{(i,j)}^{\text{pre}}$ and $d_{(i,j)}^{\text{true}}$ are the predicted distance and the true distance of inter-domain residue pair (i,j) , respectively. $\mathcal{S}_{\text{inter domain}}$ represents the set of inter-domain residue pairs, and N_{pair} is the number of inter-domain residue pairs.

12. Does the AffineNet use the cross-entropy losses on both intra- and inter-domain? If so, how about the performance of AffineNet in the intra-domain distance and orientation?

Response: As the reviewer pointed out, **the AffineNet uses the cross-entropy losses on both intra- and inter-domain.** Based on your valuable suggestion, we investigated the performance of AffineNet in the intra-domain distance and orientation. We present in **Table R6** the errors of intra- and inter-domain distance and orientation predicted by AffineNet. On average, the intra- and inter-domain distance errors are 0.453 Å and 0.560 Å, respectively. The intra- and inter-domain errors of angles (α , β , γ , θ , ϕ) representing the orientation are 0.306 vs 0.394, 0.180 vs 0.255, 0.261 vs 0.352, 0.148 vs 0.206, 0.204 vs 0.261, respectively. The results show that AffineNet has higher prediction accuracy in the intra-domain than inter-domain, which may be due to the flexibility of inter-domain orientation that makes the inter-domain interaction more difficult to be accurately captured than that of intra-domain. This also reflects that the inter-domain distance and orientation prediction is more challenging than intra-domain.

Table R6. Average errors of intra-/inter-domain distance and orientation predicted by AffineNet. α , β , γ , θ , ϕ are angles representing the inter-residue orientations.

	Distance error (Å)	Orientation error (rad)				
		α	β	γ	θ	ϕ
Inter-domain	0.560	0.394	0.255	0.352	0.206	0.261
Intra-domain	0.453	0.306	0.180	0.261	0.148	0.204

We have added the analysis of this aspect in the manuscript and supplemented the experimental data in the Supplementary Information.

Finally, we investigated the performance of AffineNet in the intra-domain distance and orientation. We present in **Supplementary Table 7** the errors of intra- and inter-domain distance and orientation predicted by AffineNet. The results show that AffineNet has higher prediction accuracy in the intra-domain than inter-domain, which may be due to the flexibility of inter-domain orientation that makes the inter-domain interaction more difficult to be accurately captured than that of intra-domain. This also reflects that the inter-domain distance and orientation prediction is more challenging than intra-domain.

Supplementary Table 7. Average errors of intra-/inter-domain distance and orientation predicted by AffineNet. α , β , γ , θ , ϕ are angles representing the inter-residue orientations.

	Distance error	Orientation error (rad)				
	(Å)	α	β	γ	θ	ϕ
Inter-domain	0.560	0.394	0.255	0.352	0.206	0.261
Intra-domain	0.453	0.306	0.180	0.261	0.148	0.204

Reference:

- [R1] Zhao, K. L., et al. Protein structure and folding pathway prediction based on remote homologs recognition using PAtreader. *Communications Biology* **6**, 243 (2023).
- [R2] Peng, C. X., et al. Structural analogue-based protein structure domain assembly assisted by deep learning. *Bioinformatics* **38**, 4513-4521 (2022).
- [R3] Zhou, X. G., et al. Assembling multidomain protein structures through analogous global structural alignments. *Proc. Natl Acad. Sci. USA* **116**, 15930-15938 (2019).
- [R4] Evans, R., et al. Protein complex prediction with AlphaFold-Multimer. *bioRxiv* (2021).

Response to Reviewer #2

We very much appreciate for your comments and suggestions, which help to significantly improve the quality and description of our manuscript. In the following, we include point-by-point response to the comments, where all changes have been highlighted in the manuscript.

Xia et al presented a new tool to predict Domain-based multi-domain protein and complex structure. It is an interesting work. However, there are already several similar tools were already published before. So, it would be interesting to make comparison between this new tool and others, including the algorithm difference, merits and demerits, applicable system.

Response: We appreciate your positive comments on this work. As the reviewer points out, there are already several similar tools were already published before, such as DEMO and SADA. Based on your valuable suggestion, we compared our tool with others in terms of algorithm difference, merits and demerits, applicable system.

(1) algorithm difference

The previously proposed DEMO and SADA are essentially a template-based method that guides domain assembly by detecting available templates [R1, R2]. DEMO constructs multi-domain protein structures by docking-based domain assembly simulations, in which inter-domain orientations are determined by the distance profiles from analogous templates as detected through domain-level structure alignments [R1]. In SADA, a multi-domain protein structure database is constructed for the full-chain analogue detection using individual domain models. Starting from the initial model constructed from the analogue, the domain assembly simulation is performed to generate the full-chain model through a two-stage differential evolution algorithm [R2]. **Different from DEMO and SADA, this new tool we proposed, DeepAssembly, is a data-driven deep learning approach for multi-domain protein and complex modeling, which does not require the time-consuming structural template alignment process.** We develop a deep learning network model trained on multi-domain proteins to directly predict the inter-domain interactions through the input co-evolutionary, template and inter-domain features, and finally perform the population-based optimization to assemble multi-domain protein or complex driven by an affine transformation-based energy potential transformed from predicted inter-domain interactions.

(2) merits and demerits

merits: (a) firstly, DEMO and SADA, as template-based approaches, both rely on structural template alignment, which is limited by the number of multi-domain proteins in PDB, and the difficulty of capturing the orientation between domains from the template may increase as the number of domains increases. However, **DeepAssembly predicts the inter-domain interactions by using a deep learning model, avoids the time-consuming template alignment process, and does not depend on templates entirely**, because the network model can also capture the accurate inter-domain interactions from co-evolutionary information in the absence of templates. (b) secondly, **the deep learning model in DeepAssembly is trained on multi-domain proteins, which pay more attention to the capture of inter-domain interactions, improving the prediction accuracy of inter-domain orientations.** It may be more urgent and important, given the fact that it is currently relatively easy to obtain high-precision single domain structures. In addition, **the network model integrates co-evolutionary information and template features through the self-attention mechanism, which further improves the prediction**

performance. (c) thirdly, AlphaFold2's inaccurate prediction for multi-domain protein structures is mainly due to the incorrect inter-domain orientation. For cases where the AlphaFold2-predicted structure is not accurate, **DeepAssembly tends to perform more satisfactorily by correcting their inter-domain orientations with large errors, providing an important solution for the update of the low confidence multi-domain structures in AlphaFold database.** (d) finally, **DeepAssembly can also be applied to complex structure prediction, which provides a lightweight way to assemble protein structures by treating domains as assembly units, reducing the requirements for computational resources.** Especially for scenarios without co-evolutionary signals, our method can identify the correct inter-chain interface through the inter-domain interactions learned from the remote templates, it could reduce the dependence on MSA to a certain extent.

demerits: (a) in the absence of sufficient co-evolutionary signals and appropriate templates, our method still has room for improvement in performance. In this case, the introduction of physical and chemical features, e.g., protein-protein interface preferences are also crucial. (b) the domain-domain interaction can be further extended, including the interaction between two domains belonging to different chains.

(3) applicable system

DEMO, SADA and DeepAssembly can all be executed in the 64-bit Linux operating system, and the deep learning model of DeepAssembly is available in Python with Tensorflow1.14.

Thank you again for your valuable suggestion. We have further explained the merits and demerits of our method and the differences from other methods in the manuscript.

hydrophobic interactions. Meanwhile, domain assembly can also be perceived as a docking problem. **The previously proposed DEMO¹ and SADA¹⁵ assemble the single domain structure by rigid body docking, which are essentially a template-based method that guides domain assembly by detecting available templates. However, structural alignment is limited by the number of multi-domain proteins in PDB, and the difficulty of capturing the orientation between domains from the template may increase as the number of domains increases¹⁵. The inter-residue distance predicted by deep learning improves the problem of insufficient number of multi-domain protein templates to a certain extent. But given the fact that it is currently relatively easy to obtain high-precision single domain structures, it may be more urgent and important to pay more attention to the capture of inter-domain interactions in deep learning.**

through inter-domain interactions specifically predicted by deep learning. **Different from DEMO and SADA, as a data-driven deep learning method, DeepAssembly avoids the time-consuming template alignment process and does not depend on templates entirely, which pay more attention to the capture of inter-domain interactions to improve the prediction accuracy of inter-domain orientations.** Experimental results improved in several aspects. **Firstly, in the absence of sufficient co-evolutionary signals and appropriate templates, our method still has room for improvement in performance. In this case, the introduction of physical and chemical features, e.g., protein-protein interface preferences are also crucial. Secondly, the domain-domain interaction can be further extended, including the interaction between two domains belonging to different chains. Studies along these lines are in progress.**

The benchmark is mainly focusing on water soluble proteins. It would be interesting to see how it perform well for membrane proteins as well.

Response: Thank you for your valuable suggestion. It is an interesting exploration. Based on your suggestion, we applied our method to transmembrane proteins.

Figure R1 displays several examples of the transmembrane protein models predicted by DeepAssembly. The first example is Monocarboxylate transporter 1 (MCT1) (PDB ID: 7CKR_A), which is a multi-domain protein that catalyzes the movement of many monocarboxylates across the plasma membrane through an ordered mechanism [R3]. The model built by DeepAssembly had the correct inter-domain orientation, yielded a TM-score of 0.882 (**Figure R1a**). In addition, the target T1024 in CASP14 (PDB ID: 6T1Z_A) is also a monomeric transmembrane protein, which consists of two domains. In this case, DeepAssembly generated a more accurate model with a TM-score of 0.957, compared to AlphaFold2's 0.797 (**Figure R2**).

DeepAssembly can also be applied to transmembrane protein complexes. **Figure R1b** and **c** show the results of DeepAssembly on two homo-oligomeric proteins with C2-symmetry type. DeepAssembly succeeded in building a high-quality model for CDP-alcohol phosphotransferase (PDB ID: 4O6M) (DockQ = 0.881), especially for the transmembrane region, achieving a higher DockQ score of 0.930 (**Figure R1b**). Similarly, DeepAssembly successfully predicted the inter-chain interface for magnesium transporter (PDB ID: 2ZY9) (DockQ = 0.796), and the transmembrane region achieved a DockQ score of 0.855 (**Figure R1c**).

Furthermore, for the acid-sensing ion channel (PDB ID: 2QTS) with three chains, the trimer model predicted by DeepAssembly yielded a DockQ score of 0.402 (**Figure R1d**). However, there is still room for improvement on the transmembrane region. Compared with the extracellular region, the predicted model has deviation with PDB structure in the transmembrane region. This may be due to the highly flexible structures within the transmembrane region, with each subunit in the PDB structure of the chalice-shaped homotrimer having a different orientation between its transmembrane helices and extracellular region, even though they share the same amino acid sequence. **Figure R1e** shows a comparison between the DeepAssembly model and the PDB structure on OmpU, an outer membrane protein of *Vibrio cholerae* (PDB ID: 6EHB). OmpU forms three channels for the diffusional uptake of small molecules, and each consisting of a 16-stranded β barrel [R4]. DeepAssembly correctly predicted the structures of its channels (average TM-score = 0.985), as well as their interfaces with a DockQ score of 0.591.

The last example is a tetrameric potassium channel KcsA (PDB ID: 2QTO), which is an ion channel that is capable of selecting potassium ions. The tetramer model for KcsA built by DeepAssembly achieved a DockQ score of 0.756 (**Figure R1f**). In addition, the DeepAssembly model gave the correct overall shape of the pore used to filter out potassium ions (shown as sphere in **Figure R1f**). These examples demonstrate the potential of DeepAssembly on transmembrane proteins, which may help to understand the function of transmembrane proteins, and provide insights into their molecular interaction mechanism.

Figure R1. Examples of the transmembrane protein structures predicted by DeepAssembly. The reference PDB structures are colored in gray, and the different domains or chains of the predicted models are colored by blue, green, orange, and purple. (a) Monocarboxylate transporter 1 (MCT1) (PDB ID: 7CKR_A). (b) CDP-alcohol phosphotransferase (PDB ID: 4O6M). (c) magnesium transporter (PDB ID: 2ZY9). (d) acid-sensing ion channel (PDB ID: 2QTS). (e) OmpU, an outer membrane protein of *Vibrio cholerae* (PDB ID: 6EHB). (f) potassium channel KcsA (PDB ID: 2QTO). The potassium ion in the ion channel is shown as sphere.

Figure R2. Structures predicted by AlphaFold2 and DeepAssembly for target T1024 from CASP14. The reference PDB structures are colored in gray, and the different domains of the predicted models are colored by blue and green.

We have added this section to the manuscript, and the following are the revisions we have made in the manuscript.

Application to transmembrane protein structure prediction.

Membrane proteins are involved in a variety of essential cellular functions including molecular transporters, ion channels and signal receptors, which constitute about half of current drug targets⁴³. Therefore, the structural determination of membrane proteins is critical to advancing our understanding of their function as well as the drug design. Here, we apply DeepAssembly to transmembrane protein structure prediction.

Fig. 7 displays several examples of the transmembrane protein models predicted by DeepAssembly. The first example is Monocarboxylate transporter 1 (MCT1) (PDB ID: 7CKR_A), which is a multi-domain protein that catalyzes the movement of many monocarboxylates across the plasma membrane through an ordered mechanism⁴⁴. The model built by DeepAssembly had the correct inter-domain orientation, yielded a TM-score of 0.882 (**Fig. 7a**). In addition, the target T1024 in CASP14 (PDB ID: 6T1Z_A) is also a transmembrane protein, which consists of two domains. In this case, DeepAssembly generated a more accurate model with a TM-score of 0.957, compared to AlphaFold2's 0.797 (**Supplementary Fig. 4a**).

DeepAssembly can also be applied to transmembrane protein complexes. **Fig. 7b** and **c** show the results of DeepAssembly on two homo-oligomeric proteins with C2-symmetry type. DeepAssembly succeeded in building a high-quality model for CDP-alcohol phosphotransferase (PDB ID: 4O6M) (DockQ = 0.881), especially for the transmembrane region, achieving a higher DockQ score of 0.930 (**Fig. 7b**). Similarly, DeepAssembly successfully predicted the inter-chain interface for magnesium transporter (PDB ID: 2ZY9) (DockQ = 0.796), and the transmembrane region achieved a DockQ score of 0.855 (**Fig. 7c**).

Furthermore, for the acid-sensing ion channel (PDB ID: 2QTS) with three chains, the trimer model predicted by DeepAssembly yielded a DockQ score of 0.402 (**Fig. 7d**). However, there is still room for improvement on the transmembrane region. Compared with the extracellular region, the predicted model has deviation with PDB structure in the transmembrane region. This may be due to the highly flexible structures within the transmembrane region, with each subunit in the PDB structure of the chalice-shaped homotrimer having a different orientation between its transmembrane helices and extracellular region, even though they share the same amino acid sequence. **Fig. 7e** shows a comparison between the DeepAssembly model and the PDB structure on OmpU, an outer membrane protein of *Vibrio cholerae* (PDB ID: 6EHB). OmpU forms three channels for the diffusional uptake of small molecules, and each consisting of a 16-stranded β barrel⁴⁵. DeepAssembly correctly predicted the structures of its channels (average TM-score = 0.985), as well as their interfaces with a DockQ score of 0.591.

The last example is a tetrameric potassium channel KcsA (PDB ID: 2QTO), which is an ion channel that is capable of selecting potassium ions. The tetramer model for KcsA built by DeepAssembly achieved a DockQ score of 0.756 (**Fig. 7f**). In addition, the DeepAssembly model gave the correct overall shape of

the pore used to filter out potassium ions (shown as sphere in Fig. 7f). These examples demonstrate the potential of DeepAssembly on transmembrane proteins, which may help to understand the function of transmembrane proteins, and provide insights into their molecular interaction mechanism.

Fig. 7 | Examples of the transmembrane protein structures predicted by DeepAssembly. The reference PDB structures are colored in gray, and the different domains or chains of the predicted models are colored by blue, green, orange, and purple. **a** Monocarboxylate transporter 1 (MCT1) (PDB ID: 7CKR_A). **b** CDP-alcohol phosphotransferase (PDB ID: 4O6M). **c** magnesium transporter (PDB ID: 2ZY9). **d** acid-sensing ion channel (PDB ID: 2QTS). **e** OmpU, an outer membrane protein of *Vibrio cholerae* (PDB ID: 6EHB). **f** potassium channel KcsA (PDB ID: 2QTO). The potassium ion in the ion channel is shown as sphere.

Moreover, I don't see too much about the sidechain conformation details. It would be necessary to present this information as well.

Response: Thank you for raising the important question. We apologize for not providing too much details on this. Based on your valuable suggestion, we have added the descriptions about the sidechain conformation in the manuscript.

We assemble the full-length protein model in the domain assembly module. To speed up the assembly, initial full-length models are represented as coarse-grained (centroid) models in Rosetta. The centroid model is a reduced representation of protein structure, in which the backbone remains fully atomic but each side chain is represented by a single artificial atom (centroid). In the iterative annealing stage, series of rotation angle sampling, new structure generation, and structure pool selection steps are repeated iteratively with coarse-grained representation. When the number of iterations reaches the maximum, the

top 10 centroid models (ranked by energy) are selected to be converted to full-atomic models by generating side chains in Rosetta, followed by full-atom refinement by *FastRelax* protocol with ref2015 energy function in Rosetta [R5].

The following are the revisions we have made in the manuscript.

We assemble the full-length protein model in the domain assembly module. The overall architecture consists of three stages: first initial model creation stage, followed by iterative model update stages where a pool of multi-structures is maintained during the population-based rotation angle optimization, and finally full atom-based refinement. At the initialization stage, 1,000 initial full-length models are generated where the rotation angles φ , ψ in the eight linker residues near their domain boundaries are set to random angles (the influence of the number of movable residues near the domain boundaries on the final structure is analyzed in **Supplementary Section 4**). To speed up the assembly, these initial models are represented as coarse-grained (centroid) models in Rosetta. The centroid model is a reduced representation of protein structure, in which the backbone remains fully atomic but each side chain is represented by a single artificial atom (centroid). In the iterative annealing stage, series of rotation angle sampling, new structure generation, and structure pool selection steps are repeated iteratively with coarse-grained representation. This process is repeated for 500 iterations (the accuracy of the final models at different iteration numbers is shown in **Supplementary Fig. 12**), and the objective function is the atomic coordinate deviation potential \mathcal{F}_{ACD} defined above. For the pool that reaches the maximum iteration, the top 10 centroid models (ranked by energy) are selected to be converted to full-atomic models by generating side chains in Rosetta, followed by full-atom refinement by *FastRelax* protocol with ref2015 energy function in Rosetta⁵⁵.

Reference:

- [R1] Zhou, X. G., et al. Assembling multidomain protein structures through analogous global structural alignments. *Proc. Natl Acad. Sci. USA* **116**, 15930-15938 (2019).
- [R2] Peng, C. X., et al. Structural analogue-based protein structure domain assembly assisted by deep learning. *Bioinformatics* **38**, 4513-4521 (2022).
- [R3] Garcia, C. K., et al. Molecular characterization of a membrane transporter for lactate, pyruvate, and other monocarboxylates: Implications for the Cori cycle. *Cell* **76**, 865-873 (1994).
- [R4] Pathania, M., et al. Unusual Constriction Zones in the Major Porins OmpU and OmpT from *Vibrio cholerae*. *Structure* **26**, 708-721 (2018).
- [R5] Rohl, C. A., et al. Protein structure prediction using rosetta. *Methods Enzymol.* **383**, 66-93 (2004).

Response to Reviewer #3

We very much appreciate for your comments and suggestions, which help to significantly improve the quality and description of our manuscript. In the following, we include point-by-point response to the comments, where all changes have been highlighted in the manuscript.

The study presented by Zhang and colleagues seeks to demonstrate that a new algorithm, DeepAssembly, is a significant advance over other multidomain/multiprotein structure prediction algorithms, Alphafold2, AlphafoldMultimer and RosettaFold2, but also previous algorithms from the lap DEMO and SADA.

Response: Thank you for your comments. As the reviewer points out, there are previous algorithms we proposed, such as DEMO and SADA. Here, we explained the merits of our method, and compared with other methods in terms of algorithm difference, merits and demerits, applicable system.

(1) algorithm difference

The previously proposed DEMO and SADA are essentially a template-based method that guides domain assembly by detecting available templates [R1, R2]. DEMO constructs multi-domain protein structures by docking-based domain assembly simulations, in which inter-domain orientations are determined by the distance profiles from analogous templates as detected through domain-level structure alignments [R1]. In SADA, a multi-domain protein structure database is constructed for the full-chain analogue detection using individual domain models. Starting from the initial model constructed from the analogue, the domain assembly simulation is performed to generate the full-chain model through a two-stage differential evolution algorithm [R2]. **Different from DEMO and SADA, this new tool we proposed, DeepAssembly, is a data-driven deep learning approach for multi-domain protein and complex modeling, which does not require the time-consuming structural template alignment process.** We develop a deep learning network model trained on multi-domain proteins to directly predict the inter-domain interactions through the input co-evolutionary, template and inter-domain features, and finally perform the population-based optimization to assemble multi-domain protein or complex driven by an affine transformation-based energy potential transformed from predicted inter-domain interactions.

(2) merits and demerits

merits: (a) firstly, DEMO and SADA, as template-based approaches, both rely on structural template alignment, which is limited by the number of multi-domain proteins in PDB, and the difficulty of capturing the orientation between domains from the template may increase as the number of domains increases. However, **DeepAssembly predicts the inter-domain interactions by using a deep learning model, avoids the time-consuming template alignment process, and does not depend on templates entirely**, because the network model can also capture the accurate inter-domain interactions from co-evolutionary information in the absence of templates. (b) secondly, **the deep learning model in DeepAssembly is trained on multi-domain proteins, which pay more attention to the capture of inter-domain interactions, improving the prediction accuracy of inter-domain orientations.** It may be more urgent and important, given the fact that it is currently relatively easy to obtain high-precision single domain structures. In addition, **the network model integrates co-evolutionary information and template features through the self-attention mechanism, which further improves the prediction performance.** (c) thirdly, AlphaFold2's inaccurate prediction for multi-domain protein structures is

mainly due to the incorrect inter-domain orientation. For cases where the AlphaFold2-predicted structure is not accurate, **DeepAssembly tends to perform more satisfactorily by correcting their inter-domain orientations with large errors, providing an important solution for the update of the low confidence multi-domain structures in AlphaFold database.** (d) finally, **DeepAssembly can also be applied to complex structure prediction, which provides a lightweight way to assemble protein structures by treating domains as assembly units, reducing the requirements for computational resources.** Especially for scenarios without co-evolutionary signals, our method can identify the correct inter-chain interface through the inter-domain interactions learned from the remote templates, it could reduce the dependence on MSA to a certain extent.

demerits: (a) in the absence of sufficient co-evolutionary signals and appropriate templates, our method still has room for improvement in performance. In this case, the introduction of physical and chemical features, e.g., protein-protein interface preferences are also crucial. (b) the domain-domain interaction can be further extended, including the interaction between two domains belonging to different chains.

(3) applicable system

DEMO, SADA and DeepAssembly can all be executed in the 64-bit Linux operating system, and the deep learning model of DeepAssembly is available in Python with Tensorflow1.14.

Despite the limitations pointed out in the concluding summary/perspective, DeepAssembly does a considerably better job than the others at all levels. However, the devil is in the details. Specifically, this reviewer is not convinced that a commonly agreed-on multidomain or complexed protein data base is used. Rather, results are reported by 219 and then 165 proteins but is is not clear how these are selected. It would be more convincing to apply DeepAssembly to Casp14 and 15 targets first.

Response: Thank you for pointing out this important issue. In this work, we trained the deep learning model on 10,593 multi-domain proteins that were collected from our previously developed multi-domain protein structure database (MPDB) (<http://zhanglab-bioinf.com/SADA/>) (until September, 2021, with 48,225 entries). We apologize for the unclear description and have further explained how the test proteins are selected. In addition, we tested the performance of DeepAssembly on the CASP14 and CASP15 targets based on your valuable suggestion.

(1) Test protein collection

In order to evaluate the performance of DeepAssembly on multi-domain proteins, we constructed two multi-domain protein test sets, the detailed process is as follows:

(a) Benchmark set of 219 multi-domain proteins

Firstly, we use multi-domain proteins with known structures from the comprehensive dataset in previous study DEMO [R1] to test the developed pipeline. These proteins share <30% sequence identity, are collected by separately clustering the proteins with different domain types and structures from the template library [R1]. Here, all the proteins in the initial DEMO dataset that have >30% sequence identity with any protein in the training set and validation set are excluded, resulting in a total of 219 structures as the final test set.

(b) Test set of 164 proteins from AlphaFold DB

Secondly, to evaluate the performance of DeepAssembly in improving the inter-domain orientation, we further construct an independent test set of 164 multi-domain proteins. All entries come from 23,391 *H. sapiens* protein structures deposited in AlphaFold database (<https://alphafold.ebi.ac.uk/>), the PDB structure for each entry is directly downloaded from the PDB at <https://www.rcsb.org/>. Of these 23,391 proteins, the protein that has any of the following features is removed: (i) without experimental structure in the PDB; (ii) is confirmed as containing only one domain by DomainParser or corresponding entries in CATH and SCOPe database; (iii) having redundancy between each other for the UniProt primary accession; (iv) with a TM-score > 0.8 for the corresponding structure in AlphaFold database; (v) with less than 85% of solved residues for the experimental structure in the PDB, leading to a final set of 164 multi-domain proteins.

The following are the revisions we have made in the manuscript.

Development set. We collected a multi-domain protein dataset from our previously developed multi-domain protein structure database (MPDB) (<http://zhanglab-bioinf.com/SADA/>) (until September, 2021, with 48,225 entries) to develop the pipeline. This dataset contains a total of 10,593 multi-domain proteins, with each protein chain having 40 to 1,000 residues, sharing <40% sequence identity and having a resolution within 3.0 Å. These multi-domain proteins contain between 2-9 domains, determined by

Test set. We use multi-domain proteins with known structures from the comprehensive dataset in previous study DEMO¹ to test the developed pipeline. These proteins share <30% sequence identity, are collected by separately clustering the proteins with different domain types and structures from the template library¹. Here, all the proteins in the initial DEMO dataset that have >30% sequence identity with any protein in the training set and validation set are excluded, resulting in a total of 219 structures as the final test set.

To evaluate the performance of DeepAssembly in improving the inter-domain orientation, we further construct an independent test set of 164 multi-domain proteins. All entries come from 23,391 *H. sapiens* protein structures deposited in AlphaFold database (<https://alphafold.ebi.ac.uk/>), the PDB structure for each entry is directly downloaded from the PDB at <https://www.rcsb.org/>. Of these 23,391 proteins, the protein that has any of the following features is removed: (i) without experimental structure in the PDB; (ii) is confirmed as containing only one domain by DomainParser or corresponding entries in CATH and SCOPe database; (iii) having redundancy between each other for the UniProt primary accession; (iv) with a TM-score > 0.8 for the corresponding structure in AlphaFold database; (v) with less than 85% of solved residues for the experimental structure in the PDB, leading to a final set of 164 multi-domain proteins.

(2) Performance on the CASP14 and 15 targets

Based on your valuable suggestion, we applied DeepAssembly to CASP14 and CASP15 targets to further evaluate its performance. We selected a total of 30 multi-domain targets from CASP14 and 15 as test proteins according to the “Domain Definition” on the CASP official website, of which 17 are from CASP14 and 13 are from CASP15. These targets all have known PDB structures that been released. We assembled full-length models by DeepAssembly on these CASP targets with single-domains generated by PAlhreader, and compared with AlphaFold2. The results of AlphaFold2 were obtained by running the

standalone package locally with the default settings.

Table R1 summarizes the results of targets from CASP14 and CASP15 predicted by AlphaFold2 and DeepAssembly. On the CASP14 targets, the average TM-score of DeepAssembly is 0.850, which is higher than 0.832 for AlphaFold2. DeepAssembly also achieved a better performance than AlphaFold2 on most of the test targets (**Figure R1a**). Especially for the target T1024, a membrane protein that consists of two domains, DeepAssembly correctly predicted its inter-domain orientation with a full-length TM-score of 0.957, which is a significant improvement over 0.797 by AlphaFold2 (**Figure R2a**). For the targets from CASP15, the models predicted by DeepAssembly achieved an average TM-score of 0.584, which is slightly higher than 0.567 for AlphaFold2. Specifically, DeepAssembly obtained models with a higher TM-score than AlphaFold2 on 62% of the targets (**Figure R1b**). **Figure R2b** shows an example of target T1121, that DeepAssembly predicted a more accurate inter-domain orientation than AlphaFold2. For the target T1121, AlphaFold2 accurately predicted its two individual domains (T1121-D1 and T1121-D2) with TM-scores of 0.944 and 0.953, respectively. However, the full-length models directly predicted by AlphaFold2 obtained a significantly lower TM-score (0.743) due to the incorrect inter-domain orientation. In this case, DeepAssembly generated a highly accurate model with a TM-score of 0.925 (**Figure R2b**), demonstrating its ability to improve the accuracy of multi-domain protein structure prediction by capturing the correct inter-domain orientation.

Furthermore, in order to analyze the effect of the single-domain structures used on the final full-length model accuracy, we further assembled the model through DeepAssembly with the experimental domain structures. The results show that the average TM-score on the CASP14 and CASP15 targets by using the experimental domain structures achieved 0.903 and 0.778, respectively, which is a significant improvement compared to using the predicted single-domains. The example, T1137s1, is a challenging target as its second domain (T1137s1-D2) is relatively difficult to predict. As illustrated in **Figure R2c**, the TM-score of AlphaFold2 model is 0.446. In this target, the two predicted single-domain structures used by DeepAssembly have TM-scores of 0.870 and 0.503, respectively, where the lower accuracy of the second domain affects the quality of full-length model, resulting in a TM-score of 0.474. However, by using experimental single-domain structures, DeepAssembly generated a highly accurate model with a TM-score of 0.949 (**Figure R2c**). This indicates that predicted accuracy of the assembled full-length model could be further improved if the high-quality single-domain models are obtained. Meanwhile, the quality of single-domain model is also an important factor in the accurate multi-domain protein structure prediction in addition to inter-domain orientation.

Table R1. Results of AlphaFold2 and DeepAssembly on CASP14 and CASP15 targets.

Method	CASP14		CASP15	
	RMSD (Å)	TM-score	RMSD (Å)	TM-score
AlphaFold2	7.45	0.832	20.58	0.567
DeepAssembly	6.68	0.850	15.94	0.584

Figure R1. Head-to-head TM-score comparison of DeepAssembly with AlphaFold2. (a) Head-to-head comparison on each CASP14 target. (b) Head-to-head comparison on each CASP15 target.

Figure R2. Examples of the CASP14 and 15 targets predicted by AlphaFold2 and DeepAssembly. The reference PDB structures are colored in gray, and the different domains of the predicted models are colored by blue and green. (a) T1024 (PDB ID: 6T1Z). (b) T1121 (PDB ID: 7TIL). (c) T1137s1 (PDB ID: 8FEF).

We have added this section to the manuscript, and the following are the revisions we have made in the manuscript.

Performance on the CASP14 and 15 targets.

we apply DeepAssembly to CASP14 and CASP15 targets to further evaluate its performance. We selected a total of 30 multi-domain targets from CASP14 and 15 as test proteins according to the “Domain Definition” on the CASP official website, of which 17 are from CASP14 and 13 are from CASP15. These targets all have known PDB structures that been released. We assembled full-length models by DeepAssembly on these CASP targets with single-domains generated by PPathreader, and compared with AlphaFold2. The results of AlphaFold2 were obtained by running the standalone package locally with the default settings.

Supplementary Table 2 summarizes the results of targets from CASP14 and CASP15 predicted by AlphaFold2 and DeepAssembly. On the CASP14 targets, the average TM-score of DeepAssembly is 0.850, which is higher than 0.832 for AlphaFold2. DeepAssembly also achieved a better performance than AlphaFold2 on most of the test targets (**Supplementary Fig. 3a**). Especially for the target T1024, a membrane protein that consists of two domains, DeepAssembly correctly predicted its inter-domain orientation with a full-length TM-score of 0.957, which is a significant improvement over 0.797 by AlphaFold2 (**Supplementary Fig. 4a**). For the targets from CASP15, the models predicted by DeepAssembly achieved an average TM-score of 0.584, which is slightly higher than 0.567 for AlphaFold2. Specifically, DeepAssembly obtained models with a higher TM-score than AlphaFold2 on 62% of the targets (**Supplementary Fig. 3b**). **Supplementary Fig. 4b** shows an example of target T1121. AlphaFold2 accurately predicted its two individual domains (T1121-D1 and T1121-D2) with TM-scores of 0.944 and 0.953, respectively. However, the full-length models directly predicted by AlphaFold2 obtained a significantly lower TM-score (0.743). In this case, DeepAssembly generated a highly accurate model with a TM-score of 0.925, demonstrating its ability to improve the accuracy of multi-domain protein structure prediction by capturing the correct inter-domain orientation.

Furthermore, in order to analyze the effect of the single-domain structures used on the final full-length model accuracy, we further assembled the model through DeepAssembly with the experimental domain structures. The results show that the average TM-score on the CASP14 and CASP15 targets by using the experimental domain structures achieved 0.903 and 0.778, respectively, which is a significant improvement compared to using the predicted single-domains. The example, T1137s1, is a challenging target as its second domain (T1137s1-D2) is relatively difficult to predict. As illustrated in **Supplementary Fig. 4c**, the TM-score of AlphaFold2 model is 0.446. In this target, the two predicted single-domain structures used by DeepAssembly have TM-scores of 0.870 and 0.503, respectively, where the lower accuracy of the second domain affects the quality of full-length model, resulting in a TM-score of 0.474.

However, by using experimental single-domain structures, DeepAssembly generated a highly accurate model with a TM-score of 0.949 (Supplementary Fig. 4c). This indicates that predicted accuracy of the assembled full-length model could be further improved if the high-quality single-domain models are obtained. Meanwhile, the quality of single-domain model is also an important factor in the accurate multi-domain protein structure prediction in addition to inter-domain orientation.

Supplementary Table 2. Results of AlphaFold2 and DeepAssembly on CASP14 and CASP15 targets.

Method	CASP14		CASP15	
	RMSD (Å)	TM-score	RMSD (Å)	TM-score
AlphaFold2	7.45	0.832	20.58	0.567
DeepAssembly	6.68	0.850	15.94	0.584

Supplementary Fig. 3 | Head-to-head TM-score comparison of DeepAssembly with AlphaFold2. a Head-to-head comparison on each CASP14 target. **b** Head-to-head comparison on each CASP15 target.

Supplementary Fig. 4 | Examples of the CASP14 and 15 targets predicted by AlphaFold2 and DeepAssembly. The reference PDB structures are colored in gray, and the different domains of the predicted models are colored by blue and green. **a** T1024 (PDB ID: 6T1Z). **b** T1121 (PDB ID: 7TIL). **c** T1137s1 (PDB ID: 8FEF).

For a more general and biological audience of communications biology, the interest in this topic would be marginal in absence of a clear prediction how cell signaling or transmembrane proteins (or their mechanism of activation) can be predicted. The paper seems more suitable for an expert journal whose readers appreciate the details of the algorithms and findings. As presented at the moment, there is too little information to introduce readers to some of the parameters, e.g. TMscore, DockQ score and cut-offs etc. The figures need a lot more explanation, e.g. what does n refer to in Fig.2.

Response: Thank you for your valuable suggestion. From the perspective of methodology, our method is important for revealing new biological mechanisms and understanding biological functions, especially for the application scenarios, e.g., cell signaling and transmembrane proteins mentioned by the reviewer. According to the valuable suggestion of the reviewer, we further applied our method to transmembrane proteins. The results show the potential of DeepAssembly on transmembrane proteins, which may help to understand their function, and provide insights into their molecular interaction mechanism. In addition, we apologize for unclear parts of the manuscript. We have supplemented some necessary information in the manuscript and Supplementary Information, e.g., parameter description, some details about the method, and further explanations to figures and tables.

Figure R3 displays several examples of the transmembrane protein models predicted by DeepAssembly. The first example is Monocarboxylate transporter 1 (MCT1) (PDB ID: 7CKR_A), which is a multi-domain protein that catalyzes the movement of many monocarboxylates across the plasma membrane through an ordered mechanism [R3]. The model built by DeepAssembly had the correct inter-domain orientation, yielded a TM-score of 0.882 (**Figure R3a**). In addition, the target T1024 in CASP14 (PDB ID: 6T1Z_A) is also a monomeric transmembrane protein, which consists of two domains. In this case, DeepAssembly generated a more accurate model with a TM-score of 0.957, compared to AlphaFold2's 0.797 (**Figure R4**).

DeepAssembly can also be applied to transmembrane protein complexes. **Figure R3b** and **c** show the results of DeepAssembly on two homo-oligomeric proteins with C2-symmetry type. DeepAssembly succeeded in building a high-quality model for CDP-alcohol phosphotransferase (PDB ID: 4O6M) (DockQ = 0.881), especially for the transmembrane region, achieving a higher DockQ score of 0.930 (**Figure R3b**). Similarly, DeepAssembly successfully predicted the inter-chain interface for magnesium transporter (PDB ID: 2ZY9) (DockQ = 0.796), and the transmembrane region achieved a DockQ score of 0.855 (**Figure R3c**).

Furthermore, for the acid-sensing ion channel (PDB ID: 2QTS) with three chains, the trimer model predicted by DeepAssembly yielded a DockQ score of 0.402 (**Figure R3d**). However, there is still room for improvement on the transmembrane region. Compared with the extracellular region, the predicted model has deviation with PDB structure in the transmembrane region. This may be due to the highly flexible structures within the transmembrane region, with each subunit in the PDB structure of the chalice-shaped homotrimer having a different orientation between its transmembrane helices and extracellular region, even though they share the same amino acid sequence. **Figure R3e** shows a comparison between the DeepAssembly model and the PDB structure on OmpU, an outer membrane protein of *Vibrio cholerae* (PDB ID: 6EHB). OmpU forms three channels for the diffusional uptake of small molecules, and each consisting of a 16-stranded β barrel [R4]. DeepAssembly correctly predicted the structures of its channels (average TM-score = 0.985), as well as their interfaces with a DockQ score of 0.591.

The last example is a tetrameric potassium channel KcsA (PDB ID: 2QTO), which is an ion channel that is capable of selecting potassium ions. The tetramer model for KcsA built by DeepAssembly achieved a DockQ score of 0.756 (**Figure R3f**). In addition, the DeepAssembly model gave the correct overall shape of the pore used to filter out potassium ions (shown as sphere in **Figure R3f**). These examples demonstrate the potential of DeepAssembly on transmembrane proteins, which may help to understand the function of transmembrane proteins, and provide insights into their molecular interaction mechanism.

Figure R3. Examples of the transmembrane protein structures predicted by DeepAssembly. The reference PDB structures are colored in gray, and the different domains or chains of the predicted models are colored by blue, green, orange, and purple. (a) Monocarboxylate transporter 1 (MCT1) (PDB ID: 7CKR_A). (b) CDP-alcohol phosphotransferase (PDB ID: 4O6M). (c) magnesium transporter (PDB ID: 2ZY9). (d) acid-sensing ion channel (PDB ID: 2QTS). (e) OmpU, an outer membrane protein of *Vibrio cholerae* (PDB ID: 6EHB). (f) potassium channel KcsA (PDB ID: 2QTO). The potassium ion in the ion channel is shown as sphere.

Figure R4. Structures predicted by AlphaFold2 and DeepAssembly for target T1024 from CASP14. The reference PDB structures are colored in gray, and the different domains of the predicted models are colored by blue and green.

We have added this section to the manuscript, and the following are the revisions we have made in the manuscript.

Application to transmembrane protein structure prediction.

Membrane proteins are involved in a variety of essential cellular functions including molecular transporters, ion channels and signal receptors, which constitute about half of current drug targets⁴³. Therefore, the structural determination of membrane proteins is critical to advancing our understanding of their function as well as the drug design. Here, we apply DeepAssembly to transmembrane protein structure prediction.

Fig. 7 displays several examples of the transmembrane protein models predicted by DeepAssembly. The first example is Monocarboxylate transporter 1 (MCT1) (PDB ID: 7CKR_A), which is a multi-domain protein that catalyzes the movement of many monocarboxylates across the plasma membrane through an ordered mechanism⁴⁴. The model built by DeepAssembly had the correct inter-domain orientation, yielded a TM-score of 0.882 (**Fig. 7a**). In addition, the target T1024 in CASP14 (PDB ID: 6T1Z_A) is also a transmembrane protein, which consists of two domains. In this case, DeepAssembly generated a more accurate model with a TM-score of 0.957, compared to AlphaFold2's 0.797 (**Supplementary Fig. 4a**).

DeepAssembly can also be applied to transmembrane protein complexes. **Fig. 7b** and **c** show the results of DeepAssembly on two homo-oligomeric proteins with C2-symmetry type. DeepAssembly succeeded in building a high-quality model for CDP-alcohol phosphotransferase (PDB ID: 4O6M) (DockQ = 0.881), especially for the transmembrane region, achieving a higher DockQ score of 0.930 (**Fig. 7b**). Similarly, DeepAssembly successfully predicted the inter-chain interface for magnesium transporter (PDB ID: 2ZY9) (DockQ = 0.796), and the transmembrane region achieved a DockQ score of 0.855 (**Fig. 7c**).

Furthermore, for the acid-sensing ion channel (PDB ID: 2QTS) with three chains, the trimer model predicted by DeepAssembly yielded a DockQ score of 0.402 (**Fig. 7d**). However, there is still room for improvement on the transmembrane region. Compared with the extracellular region, the predicted model has deviation with PDB structure in the transmembrane region. This may be due to the highly flexible structures within the transmembrane region, with each subunit in the PDB structure of the chalice-shaped homotrimer having a different orientation between its transmembrane helices and extracellular region, even though they share the same amino acid sequence. **Fig. 7e** shows a comparison between the DeepAssembly model and the PDB structure on OmpU, an outer membrane protein of *Vibrio cholerae* (PDB ID: 6EHB). OmpU forms three channels for the diffusional uptake of small molecules, and each consisting of a 16-stranded β barrel⁴⁵. DeepAssembly correctly predicted the structures of its channels (average TM-score = 0.985), as well as their interfaces with a DockQ score of 0.591.

The last example is a tetrameric potassium channel KcsA (PDB ID: 2QTO), which is an ion channel that is capable of selecting potassium ions. The tetramer model for KcsA built by DeepAssembly achieved a DockQ score of 0.756 (**Fig. 7f**). In addition, the DeepAssembly model gave the correct overall shape of

the pore used to filter out potassium ions (shown as sphere in Fig. 7f). These examples demonstrate the potential of DeepAssembly on transmembrane proteins, which may help to understand the function of transmembrane proteins, and provide insights into their molecular interaction mechanism.

Fig. 7 | Examples of the transmembrane protein structures predicted by DeepAssembly. The reference PDB structures are colored in gray, and the different domains or chains of the predicted models are colored by blue, green, orange, and purple. **a** Monocarboxylate transporter 1 (MCT1) (PDB ID: 7CKR_A). **b** CDP-alcohol phosphotransferase (PDB ID: 4O6M). **c** magnesium transporter (PDB ID: 2ZY9). **d** acid-sensing ion channel (PDB ID: 2QTS). **e** OmpU, an outer membrane protein of *Vibrio cholerae* (PDB ID: 6EHB). **f** potassium channel KcsA (PDB ID: 2QTO). The potassium ion in the ion channel is shown as sphere.

In addition, we have supplemented some necessary information in the manuscript and Supplementary Information, e.g., parameter description, some details about the method, and further explanations to figures and tables.

(1) Parameter description

In this work, we used RMSD, TM-score and DockQ score to evaluate the quality of the multi-domain protein and complex model built by DeepAssembly. We also reported the inter-domain distance error to evaluate the predicted inter-domain distance precision. The detailed descriptions of these metrics follow:

(a) RMSD

Root-mean-square deviation (RMSD) ($= \sqrt{\frac{1}{N} \sum_{i=1}^N d_i^2}$) is calculated as an average of distance error (d_i)

with equal weight over all residue pairs. The lower value indicates closer structural similarity.

(b) TM-score

Template modeling score (TM-score) [R5] is a metric for evaluating the topological similarity between protein structures, which can be calculated by

$$\text{TM-score} = \max \left[\frac{1}{N_{\text{res}}} \sum_{i=1}^{N_{\text{aligned}}} \frac{1}{1 + \left(\frac{d_i}{d_0(N_{\text{res}})} \right)^2} \right]$$

where N_{res} is the amino acid sequence length of the target protein, N_{aligned} is the length of the aligned residues to the reference (native) structure, d_i is the distance between the i -th pair of aligned residues, $d_0(N_{\text{res}}) = 1.24\sqrt[3]{N_{\text{res}} - 15} - 1.8$ is a scale to normalize the match difference, and ‘max’ refers to the optimized value selected from various rotation and translation matrices for structure superposition. The value of TM-score ranges in (0,1], where a higher value indicates closer structural similarity. Stringent statistics showed that TM-score > 0.5 corresponds to a similarity with two structures having the same fold and/or domain orientations [R6].

(c) DockQ score

DockQ [R7] is a score in the range [0,1] that can be used to measure the quality of the interface. Interface with score greater than 0.23 is considered as successfully predicting the interface, greater than 0.49 and less than 0.8 is considered as medium quality, and greater than 0.8 is considered as high quality. DockQ score is calculated as:

$$\text{DockQ}(F_{\text{nat}}, \text{LRMS}, \text{iRMS}, d_1, d_2) = (F_{\text{nat}} + \text{RMS}_{\text{scaled}}(\text{LRMS}, d_1) + \text{RMS}_{\text{scaled}}(\text{iRMS}, d_2))/3$$

where F_{nat} is the fraction of native interfacial contacts preserved in the interface of the predicted complex. LRMS is the Ligand Root Mean Square deviation calculated for the backbone of the shorter chain (ligand) of the model after superposition of the longer chain (receptor) [R8]. iRMS, the receptor-ligand interface in the target (native) is redefined at a relatively relaxed atomic contact cutoff of 10Å which is twice the value used to define inter-residue ‘interface’ contacts in case of F_{nat} . The backbone atoms of these ‘interface’ residues are then superposed on their equivalents in the predicted complex (model) to compute the iRMS [R8]. $\text{RMS}_{\text{scaled}}(\text{RMS}, d_i)$ is defined as:

$$\text{RMS}_{\text{scaled}}(\text{RMS}, d_i) = \frac{1}{1 + \left(\frac{\text{RMS}}{d_i} \right)^2}$$

which represents the scaled RMS deviations corresponding to any of the two terms, LRMS or iRMS (RMS) and d_i is a scaling factor, d_1 for LRMS and d_2 for iRMS, optimized to $d_1 = 8.5\text{Å}$ and $d_2 = 1.5\text{Å}$.

(d) Inter-domain distance error

We define the inter-domain distance error, err_{dist} , to evaluate the predicted inter-domain distance precision, which is calculated as the errors (Å) between the predicted inter-domain distance and the true inter-domain distance extracted from experimental structure, with smaller value indicating higher predicted distance precision.

$$err_{\text{dist}} = \frac{1}{N_{\text{pair}}} \sum_{(i,j)} |d_{(i,j)}^{\text{pre}} - d_{(i,j)}^{\text{true}}|, \quad \forall (i,j) \in \mathcal{S}_{\text{inter domain}}, \quad i < j$$

where $d_{(i,j)}^{\text{pre}}$ and $d_{(i,j)}^{\text{true}}$ are the predicted distance and the true distance of inter-domain residue pair (i,j) , respectively. $\mathcal{S}_{\text{inter domain}}$ represents the set of inter-domain residue pairs, and N_{pair} is the number of inter-domain residue pairs.

We have supplemented the detailed descriptions of above metrics in the manuscript and Supplementary Information.

(PThreader) from the segmented sequences, as depicted in the DeepAssembly protocol. We use root-mean-square deviation (RMSD) and template modeling score³⁴ (TM-score) to evaluate the accuracy of the built models. Here, the TM-score is a metric defined to evaluate the topological similarity between protein structures (Supplementary Section 1), taking values (0,1], where a higher value indicates closer structural similarity. Fig. 2a shows the full-chain TM-score and RMSD of the built multi-domain protein assembling multi-domain proteins. In addition, the built models are evaluated using DockQ score³⁹. DockQ measures the interface quality (Supplementary Section 1), interfaces with score >0.23 are considered correct. The success rate (SR) represents the percentage of cases whose DockQ are greater than 0.23.

Predicted inter-domain interactions include inter-residue distance between domains. We compare the DeepAssembly with AlphaFold2 and RoseTTAFold according to the predicted inter-domain distances. The inter-domain distance of DeepAssembly is directly from AffineNet, and distances predicted by AlphaFold2 and RoseTTAFold are extracted from their output pkl and npz files, respectively. Here we define the inter-domain distance error to evaluate the predicted distance precision (Supplementary Section 1), which is calculated as the errors (Å) between the predicted and true inter-domain distances. We present in Fig. 9b

Evaluation metrics.

We use three types of metrics to evaluate the quality of the multi-domain protein and complex model built by DeepAssembly, and the predicted inter-domain distance precision. The first type of metrics is one that measure the closeness between the built model and the experimental structure in PDB. In this respect, we adopt the RMSD and TM-score³⁴ between the predicted multi-domain protein model and the corresponding experimental structure calculated by TM-score tool downloaded at <https://zhanggroup.org/TM-score/>. The second type is one that measures the interface quality of predicted protein complex model. In this regard, we report DockQ score³⁹ calculated by public tool (<https://github.com/bjornwallner/DockQ/>). Besides the above metrics, we also reported the inter-domain distance error to evaluate the predicted inter-domain distance precision. The detailed descriptions of above metrics are given in Supplementary Section 1.

Supplementary Section 1. Description of evaluation metrics

RMSD. RMSD ($= \sqrt{\frac{1}{N} \sum_{i=1}^N d_i^2}$) is calculated as an average of distance error (d_i) with equal weight over all residue pairs. The lower value indicates closer structural similarity.

TM-score. TM-score¹ is a metric for evaluating the topological similarity between protein structures, which can be calculated by

$$\text{TM-score} = \max \left[\frac{1}{N_{\text{res}}} \sum_{i=1}^{N_{\text{aligned}}} \frac{1}{1 + \left(\frac{d_i}{d_0(N_{\text{res}})} \right)^2} \right] \quad (\text{S1})$$

where N_{res} is the amino acid sequence length of the target protein, N_{aligned} is the length of the aligned residues to the reference (native) structure, d_i is the distance between the i -th pair of aligned residues, $d_0(N_{\text{res}}) = 1.24^3 \sqrt{N_{\text{res}}} - 15 - 1.8$ is a scale to normalize the match difference, and 'max' refers to the optimized value selected from various rotation and translation matrices for structure superposition. The value of TM-score ranges in (0, 1], where a higher value indicates closer structural similarity. Stringent statistics showed that TM-score > 0.5 corresponds to a similarity with two structures having the same fold and/or domain orientations².

DockQ. DockQ³ is a score in the range [0, 1] that can be used to measure the quality of the interface. Interface with score greater than 0.23 is considered as successfully predicting the interface, greater than 0.49 and less than 0.8 is considered as medium quality, and greater than 0.8 is considered as high quality. DockQ score is calculated as:

$$\text{DockQ}(F_{\text{nat}}, \text{LRMS}, \text{iRMS}, d_1, d_2) = (F_{\text{nat}} + \text{RMS}_{\text{scaled}}(\text{LRMS}, d_1) + \text{RMS}_{\text{scaled}}(\text{iRMS}, d_2))/3 \quad (\text{S2})$$

where F_{nat} is the fraction of native interfacial contacts preserved in the interface of the predicted complex. LRMS is the Ligand Root Mean Square deviation calculated for the backbone of the shorter chain (ligand) of the model after superposition of the longer chain (receptor)⁴. iRMS, the receptor-ligand interface in the target (native) is redefined at a relatively relaxed atomic contact cutoff of 10Å which is twice the value used to define inter-residue 'interface' contacts in case of F_{nat} . The backbone atoms of these 'interface' residues are then superposed on their equivalents in the predicted complex (model) to compute the iRMS⁴. $\text{RMS}_{\text{scaled}}(\text{RMS}, d_i)$ is defined as:

$$\text{RMS}_{\text{scaled}}(\text{RMS}, d_i) = \frac{1}{1 + \left(\frac{\text{RMS}}{d_i} \right)^2} \quad (\text{S3})$$

which represents the scaled RMS deviations corresponding to any of the two terms, LRMS or iRMS (RMS) and d_i is a scaling factor, d_1 for LRMS and d_2 for iRMS, optimized to $d_1 = 8.5\text{Å}$ and $d_2 = 1.5\text{Å}$.

Inter-domain distance error. We define the inter-domain distance error, err_{dist} , to evaluate the predicted inter-domain distance precision, which is calculated as the errors (Å) between the predicted inter-domain distance and the true inter-domain distance extracted from experimental structure, with smaller value indicating higher predicted distance precision.

$$\text{err}_{\text{dist}} = \frac{1}{N_{\text{pair}}} \sum_{(i,j)} \left| d_{(i,j)}^{\text{pre}} - d_{(i,j)}^{\text{true}} \right|, \quad \forall (i,j) \in S_{\text{inter domain}}, \quad i < j \quad (\text{S4})$$

where $d_{(i,j)}^{\text{pre}}$ and $d_{(i,j)}^{\text{true}}$ are the predicted distance and the true distance of inter-domain residue pair (i,j) , respectively. $S_{\text{inter domain}}$ represents the set of inter-domain residue pairs, and N_{pair} is the number of inter-domain residue pairs.

(2) Explanations to figures and tables

(a) In Fig. 2b, the "n" refers to the number of points on either side of the diagonal, which reflects the performance of the two methods in terms of generating more models with higher accuracy. For example, the "n = 144" represents that DeepAssembly achieves a higher TM-score than AlphaFold2 on 144 test

cases. Similarly, the “n” in Fig. 4c also represents the number of points on either side of the diagonal.

We have supplemented the corresponding description in the figures.

Fig. 2 | Results of assembling predicted domain structures. **a** Average TM-score over 219 multi-domain proteins for DeepAssembly and AlphaFold2. The y-axis on the right represents the full-chain RMSD of proteins. **b** Head-to-head comparison of the TM-score and RMSD values on each test case between DeepAssembly and AlphaFold2. “n” refers to the number of points on either side of the diagonal. **c** Histogram of backbone RMSD for full-chain models by DeepAssembly and AlphaFold2. **d** TM-score comparisons between AlphaFold2 full-length model and its single domain parts (AF2 domain) on the corresponding proteins of the models predicted by AlphaFold2 with TM-score less than each cutoff. **e** TM-score comparisons between DeepAssembly and AlphaFold2 on the corresponding proteins of the models predicted by AlphaFold2 with TM-score less than each cutoff. “cutoff” refers to a TM-score value on the x-axis, which reflects a TM-score interval less than this value.

Fig. 4 | DeepAssembly corrects the inter-domain orientation of multi-domain protein structures in AlphaFold database. **a** Average TM-score over 164 multi-domain proteins for DeepAssembly and AlphaFold2. The y-axis on the right represents the percentage of targets with TM-score>0.7. **b** Distribution of the full-chain TM-score for the predicted models. **c** Head-to-head comparison of the TM-score value on each test case between DeepAssembly and AlphaFold2. “n” refers to the number of points on either side of the diagonal. **d, f,**

(b) In Fig. 2, the “cutoff” refers to a TM-score value on the x-axis, which reflects a TM-score interval, for example, “cutoff = 0.8”, representing a TM-score interval from 0 to 0.8. Similarly, the “cutoff” in Fig. 5 refers to a DockQ score value, which reflects a DockQ score interval from this value to 1. The “cutoff” in Fig. 8 refers to a TM-score value, which reflects a TM-score interval from this value to 1.

We have supplemented the corresponding description in the figures.

Fig. 2 | Results of assembling predicted domain structures. **a** Average TM-score over 219 multi-domain proteins for DeepAssembly and AlphaFold2. The y-axis on the right represents the full-chain RMSD of proteins. **b** Head-to-head comparison of the TM-score and RMSD values on each test case between DeepAssembly and AlphaFold2. “n” refers to the number of points on either side of the diagonal. **c** Histogram of backbone RMSD for full-chain models by DeepAssembly and AlphaFold2. **d** TM-score comparisons between AlphaFold2 full-length model and its single domain parts (AF2 domain) on the corresponding proteins of the models predicted by AlphaFold2 with TM-score less than each cutoff. **e** TM-score comparisons between DeepAssembly and AlphaFold2 on the corresponding proteins of the models predicted by AlphaFold2 with TM-score less than each cutoff. “cutoff” refers to a TM-score value on the x-axis, which reflects a TM-score interval less than this value.

(c) In Supplementary Table 1, “#TM-score>0.9” represents the number of models with TM-score greater than 0.9.

We have supplemented the corresponding description in the tables.

Supplementary Table 1. Summary of the performance of DeepAssembly and AlphaFold2 for predicting multi-domain proteins. #TM-score>0.9 represents the number of models with TM-score > 0.9.

Method	RMSD (Å)	TM-score	#TM-score>0.9
DeepAssembly	2.91	0.922	178
DeepAssembly (AF2 domain)	3.11	0.919	176
AlphaFold2	3.58	0.900	166

(d) In Supplementary Tables 4, 5 and 6, “2dom”, “3dom”, and “m4dom” represent the classification of proteins with two, three, and more than four domains, respectively.

We have supplemented the corresponding description in the tables.

Supplementary Table 4. TM-score of the models assembled by DeepAssembly, SADA, DEMO and AIDA using experimental domain structures. "2dom", "3dom", and "m4dom" represent the classification of proteins with two, three, and more than four domains, respectively.

Method	2dom		3dom		m4dom		all	
	average	median	average	median	average	median	average	median
DeepAssembly	0.896	0.985	0.851	0.984	0.725	0.840	0.856	0.976
SADA	0.840	0.903	0.709	0.688	0.582	0.555	0.763	0.782
DEMO	0.779	0.782	0.649	0.628	0.517	0.504	0.702	0.698
AIDA	0.671	0.662	0.514	0.497	0.424	0.405	0.589	0.581

Supplementary Table 5. Summary of errors (Å) in inter-domain distances predicted by different methods. "2dom", "3dom", and "m4dom" represent the classification of proteins with two, three, and more than four domains, respectively.

Method	2dom		3dom		m4dom		all	
	average	median	average	median	average	median	average	median
DeepAssembly	0.629	0.344	0.515	0.396	0.383	0.251	0.560	0.343
AlphaFold2	0.833	0.462	0.597	0.520	0.548	0.483	0.724	0.476
RoseTTAFold	1.252	1.029	1.028	0.943	0.990	0.951	1.151	0.984

Supplementary Table 6. Ablation results of inter-domain distance prediction accuracy. "2dom", "3dom", and "m4dom" represent the classification of proteins with two, three, and more than four domains, respectively.

Model no.	Input features	2dom	3dom	m4dom	all
1	All	0.629	0.515	0.383	0.560
2	No inter-domain features	0.654	0.487	0.463	0.580
3	No PAAreader (use HHsearch templates)	0.692	0.571	0.658	0.655
4	No templates	1.700	1.492	1.551	1.621

Finally, we tried to use the DeepAssembly server but did not receive any feedback/results within a week - no information on queue status etc. was given. So the server might not be working yet.

Response: We apologize for the inconvenience caused by this situation. This happened because our server was deployed on a public cluster of the college, however, the cluster was undergoing maintenance upgrades during the previous period, which caused our server to be suspended and the submitted tasks were not running properly. After noticing this situation, we dealt with it timely and now our server is working properly (<http://zhanglab-bioinf.com/DeepAssembly/>). We look forward to your valuable suggestions.

The following is the task submission page for our server:

DeepAssembly On-line Server [View multidomain example of output] [Back to server]

Input the multidomain protein full-chain sequence in **FASTA format** (mandatory, Click for an example input).

```
>fasta
CGEILTESTGTIQSPGHVYYPHGINCTWHILVQPNHLIHLMFETFHLEFHYNCTNDYLEVYDTSSETSLGRYCG
KSIPPSLTSSGNSLMLVFDSDLAYEGFLINYEAIASAATAQLQDYDDDLGFTSPNFPNNYPNNWECYRITVRT
GQLIAVHFTNFSLEEAIQNYTDFLEIRDDGGYEKSPLLGIFYGNSLPTIISHSNKLLWLFKFSQDIDTRSGFSAYWD
GSSTGCGGNLTSSGTFISPNYPMPYYHSSECYWWLKSSHGSAFELEFKDFHLEHHPNCTLDYLAVYDGPSSN
SHLLTQLCGDQKPPILRSSGDSMFIKLRDDEGQQGRGFKAEYRQTCENVVIVNQTYGILESIGYPNPYSENQHC
NWTIRATTGNTVNYTFLAFDLEHHINCSTDYLELYDGPROMGRYCGVDLPPPGSTTSKLVLLTLDGVRREK
GFQMQWV
```

Or, upload the full-chain sequence file:
 未选择任何文件

Input the protein domain 1 in **PDB format** (Click for an example input).

```
ATOM 1 N MET A 1 -19.010 4.841 -6.288 1.00 60.49 N
ATOM 2 H MET A 1 -19.331 4.046 -5.755 1.00 60.49 H
ATOM 3 H2 MET A 1 -19.795 5.420 -6.550 1.00 60.49 H
ATOM 4 H3 MET A 1 -18.546 4.484 -7.111 1.00 60.49 H
ATOM 5 CA MET A 1 -18.048 5.591 -5.451 1.00 60.49 C
ATOM 6 HA MET A 1 -17.578 6.380 -6.037 1.00 60.49 H
ATOM 7 C MET A 1 -16.975 4.604 -5.040 1.00 60.49 C
ATOM 8 CB MET A 1 -18.723 6.218 -4.222 1.00 60.49 C
```

Or, upload the structural model file:
 未选择任何文件

Input the protein domain 2 in **PDB format** (Click for an example input).

```
ATOM 1355 N PRO A 91 -3.980 8.761 -12.659 1.00 85.70 N
ATOM 1356 CA PRO A 91 -3.166 7.617 -13.050 1.00 85.70 C
ATOM 1357 HA PRO A 91 -3.815 6.779 -13.304 1.00 85.70 H
ATOM 1358 C PRO A 91 -2.265 7.198 -11.880 1.00 85.70 C
ATOM 1359 CB PRO A 91 -2.393 8.047 -14.303 1.00 85.70 C
ATOM 1360 HB2 PRO A 91 -2.930 7.708 -15.189 1.00 85.70 H
ATOM 1361 HB3 PRO A 91 -1.375 7.659 -14.314 1.00 85.70 H
ATOM 1362 O PRO A 91 -1.928 8.011 -11.021 1.00 85.70 O
```

Task queue page:

Home Zhang Lab Server Example

[back to server]

DeepAssembly Results for job DeepAssembly573828833 (test)

Your job with name **"test"** has been successfully submitted and is being processed

This page reloads every 5 seconds and the results will be displayed when the task is complete.
You can bookmark this page to check the results later.

Result page:

[Back to server]

DeepAssembly Results for job example

[Click on example.tar.gz to download all result files]

Input Sequence and Domain Definition

>fasta

```
CGEILTESTGTIQSPGHNVYPHGINCTWHILVQPNHLIHLMFETHLEFHYNCTNDYLEVYDSTDSETSLGRYCGKSIPPSLTSSGNSLMLVFTDSDLA  
YEGFLINYEAIISAATAQLQDYTDLLGTFSTPNFPNNYPNNWECIYRITVVRTGQLIAVHFTNFSLEEAIGNYYTDFLEIRDDGGYEKSPLLGIFYGSNLPPT  
IISHSNKLWLKFKSDQIDTRSGFSAYWDGSSSTGCGGNLTTSSGTFISPNYMPYHSSECYWLLKSSHGSAFELEFKDFHLEHHPNCTLDYLAVYDGPSS  
NSHLLTQLCGDEKPPILIRSSGDSMFIKLRDDEGQQGRGFKAEYRQTCENVVIVNQTYGILESIGYPNPYSENQHCNWTIRATTGNTVNYTFLAFDLEHHI  
NCSTDYLELYDGRQMGRYCGVDLPPPGSTTSSKLQVLLLTGCVGRREKGFQMWFV
```

Individual Domain Structures

Final Full-length Structures

Reference:

[R1] Zhou, X. G., et al. Assembling multidomain protein structures through analogous global structural alignments. *Proc. Natl Acad. Sci. USA* **116**, 15930-15938 (2019).

[R2] Peng, C. X., et al. Structural analogue-based protein structure domain assembly assisted by deep learning. *Bioinformatics* **38**, 4513-4521 (2022).

[R3] Garcia, C. K., et al. Molecular characterization of a membrane transporter for lactate, pyruvate, and other monocarboxylates: Implications for the Cori cycle. *Cell* **76**, 865-873 (1994).

[R4] Pathania, M., et al. Unusual Constriction Zones in the Major Porins OmpU and OmpT from *Vibrio cholerae*. *Structure* **26**, 708-721 (2018).

[R5] Zhang, Y. & Skolnick, J. Scoring function for automated assessment of protein structure template quality. *Proteins* **57**, 702-710 (2004).

[R6] Xu, J. R. & Zhang, Y. How significant is a protein structure similarity with TM-score=0.5? *Bioinformatics* **26**, 889-895 (2010).

[R7] Basu, S. & Wallner, B. DockQ: A Quality Measure for Protein-Protein Docking Models. *PLoS ONE* **11**, e0161879 (2016).

[R8] Mendez, R., et al. Assessment of blind predictions of protein-protein interactions: Current status of docking methods. *Proteins* **52**, 51-67 (2003).

Reviewers' comments:

Reviewer #1 (Remarks to the Author):

The authors have properly addressed my concerns. I have no more comments.

Reviewer #2 (Remarks to the Author):

The manuscript at the moment improved to certain extent.

However, when we looked carefully for the results from Figure 2b & 8b I am afraid that there is almost no any correlations between TM-score and RMSD values on each test case between DeepAssembly and AlphaFold2. I am quite concerned for this point. It would be necessary to clarify this issue.

Moreover, for the prediction results in Figure 7, in my opinion there is a huge difference between experimental structures and predicted models. There are quite noticeable shifts even in TM regions. There are 190K experimental structures resolved in PDB, of which only 3-4% were membrane proteins. I could imagine that the training sets of the AI model is so limited. How reliable will the results of membrane protein predictions really be a big questionable mark. This critical question should be addressed and discussed in detail.

Reviewer #3 (Remarks to the Author):

The authors have gone to great lengths, it seems to answer all the reviewers' queries. My concerns have been adequately alleviated. However, it is a little hard to follow the paper as there are many sections, tests with different protein sets and it would be good to have a "bottom-line" message... "this program is (on average) best for x, y and z kind of system". In addition, it would be helpful to understand the limitations, also in an upfront manner. Specifically in running a protein on the authors' web-server, which now works, we noticed that sequence analysis can group several sub-domains together and only one AlphaFold prediction seems to be used for this group and then docked. Is there a way to manually define domain limits or have a more sensitive domain analysis, so that the subdomains are also introduced as docking partners?

Gui-Jun Zhang
College of Information Engineering
Zhejiang University of Technology
Hangzhou 310023, China
zgj@zjut.edu.cn
September 27, 2023

Dear Reviewers,

Thank you for reviewing our manuscript entitled “Domain-based multi-domain protein and complex structure prediction using inter-domain interactions from deep learning” (COMMSBIO-23-1703A) and for proposing valuable suggestions. The comments helped us improve our manuscript and served as important guidance to our research. We have carefully considered the comments and have revised the manuscript accordingly. All modified parts are highlighted in the manuscript.

Thank you again for reviewing our manuscript. We hope you find that the changes are satisfactory.

Sincerely,

Yuhao Xia, Kailong Zhao, Dong Liu, Xiaogen Zhou, and Guijun Zhang

Response to Reviewer #1

The authors have properly addressed my concerns. I have no more comments.

Response: We appreciate for your comments on this work. The comments helped us improve our manuscript and served as important guidance to our research.

Response to Reviewer #2

We very much appreciate for your comments and suggestions, which help to significantly improve the quality and description of our manuscript. In the following, we include point-by-point response to the comments, where all changes have been highlighted in the manuscript.

The manuscript at the moment improved to certain extent.

Response: We greatly appreciate your comments, which helped us improve our manuscript and served as important guidance to our research.

However, when we looked carefully for the results from Figure 2b & 8b I am afraid that there is almost no any correlations between TM-score and RMSD values on each test case between DeepAssembly and AlphaFold2. I am quit concern for this point. It would be necessary to clarify this issue.

Response: Thank you for raising this important point. We apologize for the unclear description in the manuscript.

(1) We present Figure 2b to show the accuracy comparison between DeepAssembly and AlphaFold2 on 219 multi-domain targets. Specifically, the left panel in Figure 2b shows a head-to-head comparison of the TM-scores on each test case between DeepAssembly and AlphaFold2, and the right panel shows a head-to-head comparison of the RMSDs on each case between DeepAssembly and AlphaFold2. The results show that DeepAssembly achieves a higher TM-score than AlphaFold2 on 66% of the test cases, and a lower RMSD on 67% cases.

(2) Similarly, Figure 8c shows a head-to-head comparison between DeepAssembly and other methods (SADA, DEMO and AIDA) based on TM-score of the predicted models for 219 multi-domain targets. Figure 8b shows the comparison between DeepAssembly, SADA, DEMO and AIDA on 219 multi-domain proteins based on the fraction of proteins with TM-score higher than each TM-score cutoff.

We found that there are some ambiguous descriptions in the manuscript, so we revised the corresponding descriptions in the manuscript to avoid misunderstandings.

Å, which are both better than 0.900 and 3.58 Å of AlphaFold2. We present in Fig. 2b the accuracy comparison between DeepAssembly and AlphaFold2 on each target. DeepAssembly achieves a higher TM-score than AlphaFold2 on 66% of the test cases, and a lower RMSD on 67% cases (Fig. 2b). Especially,

Fig. 2 | Results of assembling predicted domain structures. **a** Average TM-score over 219 multi-domain proteins for DeepAssembly and AlphaFold2. The y-axis on the right represents the full-chain RMSD of proteins. **b** Head-to-head comparison of the TM-scores (left panel) and RMSDs (right panel) on each test case between DeepAssembly and AlphaFold2. "n" refers to the number of points on either side of the diagonal. **c** Histogram of backbone RMSD for full-chain models by DeepAssembly and AlphaFold2. **d** TM-score comparisons between AlphaFold2 full-length model and its single domain parts (AF2 domain) on the corresponding proteins of the models predicted by AlphaFold2 with TM-score less than each cutoff. **e** TM-score comparisons between DeepAssembly and AlphaFold2 on the corresponding proteins of the models predicted by AlphaFold2 with TM-score less than each cutoff. "cutoff" refers to a TM-score value on the x-axis, which reflects a TM-score interval less than this value.

Moreover, for the prediction results in Figure 7, in my opinion there is a huge different between experimental structures and predicted models. There are quit noticeable shift even in TM regions. There are 190K experimental structures were resolved in PDB, of which only 3-4% were membrane proteins. I could imagine that the training sets of the AI model is so limited. How reliable will the results of membrane protein predictions is really a big questionable mark. This critical questions should be addressed and discussed in details.

Response: Thank you for raising the important question. Based on your valuable suggestions, we added the discussion about limits on membrane protein prediction and promising directions to address these challenges in the manuscript.

Membrane proteins play crucial roles in living cells, yet accurate prediction of their three-dimensional structures remains a challenge. Although we have made some explorations and attempts in structural modeling of membrane proteins, there is still room for further improvement. Since our deep learning model is only trained on the data set of mainly soluble multi-domain proteins, which are different from the structural characteristics of membrane proteins, our method still has some limitations for the assembly of some membrane proteins that are large or flexible across transmembrane regions. For example, for the target shown in Figure 7f, there is a deviation between experimental structure (gray) and predicted model (colored), especially in the transmembrane region.

Moreover, as the reviewer points out, the number of membrane proteins is rather limited compared with a great deal of soluble protein structures resolved in PDB, as their experimental determination is subject to the complicated membrane environment. This poses a major obstacle to training AI model directly on such a small dataset of membrane proteins [R1]. To address the challenge, some statistical and transfer learning-based methods may be promising, such as DeepTMP [R1], which predicts the transmembrane protein inter-chain contact by transferring the knowledge learned from the initial model trained on a large dataset of soluble protein complexes. This approach alleviates the limitation of training deep learning model directly on few membrane proteins, and simultaneously provides new insights into how we further improve the complex structure prediction by leveraging multi-domain protein data.

The following are the revisions we have made in the manuscript.

Fig. 7b and c show the results of DeepAssembly on two homo-oligomeric proteins with C2-symmetry type. The models predicted by DeepAssembly achieved DockQ scores of 0.881 (**Fig. 7b**) and 0.796 (**Fig. 7c**), respectively. Furthermore, for OmpU, an outer membrane protein of *Vibrio cholerae*⁴⁵ (PDB ID: 6EHB), DeepAssembly correctly predicted the structures of its three channels (average TM-score = 0.985), as well as their interfaces (DockQ = 0.591) (**Fig. 7d**). **Fig. 7e** displays an example of tetrameric potassium channel KcsA (PDB ID: 2QTO), which is an ion channel that is capable of selecting potassium ions. The tetramer model for KcsA built by DeepAssembly achieved a DockQ score of 0.756, and the model gave the correct overall shape of the pore used to filter out potassium ions (shown as sphere in **Fig. 7e**). These examples demonstrate the potential of DeepAssembly on transmembrane proteins, which may help to understand the function of transmembrane proteins, and provide insights into their molecular interaction mechanism.

However, there is still room for further improvement. Since our deep learning model is only trained on the data set of mainly soluble multi-domain proteins, which are different from the structural characteristics of membrane proteins, our method still has some limitations for the assembly of some membrane proteins that are large or flexible across transmembrane regions. For example, for the target of acid-sensing ion channel (PDB ID: 2QTS) with three chains, there is a deviation between experimental structure and predicted model (DockQ = 0.402), especially in the transmembrane region (Fig. 7f). This may be due to the highly flexible structures within the transmembrane region, with each subunit in the PDB structure of the chalice-shaped homotrimer having a different orientation between its transmembrane helices and extracellular region, even though they share the same amino acid sequence.

Moreover, the number of membrane proteins is rather limited compared with a great deal of soluble protein structures resolved in PDB, as their experimental determination is subject to the complicated membrane environment. This poses a major obstacle to training deep learning model directly on such a small dataset of membrane proteins⁴³. To address the challenge, some statistical and transfer learning-based methods may be promising, such as DeepTMP⁴³, which predicts the transmembrane protein inter-chain contact by transferring the knowledge learned from the initial model trained on a large dataset of soluble protein complexes. This approach alleviates the limitation of training deep learning model directly on few membrane proteins, and simultaneously provides new insights into how we further improve the complex structure prediction by leveraging multi-domain protein data.

Reference:

[R1] Lin, P., Yan, Y., Tao, H. et al. Deep transfer learning for inter-chain contact predictions of transmembrane protein complexes. *Nat Commun* **14**, 4935 (2023). <https://doi.org/10.1038/s41467-023-40426-3>

Response to Reviewer #3

We very much appreciate for your comments and suggestions, which help to significantly improve the quality and description of our manuscript. In the following, we include point-by-point response to the comments, where all changes have been highlighted in the manuscript.

The authors have gone to great lengths, it seems to answer all the reviewers queries. My concerns have been adequately alleviated.

Response: We very much appreciate for your comments on this work. The comments helped us improve our manuscript and served as important guidance to our research.

However, it is a little hard to follow the paper as there are many sections, tests with different protein sets and it would be good to have a "bottom-line" message..."this program is (on average) best for x, y and z kind of system". In addition, it would be helpful to understand the limitations, also in an upfront manner.

Response: Thank you for pointing out this important issue. We deeply apologize for the lack of clarity in the manuscript. Based on your valuable suggestions, we summarized the performance of our method on different test sets in the conclusion section, including its advantages and limitations.

In general, our method performs best on the test set of multi-domain proteins because this deep learning model is trained specifically on multi-domain proteins to accurately capture domain-domain interactions. In addition, the method exhibits more excellent performance for assembling multi-domain proteins with experimental domain structures. Moreover, we demonstrate that our method can be applied to protein complex structure prediction by using inter-domain interactions learned on monomeric multi-domain proteins. However, for targets without sufficient inter-chain co-evolutionary signals or suitable templates, our method still has a lot of room for improvement, because the inter-domain interactions learned only from monomeric multi-domain proteins are relatively limited after all. Finally, we apply our method to several transmembrane protein complex cases, and the results show that we could correctly predict the interface of some homo-oligomeric membrane protein complexes, but the performance for targets that are large or flexible across transmembrane regions is still not satisfactory. This is because the number of membrane proteins is rather limited and they have different structural characteristics, making it difficult for the model trained on mainly soluble proteins to effectively capture their inter-chain interactions.

The following are the revisions we have made in the manuscript.

In general, DeepAssembly performs best on the test set of multi-domain proteins because this deep learning model is trained specifically on multi-domain proteins to accurately capture domain-domain interactions. Combined with the single-domain enhanced by our PAtreader, DeepAssembly constructs multi-domain protein structure with higher average accuracy than that modeled directly by AlphaFold2 on the given set. We found that AlphaFold2's inaccurate prediction for multi-domain protein structures is mainly due to the incorrect inter-domain orientation. For cases where the AlphaFold2-predicted structure is not accurate, DeepAssembly tends to perform more satisfactorily. Meanwhile, DeepAssembly shows the potential to improve the accuracy of structures with low confidence deposited in AlphaFold database,

providing an important solution for AlphaFold database update. In addition, DeepAssembly exhibits more excellent performance for assembling multi-domain proteins with experimental domain structures.

Moreover, we demonstrate that DeepAssembly can be applied to protein complex structure prediction by using inter-domain interactions learned on multi-domain proteins. It provides a more lightweight way to assemble protein structures by treating domains as assembly units, reducing the requirements for computational resources to some extent. Despite the promising assembly results, the applicability and accuracy of DeepAssembly could be further improved in several aspects. Firstly, in the absence of sufficient co-evolutionary signals and appropriate templates, our method still has room for improvement in performance. In this case, the introduction of physical and chemical features, e.g., protein-protein interface preferences are also crucial. Secondly, the domain-domain interaction can be further extended, because the inter-domain interactions learned only from monomeric multi-domain proteins are relatively limited after all. We also apply DeepAssembly to transmembrane protein complexes. The results show that it could correctly predict the interface of some homo-oligomeric membrane proteins, but the performance for targets that are large or flexible across transmembrane regions is still not satisfactory. This is because the number of membrane proteins is rather limited and they have different structural characteristics, making it difficult for the model trained on mainly soluble proteins to effectively capture their inter-chain interactions. Address this challenge, it may be promising to borrow from some of the transfer learning methods.

Specifically in running a protein on the authors web-server, which now works, we noticed that sequence analysis can group several sub-domains together and only one alphafold prediction seems to be used for this group and then docked. Is there a way to manually define domain limits or have a more sensitive domain analysis, so that the subdomains are also be introduced as docking partners?

Response: Thank you very much, this is a great suggestion. This contributes to the wider impact of our web server. In DeepAssembly, the input sequence is first split into single-domain sequences through a domain segmentation tool, then the structure for each domain is generated by a single-domain structure predictor, PAtreader, and finally these single-domain structures are assembled into full-length structure model. Based on your valuable suggestions, we optimized our web server and added an option to manually define domain boundaries. If the user provides domain definition information, the web server will divide the input sequence according to the provided domain definition, and then assemble the corresponding predicted domain structures into full-length proteins.

The option for providing domain definition:

DeepAssembly On-line Server [View multidomain example of output] [Back to server]

Input the multidomain protein full-chain sequence in **FASTA format (mandatory)**, Click for an example input.

```
> 1prA
MANITVFYNEDFQGGKQVDLPPGNYTRAQLAALGIENNTISSVKVPPGVKAILYQNDGFAGDQIEVVANAEEELGPL
NNNVSSIRVISVPVQPRARFFYKEQFDGKEVDLPPGGYQAELEERYGIDNNTISSVKPQGLAVVLFKDNFSGD
TLPVNSDAPTLGAMNNNTSSIRIS
```

Or, upload the full-chain sequence file:
 未选择任何文件

▼ **Option: Provide your own domain definition.** ⓘ
 Please copy and paste your domain definition here. Sample input
 1-90;91-173;

Example:

(1) Input the multi-domain protein full-chain sequence in FASTA format.

DeepAssembly On-line Server [View multidomain example of output] [Back to server]

Input the multidomain protein full-chain sequence in **FASTA format (mandatory)**, Click for an example input.

```
> 1prA
MANITVFYNEDFQGGKQVDLPPGNYTRAQLAALGIENNTISSVKVPPGVKAILYQNDGFAGDQIEVVANAEEELGPL
NNNVSSIRVISVPVQPRARFFYKEQFDGKEVDLPPGGYQAELEERYGIDNNTISSVKPQGLAVVLFKDNFSGD
TLPVNSDAPTLGAMNNNTSSIRIS
```

Or, upload the full-chain sequence file:
 未选择任何文件

▼ **Option: Provide your own domain definition.** ⓘ
 Please copy and paste your domain definition here. Sample input
 1-90;91-173;

(2) Input your own domain definition in the following format.

DeepAssembly On-line Server [View multidomain example of output] [Back to server]

Input the multidomain protein full-chain sequence in **FASTA format (mandatory)**, Click for an example input.

```
> 1prA
MANITVFYNEDFQGGKQVDLPPGNYTRAQLAALGIENNTISSVKVPPGVKAILYQNDGFAGDQIEVVANAEEELGPL
NNNVSSIRVISVPVQPRARFFYKEQFDGKEVDLPPGGYQAELEERYGIDNNTISSVKPQGLAVVLFKDNFSGD
TLPVNSDAPTLGAMNNNTSSIRIS
```

Or, upload the full-chain sequence file:
 未选择任何文件

▼ **Option: Provide your own domain definition.** ⓘ
 Please copy and paste your domain definition here. Sample input
 1-90;91-173;

(3) Input the Email for receiving results.

Click the "Add model" and "Remove model" buttons below to add or remove structural models.

Option:

Email: (optional, where results will be sent to)

Job name: (optional, your given name to this job)

(4) Submit and confirm.

Click the "Add model" and "Remove model" buttons below to add or remove structural models.

Option:

Email: (optional, where results will be sent to)

Job name: (optional, your given name to this job)

the program to not work properly.

Or, upload the structural model file:
 未选择任何文件

Click the "Add model" and "Remove model" buttons below to add or remove structural models.

Option:

Email: (optional, where results will be sent to)

Job name: (optional, your given name to this job)

success

Please confirm your job and the results will be sent to your email!

Job name:
#Domains: 0
Email: xiayh@zjut.edu.cn

Results:

Input Sequence and Domain Definition

```
>fasta
MANITVFYNEDFQGGKQVDLPPGNYTRAQLAALGIENNTISSVKVPPGVKAILYQNDGFAGDQIEVVANAABELGPLNNVSSIRVISVPVQPRARFFYKEQ
FDGKEVDLPPGQYTQABLERYGIDNNTISSVKPQLAVLFLKDNFSGDTLPVNSDAPTLGAMNNNTSSIRIS
```

0 20 40 60 80 100

Individual Domain Structures

Structure of domain 1 (dom1.pdb)

Structure of domain 2 (dom2.pdb)

Final Full-length Structures

model 1 download model

Local IDDT score

Per-residue IDDT (Global IDDT: 79.52)

REVIEWERS' COMMENTS:

Reviewer #2 (Remarks to the Author):

My concerns were address. I have no more comments.

Reviewer #3 (Remarks to the Author):

The reviewer's concerns have been adequately addressed in this revision.